# RetroInText: A Multimodal Large Language Model Enhanced Framework for Retrosynthetic Planning via In-Context Representation Learning

**Chenglong Kang**[1]    **Xiaoyi Liu**[2*]    **Fei Guo**[1*]

[1]School of Computer Science and Engineering, Central South University

[2]School of Chinese Materia Medica, Beijing University of Chinese Medicine

`xiaoyi.liu@bucm.edu.cn`
`guofei@csu.edu.cn`

## Abstract

Development of robust and effective strategies for retrosynthetic planning requires a deep understanding of the synthesis process. A critical step in achieving this goal is accurately identifying synthetic intermediates. Current machine learning-based methods often overlook the valuable context from the overall route, focusing only on predicting reactants from the product, requiring cost annotations for every reaction step, and ignoring the multi-faced nature of molecular, resulting in inaccurate synthetic route predictions. Therefore, we introduce RetroInText, an advanced end-to-end framework based on a multimodal Large Language Model (LLM), featuring in-context learning with TEXT descriptions of synthetic routes. First, RetroInText including ChatGPT presents detailed descriptions of the reaction procedure. It learns the distinct compound representations in parallel with corresponding molecule encoders to extract multi-modal representations including 3D features. Subsequently, we propose an attention-based mechanism that offers a fusion module to complement these multi-modal representations with in-context learning and a fine-tuned language model for a single-step model. As a result, RetroInText accurately represents and effectively captures the complex relationship between molecules and the synthetic route. In experiments on the USPTO pathways dataset RetroBench, RetroInText outperforms state-of-the-art methods, achieving up to a $5\%$ improvement in Top-1 test accuracy, particularly for long synthetic routes. These results demonstrate the superiority of RetroInText by integrating with context information over routes. They also demonstrate its potential for advancing pathway design and facilitating the development of organic chemistry. Code is available at `https://github.com/guofei-tju/RetroInText`.

## 1 Introduction

Retrosynthesis stands as an essential strategy in organic chemistry, critically important for advancements in drug discovery and chemical biology (Corey, 1991; Zheng et al., 2022). Single-step retrosynthesis refers to predicting a single reaction step that breaks down a target molecule into simpler, more accessible precursors (Jiang et al., 2023). Recent advances in deep learning have facilitated the development of various approaches in single-step retrosynthesis, which can be categorized as template-based, semi-template-based, and template-free approaches (Zhong et al., 2023; Obonyo et al., 2023; Chen et al., 2020; Coley et al., 2017). Specifically, significant advancements have been achieved in single-step models through encompassing sequence-based, graph-based, and text-based methods. Sequence-based strategies leverage Simplified Molecular Input Line Entry System (SMILES) notation to represent reactions as sequential tokens, facilitating transformer-based architectures to predict the precursors for a target molecule. Graph-based approaches, conversely,

---

*Corresponding Author.

prioritize explicit molecular graph representations to model atomic connectivity and bond dynamics, enabling precise identification of reaction centers and transformation patterns. Graph Neural Networks (GNNs) have emerged as particularly effective tools in this domain due to their capacity to encode topological and physicochemical features. From a text-based perspective, inspired by Natural Language Processing (NLP) paradigms, text information associated with reactions is utilized to augment the model's comprehension of reaction mechanisms or to facilitate the reordering of single-step reaction predictions based on the insights derived from the textual data. Nevertheless, existing methods primarily rely on graph or SMILES representations, and are often limited in capturing the intricate complexities of chemical structures, thereby constraining their scalability and effectiveness in addressing complex retrosynthetic challenges.

Multi-step retrosynthesis planning is a fundamental strategy in organic chemistry, crucial for drug discovery and chemical biology, as it systematically breaks down complex target molecules into simpler, easily accessible precursors (Zheng et al., 2022; Zhong et al., 2023). Most existing retrosynthetic planning strategies (Tripp et al., 2024; Liu et al., 2024c) conceptualize retrosynthetic planning as a search problem, where the synthetic route is represented as a tree or graph, with molecules as nodes. However, a significant limitation of these approaches lies in their reliance on heuristic search algorithms to determine which nodes (molecules) should be expanded. This dependency often leads to several critical challenges, such as ensuring that the expanded nodes are commercially available compounds, avoiding computational inefficiencies, and maintaining the overall feasibility of the synthetic routes (Liu et al., 2023a). Additionally, due to the complex chemical space, each molecule can exhibit a large number of potential transformations—up to 10K (Szymkuć et al., 2016). The depth of the search tree, which corresponds to the route length, often varies between 10 to 20 steps, depending on the complexity of the target molecule (Obonyo et al., 2023). This vast combinatorial space, combined with the scarcity of high-quality structure data for retrosynthetic tasks, limits current methods' ability to effectively explore and prioritize routes, leading to inefficiencies and sub-optimal solutions.

Inspired by the success of Large Language Models (LLM), which trained on extensive text corpora, generating coherent text encompassing a wide range of topics and sentiments (Ying et al., 2021). Liu et al. (Liu et al., 2024d) introduce a text-assisted retrosynthesis planning method that utilizes pre-trained language models to aid reactant generation. In addition, Bran et al. (M. Bran et al., 2024) propose ChemCrow, by integrating 17 expert-designed tools, ChemCrow enhances the LLM performance in chemistry. However, the current LLM methods do not include a route length adjustment to guide future searches (Liu et al., 2023a). Despite these advancements, current LLM-based methods exhibit notable limitations, particularly the lack of an effective adaptation mechanism for route length, which is critical for guiding retrosynthetic planning (Liu et al., 2023a).

In particular, we first use ChatGPT to obtain a description of the entire pathway, starting with the target product based on its name for single-step retrosynthesis. This textual description, along with the molecular 3D geometry information is used as input information for training. For each selection step in multi-step retrosynthesis, we introduce multiple value functions, such as Synthetic Complexity Score (ScScore) (Coley et al., 2018) and the text captioning score to rank candidate reactants. We employ an existing pre-trained MolT5 as our single-step approach to intermediate prediction. Therefore, RetroInText is a context-aware model that integrates molecular captioning and context embeddings. RetroInText utilizes contextual information from previous steps for the entire pathway, thereby enhancing retrosynthesis prediction accuracy.

We evaluated RetroInText on the RetroBench dataset constructed by Liu et al. (Liu et al., 2023a), determining all possible synthetic routes for each target, resulting in a comprehensive set of routes for $128, 469$ molecules. RetroBench dataset is constructed based on the USPTO-full dataset (Chen et al., 2020). Extensive experimental results on retrosynthetic planning tasks demonstrate that RetroInText outperforms template-free baselines, achieving up to a $5\%$ improvement in Top-1 test accuracy. Additionally, ablation experiments confirm the effectiveness of textual information and LLM. We highlight our main contributions as follows:

- We propose the RetroInText framework as a template-free approach for retrosynthesis prediction. When predicting subsequent steps in retrosynthesis, this framework integrates in-context textual information from previous steps.
- With RetroInText, we leverage the advantage of LLM and ChatGPT as our generative models and evaluate the reactions based on their molecular descriptions. A combination of

textual information, molecular graphs, and 3D geometry information is used to select the optimal molecule in the selection phase.

- Extensive experiments have demonstrated that RetroInText achieves a competitive level of performance. Furthermore, RetroInText is tested in experiments to show its ability to predict complex reactions.

## 2 RELATED WORK

**Single Step Retrosynthesis.** Existing single-step retrosynthesis methods are categorized into template-based, semi-template-based, and template-free approaches. Template-based methods extract reaction templates from chemical reaction databases and model retrosynthesis as a classification or template retrieval task, mapping the product to reactants using predicted templates (Gaiński et al., 2024; Chen & Jung, 2021; Xie et al., 2023; Zhang et al., 2024a). Semi-template-based methods decompose the retrosynthesis problem into two steps. Including identifying reaction centers to generate synthons, and converting these synthons into reactants using generative models or adding leaving groups (Zhong et al., 2023; Somnath et al., 2021; Zhu et al., 2023; Lan et al., 2024). Template-free methods treat retrosynthesis as either a sequence-to-sequence task using SMILES or a graph-editing task to modify atoms and bonds (Igashov et al., 2024; Andronov et al., 2024; Laabid et al., 2024; Yao et al., 2024; Liu et al., 2024d; Zhang et al., 2024b). With the development of multimodal LLMs, reasoning capabilities are being extended to retrosynthesis (M. Bran et al., 2024). Although textual information from LLM such as ChatGPT has been employed in single-step retrosynthesis models (Qian et al., 2023; Liu et al., 2024d), its integration into multi-step retrosynthesis processes remains unexplored (Christofidellis et al., 2023; Liu et al., 2023b).

**Retrosynthesis Planning.** Retrosynthesis planning employs search algorithms to identify optimal candidates from single-step model predictions iteratively until all target compounds are sourced from existing commercial suppliers (Liu et al., 2023c; Zhao et al., 2023; Liu et al., 2024a; Zhang et al., 2024b; Zeng et al., 2024). These search algorithms can broadly be categorized into several types: Monte Carlo Tree Search (MCTS) employs a policy network to enhance retrosynthetic planning efficiency by effectively exploring and navigating the solution space (Segler et al., 2018). Retro* (Chen et al., 2020) proposes an AND-OR retrosynthesis planning model using an A*-like heuristic, where OR nodes (reactions) require any child, and AND nodes (products) require all children. Modeling retrosynthesis planning as an AND-OR tree has proven sound and effective. Recent works have focused on developing active frameworks (Torren-Peraire et al., 2024; Yuan et al., 2024) and new evaluation methods (Tripp et al., 2024; Tian et al., 2024; Maziarz et al., 2024). For example, (Schreck et al., 2019) and (Liu et al.) assign a uniform cost of 1 to each reaction, optimizing for the shortest route. However, shorter routes may result in lower yields compared to longer routes. Consequently, (Liu et al., 2023a) propose a novel single-step approach based on a conventional search algorithm, but it lacks an adaptation mechanism for route length and full-route information. The aforementioned methodologies require the annotation of costs for every reaction step, and incorporating reliable reaction quality data from chemists or laboratory experimentation entails significant expenses. As a result, these approaches often become economically impractical.

## 3 PRELIMINARY

### 3.1 SINGLE-STEP RETROSYNTHESIS

Define the space of all molecules as $\mathcal{M}$. The single-step retrosynthesis aims to input a target molecule $T \in \mathcal{M}$, resulting in a prediction of the potential reactions and their related reactants as outcomes. We denote it as an injection:

$$O(\cdot): \quad T \to \{R_i, \mathcal{I}_i, c(R_i)\}_{i=1}^k, \tag{1}$$

where $O(\cdot)$ represents the single-step model, which outputs at most $k$ reactions $R_i$ with their following reactant sets $\mathcal{I}_i$ and costs $c(R_i)$. The costs can be the actual price of the reaction or just a negative log-likelihood of this reaction under the model.

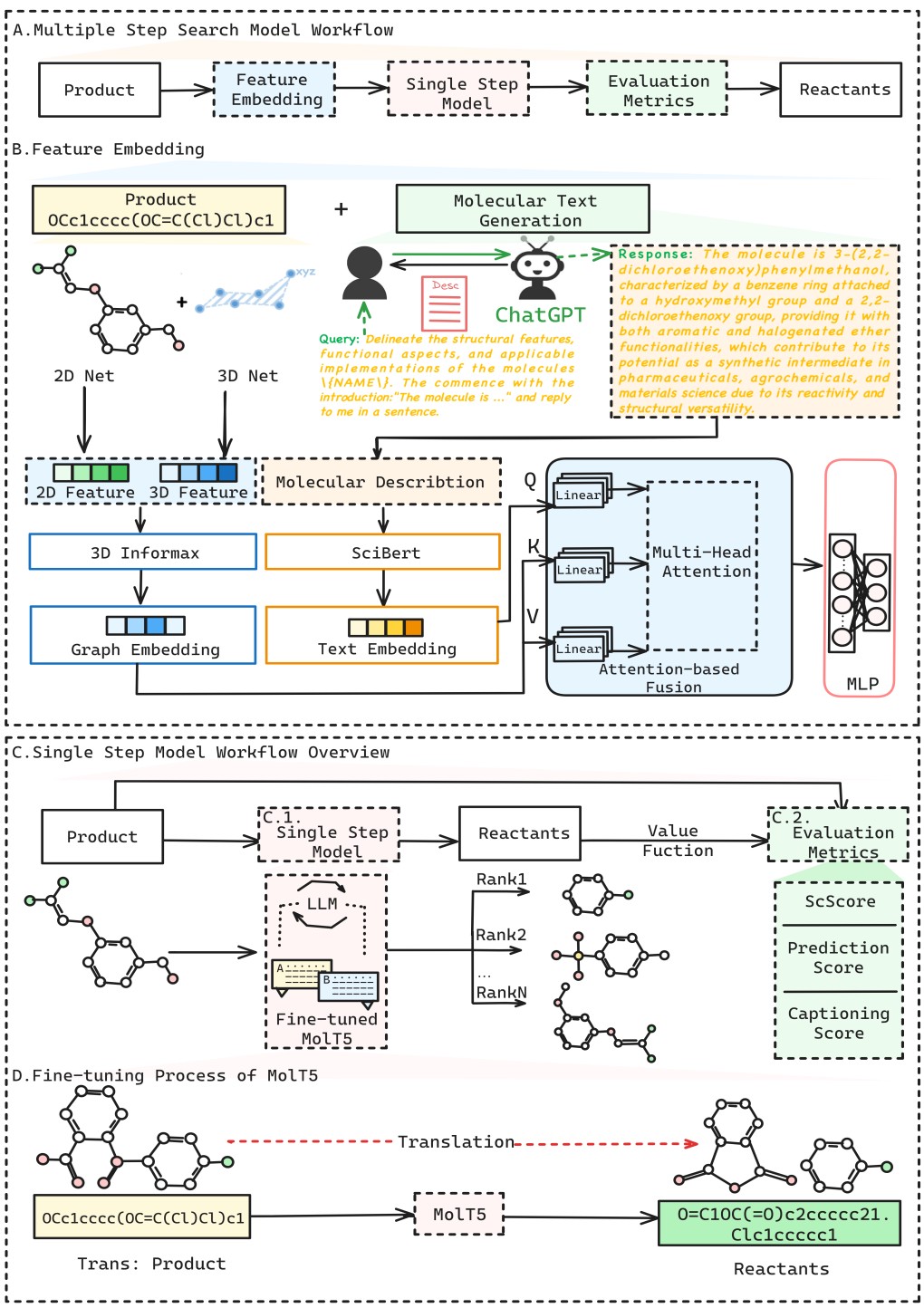

Figure 1: **Overview of the RetroInText.** **A** Multiple Step Search Model Workflow of RetroInText. **B** Feature Embedding. The product is represented as a molecular graph and 3D geometry features. It is combined with text embeddings generated by ChatGPT and processed through SciBERT for multimodal integration. **C** Single Step Model Workflow. **C.1** A fine-tuned MolT5 model generates potential reactants from the product, ranked by **C.2** Evaluation Metrics. Reactants are evaluated using ScScore, captioning score, and prediction score to determine synthetic routes' quality and feasibility. **D** MolT5 transforms the product SMILES into potential reactant structures.

## 3.2 Retrosynthetic Scoring Method

The goal of retrosynthetic planning is to find a series of reactions that transform the starting material set $\mathcal{S} \subseteq \mathcal{M}$ to the target molecule $M_t \in \mathcal{M}$:

$$M_t \to I \to \mathcal{S}, \tag{2}$$

$I = \{m_1 \ldots, m_j\} \subseteq \mathcal{M} \setminus \mathcal{S}$ stands for the set of intermediate molecules. Beginning with the target molecule $M_t$, current strategies perform series single-step retrosynthesis predictions by model $O(\cdot)$ until all molecules at the leaf nodes are from $\mathcal{S}$, form pathways to synthesis $M_t$, which can be formulated as:

$$\mathcal{P} = \{p_1, p_2, ..., p_n\}, \tag{3}$$

where $\mathcal{P}$ represents the set of pathways to synthesis $M_t$.

## 4 METHODOLOGY

As shown in Figure. 1, our proposed framework RetroInText incorporates a pre-trained molecular representation model 3DInfomax (Stärk et al., 2022), which is utilized to embed both molecular graph and 3D structural information. ChatGPT-3.5 is applied for generating contextual text information refers to captioning score along the multi-step pathway. Additionally, we propose an attention-based mechanism that offers a fusion module to complement these multi-modal representations with in-context learning and a fine-tuned language model MolT5 as a single-step retrosynthesis model.

### 4.1 RETROSYNTHESIS MODEL

#### 4.1.1 SINGLE-STEP MODEL

We adopt MolT5 (Edwards et al., 2022) as our single-step model $O(\cdot)$. Specifically, MolT5 is a transformer-based model with an encoder-decoder architecture based on the T5 model, pre-trained on 100 million molecular SMILES as well as a C4 dataset which contains 700G textual data. The model is suited to generation tasks such as molecular captioning which have contained a wealth of molecular and textual information. However, to adapt the model to our task, we apply a translation-based approach to fine-tune the model. Specifically, as shown in Figure. 1 D, we extract all reactions from the training set and treat the products and reactants in SMILES as two distinct "languages" for translation:

$$translation : products \to reactants, \tag{4}$$

where $products$ is the intermediate molecule during retrosynthetic planning, while the $reactants$ represent the corresponding reactant molecules. The details of fine-tuning MolT5 model parameters are provided in Appendix A.3. The model is equipped with the capability to handle retrosynthesis tasks. We use it as our single-step model in the expansion phase to predict Top-$k$ reactions and their corresponding reactants. The results of the single-step models can be seen in Appendix C.1.

#### 4.1.2 MOLECULAR REPRESENTATION

**Molecule Graph Encoder.** As depicted in Figure. 1 B, we use 3DInfomax as the molecular graph and 3D encoder. The 3DInfomax model consists of a 2D GNN and a 3D GNN, utilizing a contrastive learning approach in training. It aligns the molecular graphs with the 3D conformations, maximizing the mutual information between the 2D GNN and the 3D conformation GNN, allowing the model to leverage both molecular structure and 3D conformation information simultaneously. We apply 3DInfomax in the selection process to fully utilize both molecular structure and 3D conformation information. The molecule is represented as a graph $\mathcal{G} = (\mathcal{V}, \mathcal{E})$, where $\mathcal{V}$ and $\mathcal{E}$ stands for the set of molecule nodes and edges respectively. RetroInText also includes information about the molecule's conformation as 3D cloud points $\{x_1, \cdots, x_m\} \subset \mathbb{R}^3$. Then we use 3DInfomax as the $M\_Encoder$ of the graph to get the molecular model:

$$\boldsymbol{H}_m = M\_Encoder(\mathcal{G}), \tag{5}$$

where $\boldsymbol{H}_m \in \mathbb{R}^d$, wherein $d$ represents the output dimension of the model and the $\mathcal{G}$ corresponds to the graph representation of the intermediate molecules.

**Textual Generator and Encoder.** We utilize ChatGPT to generate text. In detail, based on the IUPAC names of the products and intermediates, textual description is generated of the intermediate molecules along all pathways using ChatGPT. Chemical structures are uniquely represented by IUPAC names, which are derived from a set of rules mapping structures to linguistic phrases. Chemical structures described by IUPAC names are more natural and language-like than those described by SMILES. IUPAC names serve as a bridge between chemical molecules and LLMs. Details can be seen in Appendix B. We use the following prompt to generate textual descriptions:

> *Describe the key transition states involved in the synthesis of {{products}} from the intermediates {{intermediates}}. Explain the structural changes and energy barriers for each transition state, and reply to me in a sentence.*

where {{products}} corresponds to the IUPAC name of the product, and {{intermediates}} corresponds to the IUPAC names of all intermediate molecules. In instances involving multiple intermediate molecules, they are concatenated with commas. As shown in Figure. 1 B, after obtaining text information from the pathway to the target molecule $\mathcal{T} = \{t_1, t_2, \ldots, t_n\}$, SciBERT (Beltagy et al., 2019) is used as $T\_Encoder$ for the textual modal.

$$\boldsymbol{H}_t = T\_Encoder(\mathcal{T}), \tag{6}$$

To ensure no information leakage and to eliminate variations in the textual content generated by ChatGPT in the test phase, we use only the structural information of the product molecules as the textual information for each step. This also ensures that the selected intermediates remain closely aligned with the product molecules. We generate textual descriptions using the following prompts:

> *Delineate the structural features, functional aspects, and applicable implementations of the molecules {NAME}. They commence with the introduction: "The molecule is ..." and reply to me in a sentence.*

where {NAME} corresponds to the IUPAC names of the products. SciBERT is also used as the encoder for textual information.

**Multi-modal Fusion.** As shown in Figure. 1 B, the fusion of molecular and textual representations is achieved through an attention mechanism. Specifically, while obtaining the molecular representations $\boldsymbol{H}_m$ and textual representations $\boldsymbol{H}_t$, the textual information is utilized as the query (Q), while the molecular representations are treated as the key (K) and value (V). This approach facilitates the integration of both modalities, allowing for a more effective alignment and interaction between them, thereby enhancing the overall predictive capability of the model.

$$\boldsymbol{Q} = \boldsymbol{H}_t \boldsymbol{W}^Q, \quad \boldsymbol{K} = \boldsymbol{H}_m \boldsymbol{W}^K, \quad \boldsymbol{V} = \boldsymbol{H}_m \boldsymbol{W}^V, \tag{7}$$

$$Attention = \text{softmax}\left(\frac{\boldsymbol{Q}\boldsymbol{K}^T}{\sqrt{d_k}}\right)\boldsymbol{V}, \tag{8}$$

$$\boldsymbol{H}_f = Attention(\boldsymbol{Q}, \boldsymbol{K}, \boldsymbol{V}), \tag{9}$$

where $\boldsymbol{W}^Q \in \mathbb{R}^{d \times d_q}, \boldsymbol{W}^K \in \mathbb{R}^{d \times d_k}$ and $\boldsymbol{W}^V \in \mathbb{R}^{d \times d_v}$ are trainable parameters, $\boldsymbol{H}_f$ represents the fused representation.

### 4.1.3 MODEL TRAINING

The fusion module is integrated into the model's training process, allowing it to effectively handle scenarios where textual information is included in the testing phase:

$$\hat{\boldsymbol{y}} = MLP(\boldsymbol{H}_f), \tag{10}$$

$$Loss = \frac{1}{n}\sum_{i=1}^{n}(\boldsymbol{y}_i - \hat{\boldsymbol{y}}_i)^2, \tag{11}$$

where $\hat{\boldsymbol{y}}$ represents the model's prediction score, and the model is trained using the Mean Squared Error (MSE) loss.

## 4.2 Scoring Method for Guiding the Search

To guide the retrosynthesis search process, we employ a scoring framework that effectively ranks candidate pathways by combining synthetic complexity, reaction costs, and textual alignment. The scoring framework consists of three components: ScScore (Coley et al., 2018), reaction cost score, and captioning score.

**Synthetic Complexity and Reaction Cost Scores.** The ScScore, ranging from 1 to 5, quantifies molecular complexity while considering synthetic accessibility (Coley et al., 2018). For a retrosynthesis pathway, the synthetic complexity score $V_t$ is defined in Equation 12, where $\mathcal{I}_i$ denotes the $i$-th intermediate molecule, and $n$ represents the total number of intermediates. This normalization ensures that lower scores correspond to simpler and more accessible intermediates.

The reaction cost score $V_m$, as defined in Equation 13, evaluates the cumulative cost of reactions within the pathway, where $c(R_i)$ reflects the reaction cost for $R_i$, the reaction producing the intermediate molecule. This metric accounts for the feasibility and efficiency of the associated chemical transformations.

The overall pathway score $V$, defined in Equation 14, combines the synthetic complexity and reaction cost scores, prioritizing pathways that are both synthetically accessible and cost-efficient.

$$V_t = \sum_{i=1}^{n} (1 - \frac{ScScore(\mathcal{I}_i) - 1}{4}), \tag{12}$$

$$V_m = \sum_{i=1}^{n} c(R_i), \tag{13}$$

$$V = V_t + V_m. \tag{14}$$

**Captioning Score for Pathway Selection.**

In addition to the key scoring components, we integrate the captioning score in the selection phase to enhance the selection process. The captioning score leverages textual descriptions generated during retrosynthesis planning to evaluate the alignment between the descriptions of intermediate molecules and the overall pathway context. This alignment provides an additional layer of interpretability and ensures the textual coherence of selected pathways.

For training, ScScore, reaction cost score, and textual alignment are treated as true values, allowing the model to learn a unified scoring strategy. At inference, the combined scoring framework, including the captioning score, refines pathway ranking by ensuring both chemical feasibility and contextual consistency.

The inference process is summarized in Algorithm 1, where the scoring framework ranks pathways and guides molecule expansion. This integrated approach enables RetroInText to effectively identify retrosynthesis pathways that are optimal across multiple dimensions.

---

**Algorithm 1** Retrosynthesis Planning Algorithm

**Input:** target molecule $M_t$, starting material set $S$, textual information $\mathcal{T}$
**Initialize:** reactants set $\mathcal{R} = \{\}$, path set $\mathcal{P} = \{M_t\}$
**while** $\mathcal{P}$ is not empty **do**
    Take path $p$ from $\mathcal{P}$, predict reactants $\mathcal{I}_p$ for expansion given $p$ by $O(\cdot)$
    **for** reactant $\mathcal{I}_p^{(i)}$ in $\mathcal{I}_p$ **do**
        **if** $\mathcal{I}_p^{(i)} \in \mathcal{S}$ **then** Put $\mathcal{I}_p^{(i)}$ into $\mathcal{R}$
        **else**
            rank $p' = p + [\mathcal{I}_p^{(i)}]$ by computing captioning score of $\mathcal{T}$
            put ranked $p'$ into $\mathcal{P}$
        **end if**
    **end for**
**end while**
**return** predicted reactant set $\mathcal{R}$

---

## 5 EXPERIMENTS

### 5.1 EXPERIMENTAL SETUP

**Dataset.** As shown in Table. 1, we use the public dataset RetroBench (Liu et al., 2023a) for evaluation, which includes $46,458$ molecules as the training set, $5,803$ molecules as the validation set and $5,838$ molecules as the testing dataset. The synthetic pathways for each molecule are extracted from the USTPO-full reaction network. All reactions along the pathways for each molecule in the training and validation set are extracted to fine-tune the MolT5 (Edwards et al., 2022) model.

| #Molecules ＼ Depth Datasets | 2 | 3 | 4 | 5 | 6 | 7 | 8 | 9 | 10 | 11 | 12 | 13 |
|---|---|---|---|---|---|---|---|---|---|---|---|---|
| Training | 22,903 | 12,004 | 5,849 | 3,268 | 1,432 | 594 | 276 | 107 | 25 | 0 | 0 | 0 |
| Validation | 2,862 | 1,500 | 731 | 408 | 179 | 74 | 34 | 13 | 2 | 0 | 0 | 0 |
| Test | 2,862 | 1,500 | 731 | 408 | 179 | 74 | 34 | 13 | 2 | 32 | 2 | 1 |

Table 1: Statistics of molecules at various depths summarized from the RetroBench dataset.

**Baselines.** Retrosynthetic planning strategies integrate retrosynthesis models with search algorithms. We compare our model with template-based models, including Retrosim (Coley et al., 2017), Neuralsym (Segler & Waller, 2017), and GLN (Dai et al., 2019). We also compare with template-free models, such as Transformer (Karpov et al., 2019) Megan (Sacha et al., 2021) and FusionRetro (Liu et al., 2023a), as well as semi-template-based models, including G2Gs (Shi et al., 2020) and GraphRetro (Somnath et al., 2021). Additionally, we compare RetroInText with FusionRetro+CREBM (Liu et al., 2024b) that incorporate energy functions for reranking. In detail, CREBM is a framework that enhances molecule synthesis by integrating energy functions to evaluate and rerank synthetic routes, thereby improving the quality of the generated pathways. Upon completion of the retrosynthesis training, we employ the first A*-like algorithm guided AND-OR tree search methods Retro* (Chen et al., 2020), Retro*-0, which is indeed a beam search algorithm, and Greedy DFS search algorithms.

**Evaluation Metrics.** We utilize the commonly employed evaluation performance metrics Top-$k$ ($k = 1, 2, 3, 4, 5$) exact match accuracy to evaluate the retrosynthesis performance proposed by Liu et al. (2023a). The exact match accuracy is computed by comparing predicted reactants SMILES to the dataset's ground truth on the benchmark dataset. More experimental setups can be found in the Appendix A.

### 5.2 RESULTS

**Comparison with Baselines.** The performance of all methods is presented in Table. 2. Compared with all template-free models and the reranked CREAM model and the State-Of-The-Art(SOTA) model FusitonRetro (Liu et al., 2023a), our model RetroInText achieves the best performance, exceeding the Top-1 accuracy of FusionRetro with CREBM by $1.8\%$, achieving SOTA performance. RetroInText also demonstrates superior performance across different search algorithms, even approaching the top results of template-based methods with Retro*-0 and Greedy DFS, highlighting the benefits of using LLM and route description.

**Analysis for the Depth of Routes.** To better evaluate the performance of our proposed model across varying levels of retrosynthetic complexity, we analyze the prediction accuracy at different depths using Greedy DFS, as shown in Figure. 2. Our model RetroInText demonstrates competitive performance across different depths, particularly excelling in longer synthesis routes. Compared to other baselines, our model maintains a more stable decline with increasing depth. While models like GraphRetro and Megan sharply drop beyond depth $4$, RetroInText retains a significant margin, demonstrating robustness and effectiveness in deeper, more complex retrosynthetic planning.

**Ablation Experiments.** To better understand the contribution of each component within our proposed framework, we conduct a series of ablation experiments. As shown in Table. 3, our model RetroInText, consistently outperforms the baseline model across all Top-N accuracy metrics, demonstrating the effectiveness of our proposed enhancements. For instance, in terms of Top-1 accu-

| Search Algorithm | Retro* | | | | | Retro*-0 | | | | | Greedy DFS |
|---|---|---|---|---|---|---|---|---|---|---|---|
| Single-step Models | Top-1 | Top-2 | Top-3 | Top-4 | Top-5 | Top-1 | Top-2 | Top-3 | Top-4 | Top-5 | Top-1 |
| Template-based | | | | | | | | | | | |
| Retrosim (Coley et al., 2017) | 35.1 | 40.5 | 42.9 | 44.0 | 44.6 | 35.0 | 40.5 | 43.0 | 44.1 | 44.6 | 31.5 |
| Neuralsym (Segler & Waller, 2017) | **41.7** | **49.2** | 52.1 | 53.6 | 54.4 | **42.0** | **49.3** | 52.0 | 53.6 | 54.3 | **39.2** |
| GLN (Dai et al., 2019) | 39.6 | 48.9 | **52.7** | **54.6** | **55.7** | 39.5 | 48.7 | **52.6** | **54.5** | **55.6** | 38.0 |
| Semi-template-based | | | | | | | | | | | |
| G2Gs (Shi et al., 2020) | 5.4 | 8.3 | 9.9 | 10.9 | 11.7 | 4.2 | 6.5 | 7.6 | 8.3 | 8.9 | 3.8 |
| GraphRetro (Somnath et al., 2021) | 15.3 | 19.5 | 21.0 | 21.9 | 22.4 | 15.3 | 19.5 | 21.0 | 21.9 | 22.2 | **14.4** |
| GraphRetro+CREBM (Liu et al., 2024b) | **16.3** | **20.1** | **21.6** | **22.3** | **22.7** | **16.3** | **20.2** | **21.6** | **22.3** | **22.7** | - |
| Template-free | | | | | | | | | | | |
| Transformer (Karpov et al., 2019) | 31.3 | 40.4 | 44.7 | 47.2 | 48.9 | 31.2 | 40.5 | 45.1 | 47.3 | 48.7 | 26.7 |
| Transformer+CREBM | 35.0 | 43.4 | 46.7 | 48.7 | 49.7 | 34.0 | 43.1 | 46.4 | 48.3 | 49.4 | - |
| Megan (Sacha et al., 2021) | 18.8 | 27.9 | 32.7 | 36.6 | 38.1 | 18.6 | 27.7 | 32.6 | 36.4 | 38.5 | 32.9 |
| FusionRetro (Liu et al., 2023a) | 37.5 | 45.0 | 48.3 | 50.6 | 51.5 | 37.4 | 45.0 | 48.4 | 50.4 | 51.1 | 35.2 |
| FusionRetro+CREBM (Liu et al., 2024b) | 39.4 | 46.6 | 49.3 | 50.7 | 51.5 | 39.6 | 46.7 | 49.5 | 51.0 | 51.7 | 33.8 |
| **RetroInText (Ours)** | **41.2** | **48.7** | **51.8** | **53.3** | **54.2** | **42.1** | **49.9** | **53.0** | **54.7** | **55.7** | **39.8** |

Table 2: Summary of retrosynthetic planning results for exact match accuracy (%).

racy, RetroInText achieves a $4.0\%$ increase over MolT5(SMILES). Similarly, compared to RetroInText(Graph), where we test using FusionRetro (Liu et al., 2023a), which achieves $37.5\%$, RetroInText shows a $3.7\%$ improvement. These results suggest that the synergy between structural features and text-aware components substantially enhances predictive accuracy. Additionally, the removal of the textual component, as indicated by the RetroInText (w/o text) configuration, results in a Top-1 accuracy of $40.2\%$. Compared to the complete RetroInText model, which achieves $41.2\%$, highlighting the value of the textual module in providing essential contextual information that supports more accurate predictions. Further details on the analyses and experimental setup can be found in Appendix C, which provides additional insights into the significance of each module.

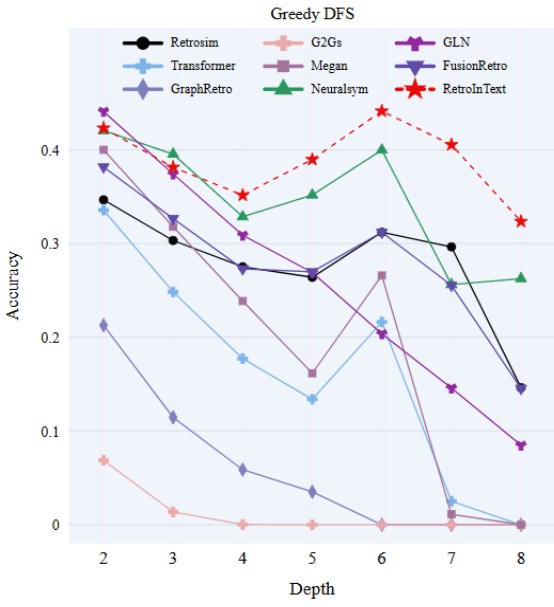

Figure 2: Test accuracy of retrosynthesis models combined with Greedy DFS at different depths. The red star stands for our method RetroInText.

Additionally, we conduct experiments across different depths, which demonstrate that incorporating textual information consistently improves performance at all levels. As shown in Table. 4 Retro* outperforms Retro*(w/o text), particularly at increasing depths, showing robustness in predicting long synthetic routes. The most pronounced gains in Top-1 to Top-5 accuracy occur at deeper paths

(Depths 5 to 8), highlighting the effectiveness of textual data in enhancing prediction accuracy for complex retrosynthetic planning tasks.

| Methods | Top-1 | Top-2 | Top-3 | Top-4 | Top-5 |
|---|---|---|---|---|---|
| MolT5 (SMILES) | 37.2 | 43.7 | 46.2 | 47.4 | 48.3 |
| RetroInText(1D SMILES) | 35.6 | 41.6 | 44.1 | 45.4 | 46.2 |
| RetroInText(2D+3D Graph) | 37.5 | 45.0 | 48.2 | 50.0 | 50.9 |
| RetroInText(w/o text) | 40.2 | 47.3 | 50.2 | 51.7 | 52.7 |
| **RetroInText** | **41.2** | **48.7** | **51.8** | **53.3** | **54.2** |

Table 3: Ablation study of RetroInText for exact match accuracy (%).

| Depth | Retro*(w/o text) | | | | | Retro*(with text) | | | | |
|---|---|---|---|---|---|---|---|---|---|---|
| | Top-1 | Top-2 | Top-3 | Top-4 | Top-5 | Top-1 | Top-2 | Top-3 | Top-4 | Top-5 |
| Depth2 | **45.0** | **52.4** | 55.4 | 57.2 | 58.3 | 44.9 | 52.3 | **55.4** | **57.3** | **58.3** |
| Depth3 | 38.9 | 45.9 | 49.3 | 50.5 | 51.5 | **40.0** | **47.9** | **51.5** | **53.0** | **53.9** |
| Depth4 | 33.7 | 40.9 | 42.5 | 43.6 | 43.6 | **36.1** | **43.6** | **46.4** | **47.7** | **48.3** |
| Depth5 | 35.5 | 41.7 | 43.4 | 44.4 | 44.4 | **39.0** | **47.8** | **50.3** | **51.2** | **51.7** |
| Depth6 | 33.0 | 36.3 | 36.9 | 38.0 | 38.0 | **36.3** | **40.8** | **41.9** | **43.0** | **44.1** |
| Depth7 | 25.7 | 31.1 | 31.1 | 31.1 | 31.1 | **28.4** | **33.8** | **35.1** | **35.1** | **35.1** |
| Depth8 | 29.4 | 41.2 | 41.2 | 41.2 | 41.2 | **32.4** | **41.2** | **44.1** | **47.1** | **47.1** |

Table 4: Exact match accuracy (%) at different depths of ground truth synthetic routes.

# 6 CONCLUSIONS

In this work, we propose RetroInText, a novel framework for retrosynthetic planning that leverages contextual information along the synthetic route through ChatGPT. RetroInText employs in-context learning to incorporate textual information from previous steps, enhancing realistic retrosynthetic planning. Additionally, we use a fine-tuned language model, MolT5 (Edwards et al., 2022), along with a pre-trained molecular representation model to integrate both molecular structure and 3D conformational data, improving the selection process. Experiments on the RetroBench dataset demonstrate that RetroInText outperforms existing template-free methods, achieving SOTA performance. Further experiments at various depths and ablation studies show the strength of text information during retrosynthetic planning. In the future, we are planning to develop an end-to-end question-answering model (Maziarz et al., 2022; Liu et al., 2023d) to further improve retrosynthetic step selection and enhance the utility of a deep learning-based retrosynthesis model.

ACKNOWLEDGMENTS

This study was supported by grants from the National Natural Science Foundation of China (NSFC 62322215). This study was supported in part by the high-performance computing center of Central South University.

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

# A    EXPERIMENTAL PROCEDURE AND SETUP

## A.1    EVALUATION METRIC

The current search algorithms (Segler et al., 2018; Chen et al., 2020; Kim et al., 2021; Yu et al., 2022; Obonyo et al., 2023; Yuan et al., 2024; Xie et al., 2024) predominantly rely on search success rate as the primary evaluation metric, without assessing whether the identified intermediates are capable of synthesizing the target molecules. As illustrated in Figure. 3, by integrating existing one-step models, which achieve top-k accuracies in the range of 60% to 80%, with the Retro* algorithm, we observe that the search success rates for the multi-step retrosynthesis process increase to between 85% and 94% (Liu et al., 2023a). This outcome appears counterintuitive, as one might expect a decrease in success rate with the addition of each synthesis step. Consequently, we adopt the new evaluation metric proposed by FusionRetro, which considers the set of precise matches between the predicted materials and the baseline reference. A prediction is deemed correct when the set of actual materials obtained from the model matches at least one of the feasible synthesis routes in the target molecule test set. Furthermore, a search for paper cuttings is conducted. The search is terminated when the length of the predicted synthesis path exceeds the depth of the true synthesis path. We use the starting materials derived from the reaction network in RetroBench as the baseline and compare them with the starting materials identified through the proposed search process.

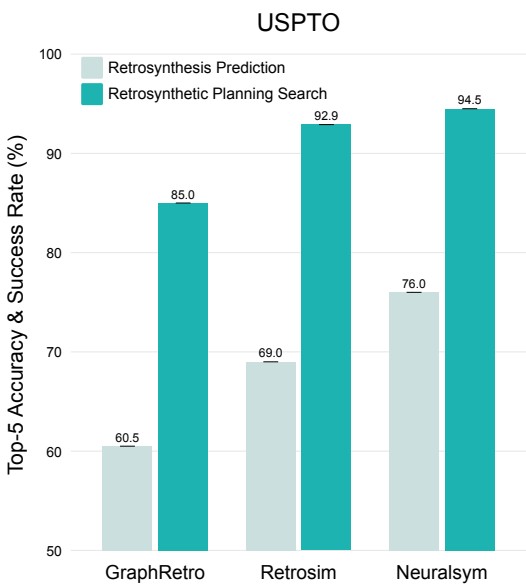

Figure 3: Performance of different retrosynthesis models for retrosynthesis prediction and multi-step planning on USPTO dataset. This result has been reproduced from FusionRetro (Liu et al., 2023a).

## A.2    IMPLEMENTATION DETAILS

We use Pytorch (Paszke et al., 2019) to implement our models. The codes of all baselines are implemented referring to the implementations of FusionRetro (Liu et al., 2023a) and CREBM (Liu et al., 2024b). All the experiments of baselines are conducted on a single NVIDIA 4090 with 24GB memory size. The softwares that we use for experiments are Python 3.8.19, CUDA 11.5.119, einops 0.7.0, pytorch 2.2.0, pytorch-scatter 2.1.2, pytorch-sparse 0.6.18, numpy 1.24.4, torchvision 0.17.0. The total inference time is 79.5 hours.

### A.3 FINE-TUNE PROCESS

All reaction in the training set is extracted as the fine-tuned dataset which includes more than $150K$ reactions. We train it for $40$ epochs and choose the best checkpoint as a single-step model. The detailed training parameters are shown in Table. 5.

| Parameter | Value | Description |
|---|---|---|
| Learning rate | $5 \cdot 10^{-5}$ | Step size for optimization |
| Batch size | 8 | Number of samples per batch |
| Epochs | 40 | Number of training iterations |
| Hidden layers | 24 | Number of layers in the model |
| Hidden units | 768 | Number of neurons per layer |
| Head number | 12 | Number of multi-head per layer |
| Save steps | 5000 | Save checkpoint step |
| Beam number | 4 | Beam search number |
| Weight decay | 0.1 | Regularization coefficient |

Table 5: Fine-tune parameters.

## B MOLECULE NAME GENERATION

Before using ChatGPT to generate the text information for the routes, we should get the IUPAC names as mentioned in Section 4.1.2. Specifically, we extract all the intermediates with the matched depth in the training set for all products then generate products and corresponding intermediate IUPAC names by PubChemPy. For the test set, we generate the IUPAC names of the products, but we only need them during creation.

## C MORE RESULTS

### C.1 SINGLE-STEP MODEL RESULTS

All the reactions in the test dataset are extracted for evaluating single-step models, with an overall $24,972$ reactions. The results are shown in Table. 6

| Models | Top-k accuracy (%) | | | |
|---|---|---|---|---|
| | Top-1 | Top-3 | Top-5 | Top-10 |
| FusionRetro | 31.1 | 39.4 | **42.3** | **47.0** |
| Transformer | 28.1 | 38.7 | 41.8 | 46.0 |
| MolT5-small | 20.8 | 30.0 | 33.9 | 38.3 |
| MolT5-base(Ours) | **33.3** | **39.9** | 42.1 | 44.5 |

Table 6: Retrosynthesis prediction results for exact match accuracy (%).

### C.2 PROMPT IMPACT ON PATHWAY DESCRIPTIONS

To test the impact of text quality on the model, specifically, two distinct prompts were evaluated, as outlined in Table. 7, to assess their impact on the textual descriptions of the molecule O=C1N=C(O)CN1N=C(O)C=Cc1ccc(Cl)c(C(F)(F)F)c1. The first prompt is designed to generate detailed, yet excessively verbose descriptions, while the second prompt focuses on producing concise and targeted textual outputs.

Prompt1 is characterized by a lack of specificity, resulting in verbose descriptions with considerable redundancy. These excessive details, which include irrelevant information, have a detrimental effect on the model's accuracy. In contrast, Prompt2 is deliberately designed to prioritize clarity and relevance, avoiding superfluous content. The generated text is both concise and well-structured,

contributing to an enhancement in downstream model performance. As illustrated in Table. 8, the use of text generated from the higher-quality prompt leads to a notable improvement in the performance of the framework. Specifically, the Top-1 accuracy increases from 39.2% to 41.2%, while the Top-5 accuracy improves from 46.9% to 54.2%. These results underscore the significance of carefully crafting prompts to generate concise, high-quality text in order to achieve better model predictions.

| Target Molecule | Prompt | Text |
|---|---|---|
| O=C1N=C(O)CN 1N=C(O)C=Cc1ccc (Cl)c(C(F)(F)F)c1 | Please describe the structural features, functional aspects, and applicable implementation methods of the molecule product, and describe the potential intermediate molecules intermediates in the synthesis pathway, with each intermediate molecule separated by "[SEP]". Additionally, provide a different reaction pathway and describe it. | The molecule 3-[4-chloro-3-(trifluoromethyl)phenyl]-N-(2,4-dioxoimidazolidin-1-yl)prop-2-enamide is a complex organic compound that features a substituted phenyl group, an imidazolidin-1-yl moiety, and a prop-2-enamide functional group. Structural features: 1. Substituted phenyl group with a chlorine atom and a trifluoromethyl group. 2. Imidazolidin-1-yl ring with two carbonyl groups. 3. Prop-2-enamide functional group. Functional aspects: 1. The substituted phenyl group provides aromaticity and potentially affects the molecule's reactivity and solubility. 2. The imidazolidin-1-yl ring may participate in hydrogen bonding interactions or act as a moiety for stabilization. 3. The prop-2-enamide functional group can participate in conjugation reactions or undergo nucleophilic addition reactions. Applicable implementation methods: 1. Synthesis of the molecule may involve various organic chemistry techniques such as Grignard reactions, amide bond formation, and cyclization reactions. 2. Purification techniques such as column chromatography, recrystallization, or distillation may be used to obtain the final product in high purity. Potential intermediate molecules: 1. 3-[4-chloro-3-(trifluoromethyl)phenyl]prop-2-enoic acid. Different reaction pathway: Instead of directly synthesizing the target molecule, an alternative pathway might involve the initial formation of 3-[4-chloro-3-(trifluoromethyl)phenyl]prop-2-enoic acid as an intermediate. Subsequently, the prop-2-enoic acid can be converted to the desired prop-2-enamide through an amide coupling reaction with the imidazolidin-1-yl moiety. Overall, the synthesis of such complex molecules requires careful planning, knowledge of organic chemistry reactions, and proper characterization techniques to confirm the structure of intermediate and final products. |
| O=C1N=C(O)CN 1N=C(O)C=Cc1ccc (Cl)c(C(F)(F)F)c1 | Describe the key transition states involved in the synthesis of products from the intermediates intermediates. Explain the structural changes and energy barriers for each transition state, and reply to me in a sentence. | The key transition states involved in the synthesis include the formation of an imine intermediate through the condensation of the prop-2-enoic acid and the amine group, followed by the cyclization of the imine to form the imidazolidin-1-yl ring. |

Table 7: Comparison of Prompts and Textual Descriptions.

| Model | Top-1 | Top-2 | Top-3 | Top-4 | Top-5 |
|---|---|---|---|---|---|
| RetroInText (Prompt1) | 39.2 | 43.4 | 45.6 | 45.7 | 46.9 |
| RetroInText (Prompt2) | 41.2 | 48.7 | 51.2 | 53.3 | 54.2 |

Table 8: Comparison of prompt influence on textual quality for molecular descriptions and reactions

| Target Molecule | Text (ChatGPT-3.5) | Text (ChatGPT-4) | Similarity |
|---|---|---|---|
| CC(C)Nc1ncnc2cscc12 | There are two key transition states involved in the synthesis: the conversion of 2-chloro-N-propan-2-ylthieno[3,4-d]pyrimidin-4-amine to N-propan-2-ylthieno[3,4-d]pyrimidin-4-amine, with the structural change being the displacement of the chlorine atom by the amine group, and the conversion of N-propan-2-ylthieno[3,4-d]pyrimidin-4-amine to N-propan-2-ylthieno[3,4-d]pyrimidin-4-amine, with the structural change being the formation of the final amine functionality; both transitions have energy barriers that must be overcome for the reactions to proceed. | The synthesis of N-propan-2-ylthieno[3,4-d]pyrimidin-4-amine involves key transition states that typically include nucleophilic substitution, where the chlorine atom in 2-chloro-N-propan-2-ylthieno[3,4-d]pyrimidin-4-amine is replaced by an amine group, with the transition state requiring the alignment of the nucleophile and electrophile, overcoming the energy barrier attributed to breaking the C-Cl bond and forming the new C-N bond, usually characterized by a partially formed bond and a high-energy, tetrahedral intermediate. | These texts by different models describe the same key transition states, including the nucleophilic substitution where the chlorine atom is replaced by an amine group and a tautomeric shift. Both mention energy barriers involved in the transition states. |
| CN=C(O)NCCNCc1cc[nH]n1 | The key transition states involved in the synthesis of 1-methyl-3-[2-(1H-pyrazol-5-ylmethylamino)ethyl]urea from the intermediates N'-(1H-pyrazol-5-ylmethyl)ethane-1,2-diamine include the formation of a urea linkage and the addition of a methyl group. These transitions involve structural rearrangements and energy barriers that facilitate the conversion of the intermediates into the desired product. | In the synthesis of 1-methyl-3-[2-(1H-pyrazol-5-ylmethylamino)ethyl]urea from N'-(1H-pyrazol-5-ylmethyl)ethane-1,2-diamine, the key transition states include the nucleophilic attack of the amine on the isocyanate carbon to form a tetrahedral zwitterionic intermediate, followed by proton transfer to establish the urethane linkage with an energy barrier influenced by sterics and electronic effects, leading to the final urea product. | The mechanisms and transformations described in both are similar, involving the formation of the urea bond and methylation. |

Table 9: Comparison of textual descriptions generated by ChatGPT-3.5 and ChatGPT-4.

## C.3 COMPARISON OF OUTPUTS BETWEEN GPT-3.5 AND GPT-4

We employ GPT-4 for text generation, and a comparison between the outputs generated by GPT-4 and GPT-3.5 reveals a high degree of similarity in both content and the structural transformations described, as illustrated by the examples provided in Table. 9. Specifically, both models effectively

characterize key transition states for the target molecules, including nucleophilic substitutions, tautomeric shifts, and the associated energy barriers for bond-breaking and bond-forming processes. However, GPT-4 presents certain practical challenges, particularly with frequent API key limitations, which disrupt the workflow and diminish its reliability for consistent use. In contrast, GPT-3.5 exhibits stable performance without such restrictions, making it a more dependable choice for our framework. Given the negligible differences in performance and the operational constraints of GPT-4, GPT-3.5 is selected as the primary text generator for the experiments.

## C.4 ABLATION STUDY

To test the role of each part in the framework, we perform the ablation study on RetroInText. First, we use the no fine-tuning MolT5 model as the single-step model and observe that the generated SMILES strings for the corresponding molecules were invalid, resulting in scores of 0 across all cases. This indicates that the original MolT5 model is not suitable for our task, and fine-tuning is necessary.

We also experimented only using the combination of SMILES and text to train the model, however, this combination is inferior to those of a multimodal approach. Next, we use the fine-tuned MolT5 as the single-step model, without incorporating the molecular representation model or textual information, and only rely on molecule fingerprints for scoring. Finally, we introduce textual information into the training process and use 3DInfomax as the molecular representation model, while excluding textual information in testing. The results demonstrate a significant improvement in multi-step accuracy when textual context information is included, indicating that using textual information in multi-step processes is highly effective. As shown in Figure. 4, the case of depth3 shows text can make accurate predictions compared to not using text.

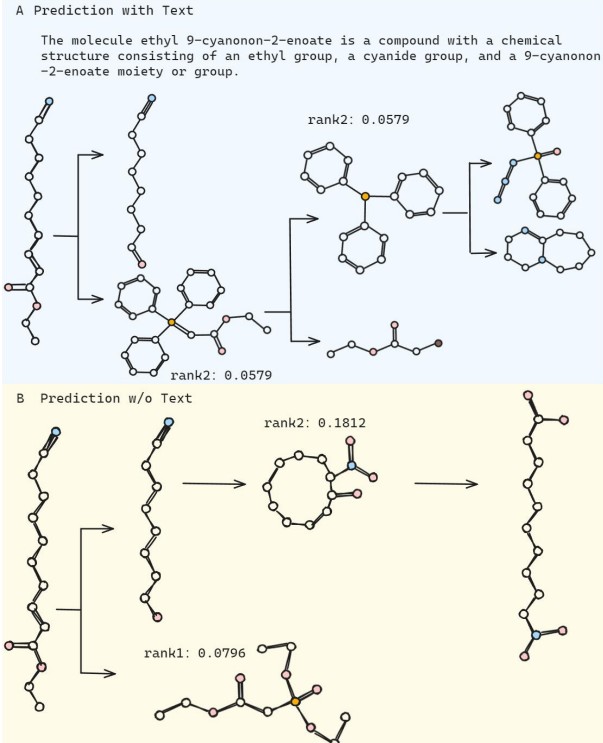

Figure 4: Comparison of retrosynthesis prediction with text (**A**) and without text (**B**).

