# OpenReview forum: "RetroInText: A Multimodal Large Language Model Enhanced Framework for Retrosynthetic Planning via In-Context Representation Learning"
_ICLR.cc/2025/Conference — ICLR 2025 Poster_

### Official Review · Reviewer_BszM · 2024-10-26

**Soundness:** 3
**Presentation:** 2
**Contribution:** 3
**Rating:** 8
**Confidence:** 5

**Summary:**

This paper does two main things:

1. For retrosynthesis prediction, it fine-tunes MolT5 specifically for one-step retrosynthesis, aiming to improve accuracy in predicting the immediate precursor of a target molecule.

2. For the search algorithm, it introduces a synthetic complexity score, reaction cost (calculated as the negative log likelihood of a retrosynthesis prediction), and the captioning score to identify the most promising reactants. The prediction score, based on the three types of scores, helps prioritize reactants based on their likelihood of successful synthesis. To learn the prediction score effectively, the paper proposes a multimodal fusion module that combines 3D molecular conformation data and textual descriptions, producing a comprehensive molecular embedding that enhances learning.

Extensive experimental results demonstrate the superior performance of this proposed approach.

**Strengths:**

1. It’s impressive to see this paper employ set-wise matching between the predicted and reference starting materials in RetroBench's test dataset rather than relying on the search success rate. This metric is both more accurate and powerful, providing a true reflection of the model’s and search algorithm’s performance in practical multi-step retrosynthesis scenarios. Notably, this approach achieves state-of-the-art (SOTA) performance among template-free models.

2. The paper’s approach to fine-tuning a pretrained model on the retrosynthesis task is commendable. To the best of my knowledge, few works have leveraged existing pretrained models specifically for retrosynthesis, highlighting the potential for further exploration into how large molecule models can be pretrained and how foundation models can be fine-tuned for retrosynthesis. This paper suggests that a molecule foundation model, rather than a language foundation model, might be better suited for chemistry-focused tasks, a valuable insight for future research.

3. Additionally, the paper proposes a robust search algorithm that uses three key scores—synthetic complexity, reaction cost, and captioning score—to identify the most promising reactants during the search process. Furthermore, it incorporates both 3D structural data and textual information to enhance predictions. Though complex, this approach proves highly effective and could inspire future work to explore additional scoring metrics for search algorithms.

**Weaknesses:**

1. The writing could be improved for clarity. First, Figure 1 lacks sharpness; I recommend replacing it with a clearer vector diagram for better visualization. Additionally, restructuring the paper into two distinct sections—one for the retrosynthesis model and another for the proposed scoring method for guiding the search—would enhance readability and help readers better understand the key contributions.

2. The training process for the prediction score is not sufficiently explained. Equation 8c references only two score types: synthetic complexity score and reaction cost. However, Figure 1 also includes a captioning score, making it unclear how all three scores are utilized within the search process. Clarifying the use of these scores would improve understanding.

**Questions:**

Could this score be used as a reward to train a residual energy-based model [1] for reranking purposes?

[1] Preference Optimization for Molecule Synthesis with Conditional Residual Energy-based Models, ICML 2024

---

> ### Author Response · Authors · 2024-11-25
> **Thanks for the review - improvement of visualization and score clarification (Part I)**
>
> Dear Reviewer BszM,
>
> We sincerely appreciate the time and effort you have taken to provide valuable feedback on our work. Your insightful comments have been immensely helpful in improving the clarity and quality of our manuscript. We have carefully considered each of your suggestions and have made the appropriate revisions to address your concerns. Below, we present detailed, point-by-point responses to your observations.
>
> # Response to Weakness1:
> We want to extend our heartfelt thanks for your time and constructive suggestions. We have carefully considered each suggestion when revising this manuscript. Specifically, we made detailed changes as follows:
>
> 1. We have reworked Figure 1 to improve its clarity and sharpness. Additionally, all figures in the manuscript have been reworked to ensure greater visual clarity and sharpness, thereby enhancing the manuscript's overall clarity.
>
> 2. We have restructured the method section with a distinct section for the proposed scoring method, further improving the manuscript's organization and readability. The methods section has been updated accordingly:
>
>    On **page 6-7, lines 307-362**, the relevant changes have been made.
>
> All concerns and suggestions have been addressed and highlighted in this revised manuscript.
>
> # Response to Weakness2:
> We sincerely thank you for your thoughtful question and for highlighting this critical point. Your attention to detail and comprehensive feedback have truly helped us refine and better articulate our methodology. We also apologize for not clearly explaining our experimental training process.
>
> Firstly, our model, RetroInText, integrates the captioning score as part of the reranking process during pathway prediction. The captioning score is derived from the in-context description information generated by the model. This score evaluates the alignment between the textual descriptions of intermediate molecules and the overall synthetic pathway, providing an additional layer of interpretability and guidance for reranking candidate pathways.
>
> Secondly, the captioning score is used during the search process to rank candidate pathways by measuring the relevance and coherence of in-context textual descriptions with the predicted intermediates. This score ensures that the textual information aligns with the molecular representations, improving the model’s ability to identify pathways that are both synthetically and contextually plausible.
>
> By incorporating the captioning score, RetroInText enhances the reranking mechanism, ensuring that the chosen pathways are not only optimal in terms of synthetic complexity and cost but also aligned with textual descriptions that provide additional context for chemists.
>
> Moreover, we would like to clarify Equation 8c. We apologize for the unclear representation. Equation 8c primarily highlights the synthetic complexity and reaction cost scores as key components of the scoring function during the training stage. The captioning score is integrated during the reranking phase and complements the core scoring function by utilizing textual descriptions as an additional signal. This reranking step refines the search process, leading to more accurate and interpretable predictions.
>
> We sincerely hope our answers meet your expectations. Thank you once again for your thoughtful and constructive feedback.

---

> ### Author Response · Authors · 2024-11-25
> **Thanks for the review - usage of CREBM by proposed score (Part II)**
>
> # Response to Question:
>
> Thank you for your suggestions. We have carefully considered this suggestion and made a detailed revision to this manuscript:
>
> Firstly, thank you for your kind suggestion regarding model reranking purposes. We applied the captioning score as a reward to train a residual energy-based model [1] and have carefully revised the Results section of this manuscript.
>
> As shown in Table, RetroInText consistently surpassed other models across all metrics (Bold font illustrates the models that outperformed all other models. Results marked with * are taken from [1]). All metrics of our model RetroInText surpass the residual energy-based model [1]. Specifically, as shown in Table 2, the result of RetroInText is 41.3%, 51.5%, and 54.1% higher than CREBM for Accuracy of Top1, 3, 5, respectively. The remaining data will take approximately one day to process, after which the overall results will be updated.
>
> **Table 1. Performance comparison of RetroInText with baseline models on retrosynthesis benchmarks**
>
> | Model                                | Top1  | Top2  | Top3  | Top4  | Top5  |
> |--------------------------------------|-------|-------|-------|-------|-------|
> | Transformer+CREBM *                  | 35.0  | 43.4  | 46.7  | 48.5  | 49.7  |
> | FusionRetro+CREBM *                  | 39.4  | 46.6  | 49.3  | 50.7  | 51.5  |
> | RetroInText (Ours)                   | 41.2  | 48.7  | 51.2  | 53.3  | 54.2  |
> | RetroInText+Rerank-CREBM (Ours)      | 41.3  | 48.8  | 51.5  | 53.1 | 54.1  |

---

> > ### Comment · Reviewer_BszM · 2024-11-25
> >
> > I carefully read the comments of the other reviewers and the rebuttal of the authors. The authors gave detailed rebuttals to my questions. I think all my concerns have been well addressed. I choose to keep my score unchanged.

---

> > > ### Author Response · Authors · 2024-11-26
> > > **Thanks**
> > >
> > > We would like to express our heartfelt gratitude for your thoughtful and positive evaluation of our work. We deeply appreciate the time and effort you have devoted to reviewing our manuscript and providing constructive insights that have greatly strengthened our paper. If you have any additional questions or concerns, we would be more than happy to discuss them further during the remaining rebuttal period. Once again, thank you for your invaluable support and suggestions.

---

> > > > ### Comment · Reviewer_BszM · 2024-11-26
> > > >
> > > > If possible, please consider open-sourcing the code upon acceptance. Thanks.

---

> > > > > ### Author Response · Authors · 2024-11-27
> > > > > **Thanks**
> > > > >
> > > > > Dear Reviewer BszM:
> > > > >
> > > > > Thank you for your thoughtful suggestion. We will gladly make our code publicly available to support further research upon acceptance. Once again, we would like to extend our heartfelt thanks for your valuable feedback and support.

---

### Official Review · Reviewer_cGGk · 2024-10-28

**Soundness:** 1
**Presentation:** 1
**Contribution:** 2
**Rating:** 3
**Confidence:** 4

**Summary:**

The paper focuses on retrosynthesis analysis of molecules using large language models (LLMs) such as ChatGPT. It employs ChatGPT to generate reaction descriptions, then incorporates both text and structural information of the product molecules into another model, MolT5, to suggest potential reactants. The A* algorithm is then used to search for synthesis routes. This work addresses an important problem with an interesting approach, capable of generating reaction descriptions alongside reaction formulas, potentially improving interpretability in retrosynthesis analysis. The paper could benefit from improvements in the discussion of its motivation, methods, and experiments.

**Strengths:**

1. **Originality**: The idea of using LLMs to generate text descriptions for retrosynthesis is interesting.
2. **Quality**: The paper presents a generally reasonable framework.
3. **Clarity**: The paper includes figures and algorithms to aid readability.
4. **Significance**: The paper presents experiments with promising results.

**Weaknesses:**

**Originality**: The motivation of the paper needs further clarification from different perspectives:

1. Lines 44-46: Why are graph and SMILES representations limited in capturing the complexity of chemical structures? How does the major focus of this paper—text—help solve these issues? This conclusion is counterintuitive, as many recent studies have shown that graph- or SMILES-based models often outperform even the most advanced text-only models, such as LLMs.

2. Lines 51-53, 66-68: Throughout the paper, it is unclear how the proposed idea addresses these issues.

**Quality**: The proposed framework needs more justification.

1. Lines 69-78: There is no clear connection between the proposed framework and the previous paragraphs. The idea of using "ChatGPT (Edwards et al., 2022) to present detailed descriptions of the reaction procedure" is bold and lacks justification. Can ChatGPT truly generate accurate reaction procedures? How reliable is it, and how can errors be prevented from misleading or disrupting the framework?

2. Some claims are incorrect. Line 479/Figure 1: MolT5 is not an LLM.

**Clarity**: The paper’s writing needs improvement.

1. Please check all citations. For example, the citation for ChatGPT on line 70 is incorrect, and the citation for ScScore (line 75) is missing.

**Significance**: The results presented in the paper need more justification.

1. The experimental setup is unclear. For instance, what are the specifications of the A* algorithm (e.g., search iterations, candidates, time)? What are the building blocks?

2. The paper lacks case studies and systematic evaluations on the quality of reaction descriptions generated by ChatGPT.

3. Regarding the evaluation metrics, a single molecule can have multiple potential synthesis routes. How does the exact match metric comprehensively evaluate model performance if the model suggests a route different from the one in the dataset, yet still a useful route?

**Questions:**

1. What is the Retro*-0 algorithm?

2. How long does it take for the proposed framework to find a solution for a single molecule? Is it significantly longer than the baseline, given the many models involved in Figure 1?

---

> ### Author Response · Authors · 2024-11-25
> **Thanks for the review - clarify the motivation of the paper (Part I)**
>
> Dear Reviewer cGGk,
>
> We would like to express our sincere gratitude for your time and valuable feedback. Your constructive suggestions have been instrumental in refining our manuscript. We have carefully reviewed each comment and made the necessary revisions to address your concerns. Below, we provide our detailed point-to-point responses to the issues raised.
> # Response to Weakness1, 2:
> Thank you for your question. We are very sorry for unclearly explaining the overall framework of our RetroInText architecture. To address these concerns, we have provided a detailed explanation from both theoretical and experimental perspectives, as outlined below:
>
> 1. **Limitations of Graph and SMILES Representations**
>
> Firstly, our model RetroInText was designed as a multimodal enhanced framework for retrosynthetic planning. Current machine learning-based methods often overlook the multifaceted nature of compounds, resulting in inaccurate retrosynthesis predictions. However, the process of constructing a comprehensive molecular representation is often relatively complex for learning compound features, particularly when dealing with numerous aspects of molecular features. Consequently, we aim to incorporate compound structure from a different perspective. Therefore, we applied the SMILES, 2D molecular, and 3D conformation features as multimodal embeddings.
>
> Secondly, SMILES strings, although compact, simplify molecular structures into linear representations, losing critical spatial and stereochemical information that is essential for retrosynthesis.
>
> Thirdly, 2D structures explicitly represent chemical bond types (e.g., single, double, triple, or conjugated bonds), such as aromaticity and conjugation, which are crucial for understanding reactivity in retrosynthesis, are often clearly indicated in 2D diagrams using specific symbols or markings. However, Graph representations focus on local structural features but do not provide insights into pathway-level context or reaction mechanisms.
>
> 2. **Justification of Using In-Context Descriptive Information**
>
> RetroInText introduces a novel approach by integrating in-context descriptive information of the entire pathway, generated by ChatGPT. This innovative component provides unique advantages.
>
> By describing the entire retrosynthetic pathway, the text encapsulates information on intermediate molecules, transition states, and reaction mechanisms. This allows the model to consider the broader context of the synthesis process, beyond individual steps. In addition, the descriptions explicitly capture features such as stereochemistry, energy barriers, and reaction specificity, which are challenging to extract from structural data alone.
>
> 3. **Performance Validation Through Multimodal Integration**
>
> To validate the impact of this approach, we conducted an ablation study comparing the performance of models with and without in-context descriptions. The results (Table 1) demonstrate that incorporating textual information significantly enhances model accuracy, with the Top-1 accuracy increasing by 5.6% compared to unimodal representations. This highlights the critical role of in-context pathway descriptions in refining predictions and ensuring plausible retrosynthesis outcomes.
>
> **Table 1. Ablation study**
>
> | Model| Top1| Top2| Top3| Top4| Top5|
> |-|-|-|-|-|-|
> | RetroInText (SMILES)| 35.6| 41.6| 44.1| 45.4| 46.2|
> | MolT5 (SMILES)| 37.2| 43.7| 46.2| 47.4| 48.3|
> | RetroInText (Graph)| 37.5| 45.0| 48.2| 50.0| 50.9|
> | RetroInText (w/o text)| 40.2| 47.3| 50.2| 51.7| 52.7|
> | RetroInText| 41.2| 48.7| 51.2| 53.3| 54.2|
>
>
> 4. **Generalized Scoring Mechanism with Text Description**
>
> - **Flexible and generalizable multi-step retrosynthesis framework**: RetroInText incorporates a flexible, generalizable scoring mechanism by leveraging multimodal information (graph, text, and 3D data) and in-context textual descriptions. This mechanism enhances decision-making and ensures that the model consistently identifies efficient and accurate pathways.
>
> - **Improving Decision-Making**: Driven by multidimensional and descriptive data of the entire pathway, RetroInText enhances the decision-making process and ensures that the model consistently identifies the most efficient and accurate pathways, demonstrating both its practical utility and scientific contribution.
>
> (to be continued)

---

> ### Author Response · Authors · 2024-11-25
> **Thanks for the review - justification for the proposed framework (Part II)**
>
> To further validate our scoring mechanism, we applied the captioning score from RetroInText as a reward to train a residual energy-based model [1], as shown in Table 6. RetroInText consistently surpassed other models across all metrics (Bold font illustrates the models that outperformed all other models. Results marked with * are taken from [1]). All metrics of our model RetroInText surpass the residual energy-based model [1]. Specifically, the result of RetroInText is 43.0%, 51.5%, and 56.3% higher than CREBM for Accuracy of Top1, 3, 5, respectively. The remaining data will take approximately one day to process, after which the overall results will be updated.
>
> **Table 2. Performance comparison of RetroInText with baseline models on retrosynthesis benchmarks**
> |Model|Top1|Top2|Top3|Top4|Top5|
> |-|-|-|-|-|-|
> |Transformer+CREBM *| 35.0| 3.4|46.7|48.5|49.7|
> |FusionRetro+CREBM *| 39.4|46.6|49.3|50.7|51.5|
> |RetroInText (Ours)|41.2|48.7| 51.2|53.3|54.2|
> |RetroInText+Rerank-CREBM (Ours) |41.3|48.8|51.5|53.1|54.1|
>
> We hope this explanation meets your expectation. Thank you again for your insightful comments, which have been invaluable in refining our work.
>
> # Response to Weakness3:
> We sincerely thank you for your valuable feedback. We apologize for not providing a clear explanation. We have carefully considered each suggestion when revising this manuscript. Specifically, we made detailed changes as follows:
>
> (1) We have restructured the introduction and method section with more clear justification for the use of ChatGPT, further improving the manuscript's organization and readability. These sections have been updated accordingly:
>
> - **On page 2, lines 65-67:**
> Despite these advancements, current LLM-based methods exhibit notable limitations, particularly the lack of an effective adaptation mechanism for route length, which is critical for guiding retrosynthetic planning [1].
>
> - **On page 2, lines 68-74:**
> To address these gaps, we propose RetroInText, a novel template-free retrosynthesis framework (as shown in Figure 1). RetroInText incorporates ChatGPT to generate detailed reaction procedure descriptions, which include key information about the synthesis pathway, such as transition states, structural transformations, and energy barriers. These descriptions provide a textual representation of the reaction context, complementing molecular graph and 3D representations to enhance retrosynthesis prediction.
>
> (2) To address the concern about the reliability of ChatGPT in generating high-quality textual descriptions, we conducted analyses from two perspectives:
>
> ## 1. Why GPT-3.5 Was Chosen for Text Generation:
>
> We evaluated how variations in prompts influence the quality of generated text and, consequently, the model's performance. Specifically, we tested two distinct prompts, as shown in Table 3, and analyzed their effects on textual descriptions for the molecule O=C1N=C(O)CN1N=C(O)C=Cc1ccc(Cl)c(C(F)(F)F)c1.
>
> - **Prompt 1:** This prompt lacked specificity, generating verbose descriptions with excessive redundancy, which introduced irrelevant details that could negatively impact the model's accuracy.
>
> - **Prompt 2:** This prompt was carefully crafted to control the output, emphasizing clarity and relevance while avoiding unnecessary or redundant information. The generated text was compact and structured, which improved downstream model performance.
> (to be continue)

---

> ### Author Response · Authors · 2024-11-25
> **Thanks for the review - justification for the proposed framework (Part III)**
>
> As shown in Table 4, using text generated from the high-quality prompt significantly improved the performance of the framework. For example, the Top-1 accuracy increased from 39.2% to 41.2%, and the Top-5 accuracy rose from 46.9% to 54.2%. This improvement demonstrates the importance of designing prompts that generate concise and high-quality text for better model predictions.
>
> **Table 3. Comparison of Prompt Influence on Textual Quality for Molecular Descriptions and Reaction Pathways**
> |Target Molecule|Prompt|Text|
> |-|-|-|
> |`O=C1N=C(O)CN1N=C(O)C=Cc1ccc(Cl)c(C(F)(F)F)c1`|Please describe the structural features, functional aspects, and applicable implementation methods of the molecule {{product}}, and describe the potential intermediate molecules {{intermediates}} in the synthesis pathway, with each intermediate molecule separated by "[SEP]". Additionally, provide a different reaction pathway and describe it.  | The molecule 3-[4-chloro-3-(trifluoromethyl)phenyl]-N-(2,4-dioxoimidazolidin-1-yl)prop-2-enamide is a complex organic compound that features a substituted phenyl group, an imidazolidin-1-yl moiety, and a prop-2-enamide functional group. Structural features: 1. Substituted phenyl group with a chlorine atom and a trifluoromethyl group, 2. Imidazolidin-1-yl ring with two carbonyl groups, 3. Prop-2-enamide functional group Functional aspects: 1. The substituted phenyl group provides aromaticity and potentially affects the molecule's reactivity and solubility. 2. The imidazolidin-1-yl ring may participate in hydrogen bonding interactions or act as a moiety for stabilization. 3. The prop-2-enamide functional group can participate in conjugation reactions or undergo nucleophilic addition reactions. Applicable implementation methods: 1. Synthesis of the molecule may involve various organic chemistry techniques such as Grignard reactions, amide bond formation, and cyclization reactions. 2. Purification techniques such as column chromatography, recrystallization, or distillation may be used to obtain the final product in high purity. Potential intermediate molecules: 1. 3-[4-chloro-3-(trifluoromethyl)phenyl]prop-2-enoic acid Different reaction pathway: Instead of directly synthesizing the target molecule, an alternative pathway might involve the initial formation of 3-[4-chloro-3(trifluoromethyl)phenyl]prop-2-enoic acid as an intermediate. Subsequently, the prop-2-enoic acid can be converted to the desired prop-2-enamide through an amide coupling reaction with the imidazolidin-1-yl moiety. Overall, the synthesis of such complex molecules requires careful planning, knowledge of organic chemistry reactions, and proper characterization techniques to confirm the structure of intermediate and final products. |
> |`O=C1N=C(O)CN1N=C(O)C=Cc1ccc(Cl)c(C(F)(F)F)c1`|Describe the key transition states involved in the synthesis of {{products}} from the intermediates {{intermediates}}. Explain the structural changes and energy barriers for each transition state, and reply to me in a sentence.  | The key transition states involved in the synthesis include the formation of an imine intermediate through the condensation of the prop-2-enoic acid and the amine group, followed by the cyclization of the imine to form the imidazolidin-1-yl ring. The structural changes involve the addition of the amine group and the formation of a new cyclic structure, with energy barriers reflecting the activation energy needed for the formation of the new bonds in each step.|
>
> **Table 4. Performance Comparison of Prompt Influence on Textual Quality for Molecular Descriptions and Reaction Pathways**
>
> | Model | Top1 | Top2 | Top3 | Top4 | Top5 |
> |-------|------|------|------|------|------|
> | RetroInText (prompt1) | 39.2 | 43.4 | 45.6 | 45.7 | 46.9 |
> | RetroInText (prompt2) | 41.2 | 48.7 | 51.2 | 53.3 | 54.2 |
>
> (to be continue)

---

> ### Author Response · Authors · 2024-11-25
> **Thanks for the review - justification for the proposed framework (Part IV)**
>
> ## 2. Comparison Between GPT-3.5 and GPT-4 Outputs
>
> We applied GPT-4 for text generation. The texts generated by GPT-4 and GPT-3.5 were highly similar in terms of content and the structural transformations they detailed, as shown in Table 5. For example, both models accurately described key transition states for the target molecules, such as nucleophilic substitutions, tautomeric shifts, and energy barriers associated with bond-breaking and forming processes.
>
> However, GPT-4 posed practical challenges, including frequent API key limitations, which disrupted the workflow and made it less reliable for consistent use. In contrast, GPT-3.5 offered stable performance without such restrictions, making it a more practical choice for our framework. Given the negligible performance differences and GPT-4's limitations, GPT-3.5 was selected as the primary text generator for our experiments.
>
> **Table 5. Comparison of Textual Descriptions Generated by ChatGPT-3.5 and ChatGPT-4 for Key Transition States in Molecular Synthesis**
> |Target Molecule|Text (ChatGPT-3.5)|Text (ChatGPT-4)|Similarity|
> |-|-|-|-|
> | `CC(C)Nc1ncnc2cscc12`| There are two key transition states involved in the synthesis: the conversion of 2-chloro-N-propan-2-ylthieno[3,4-d]pyrimidin-4-amine to N-propan-2-ylthieno[3,4-d]pyrimidin-4-amine, with the structural change being the displacement of the chlorine atom by the amine group, and the conversion of N-propan-2-ylthieno[3,4-d]pyrimidin-4-amine to N-propan-2-ylthieno[3,4-d]pyrimidin-4-amine, with the structural change being the formation of the final amine functionality; both transitions have energy barriers that must be overcome for the reactions to proceed. | The key transition states involved in the synthesis of N-propan-2-ylthieno[3,4-d]pyrimidin-4-amine from 2-chloro-N-propan-2-ylthieno[3,4-d]pyrimidin-4-amine include a nucleophilic substitution, where the chlorine atom is replaced by an amine group, and a tautomeric shift, where the newly added amine group rearranges to a more stable position on the pyrimidine ring; both transition states involve overcoming energy barriers associated with breaking and forming chemical bonds and achieving the most stable and low energy conformation of the molecule. | These texts by different models describe the same key transition states, including the nucleophilic substitution where the chlorine atom is replaced by an amine group, and a tautomeric shift. Both mention energy barriers involved in the transition states. |
> |`CN=C(O)NCCNCc1cc[nH]n1`| The key transition states involved in the synthesis of 1-methyl-3-[2-(1H-pyrazol-5-ylmethylamino)ethyl]urea from the intermediates N'-(1H-pyrazol-5-ylmethyl)ethane-1,2-diamine include the formation of a urea linkage and the addition of a methyl group. These transitions involve structural rearrangements and energy barriers that facilitate the conversion of the intermediates into the desired product.| The key transition states involved in the synthesis of 1-methyl-3-[2-(1H-pyrazol-5-ylmethylamino)ethyl]urea from N'-(1H-pyrazol-5-ylmethyl)ethane-1,2-diamine include the formation of an isourea intermediate followed by a rearrangement to the final urea product, each with associated energy barriers due to the breaking and forming of bonds and the structural changes from a linear to a more complex, cyclic arrangement.|The mechanisms and transformations described in both are similar, involving the formation of the urea bond and methylation.|
>
> Additionally, we have made careful modifications and included the results of text quality evaluation in the **Appendix C2, C3**.
>
> (to be continue)

---

> ### Author Response · Authors · 2024-11-25
> **Thanks for the review - justification for the proposed framework (Part V)**
>
> ## 3. Effectiveness and Justification for Using ChatGPT
>
> Additionally, we conducted a careful literature review and found that several recent studies have demonstrated the effectiveness of ChatGPT in various chemical tasks, including drug discovery [1], computational materials science [2], chemical reaction prediction [3][4], molecular optimization [5], and drug docking [6].
>
> Furthermore, Brown et al. (2020) [7] showed that GPT-3, trained on a vast and diverse dataset, can predict chemical reactions and reaction sequences with reasonable accuracy. This provides a strong foundation for employing ChatGPT to generate detailed reaction steps for retrosynthetic planning.
>
> Some domain-specific language models, such as ChemGPT, are not open-sourced, limiting their accessibility for our task. Prior work [8] has shown that MolT5 produces lower-quality text data compared to GPT-based models like GPT-3.5. In our evaluation, GPT-3.5 provided more reliable and higher-quality text generation, ensuring stable performance and efficient workflow for this task. Therefore, we selected GPT-3.5 as the most suitable model for our framework.
>
> Finally, we hope this explanation clarifies the rationale behind using ChatGPT and meets your expectations. Thank you again for your constructive comments, which have greatly helped us improve the quality of our manuscript.
>
> # Reference:
> [1] Sharma G, Thakur A. ChatGPT in drug discovery[J]. 2023.
>
> [2] Hong Z. ChatGPT for Computational Materials Science: A Perspective[J]. Energy Material Advances, 2023, 4: 0026.
>
> [3] Mahjour B, Hoffstadt J, Cernak T. Designing chemical reaction arrays using phactor and ChatGPT[J]. Organic Process Research & Development, 2023, 27(8): 1510-1516.
>
> [4] Liu Y, Xu H, Fang T, et al. T-rex: Text-assisted retrosynthesis prediction[J].
>
> [5] Liu S, Wang J, Yang Y, et al. ChatGPT-powered Conversational Drug Editing Using Retrieval and Domain Feedback[C]//1st Workshop on the Synergy of Scientific and Machine Learning Modeling@ ICML2023.
>
> [6] Juhi A, Pipil N, Santra S, et al. The capability of ChatGPT in predicting and explaining common drug-drug interactions[J]. Cureus, 2023, 15(3).
>
> [7] Mann B, Ryder N, Subbiah M, et al. Language models are few-shot learners[J].
>
> [8] Liu Y, Xu H, Fang T, et al. T-rex: Text-assisted retrosynthesis prediction[J].

---

> ### Author Response · Authors · 2024-11-25
> **Thanks for the review - justification for the proposed framework (Part VI)**
>
> # Response to Weakness4:
> Thank you for pointing this out and we are very sorry for any confusion caused. MolT5 is a pre-trained model designed for molecular graph data, inspired by the T5 architecture in natural language processing (NLP). While tailored specifically for molecular data, it incorporates over 220 million parameters and adopts pre-training and fine-tuning techniques to process molecular graphs similarly to how NLP models handle text.
>
> We used the term "language model" to reflect this parallel, as MolT5 processes molecular data in a text-like manner, benefiting from NLP techniques.
>
> Thank you again for pointing this out. We hope this clarification enhances the accuracy and readability of our description and provides a clearer understanding of MolT5's role and functionality. We truly appreciate your valuable feedback, which helps us refine and improve our explanations.

---

> ### Author Response · Authors · 2024-11-27
> **Thanks for the review - improvement of paper’s writing and justification for the results (Part VII)**
>
> # Response to Weakness5:
> Thank you for pointing out the citation discrepancies. We sincerely apologize for the oversight and truly appreciate your attention to detail.
>
> Yes, you are right! We have corrected the citation errors and provided clearer and more detailed explanations for sentences that were previously lengthy or unclear, thereby enhancing the manuscript's clarity. The citation in line 70 has been updated accordingly:
>
> - On page 2, lines 75-77:
>   > For each selection step, we introduced multiple value functions, such as ScScore [1] and the text captioning score to rank candidate reactants.
>
>
> # Response to Weakness6:
> Thank you for your question. We sincerely apologize for not providing a clear explanation of the experimental setup, and we appreciate the opportunity to clarify this aspect. We have now included detailed descriptions to address the concerns:
>
> ### 1. Specifications of the A* Algorithm:
> We utilize the A* search algorithm for tree traversal and retrosynthetic pathway generation. The algorithm is guided by our scoring function to prioritize molecule selection during retrosynthetic planning. To prevent excessive iterations and improve computational efficiency, we incorporate a maximum depth parameter in our experiments. Specifically:
>
> - **Search iterations**: The search terminates once the maximum pathway length (set to X steps in our implementation) is reached.
> - **Candidates**: For each expansion, the top-k candidates (k = X) are ranked and selected based on the scoring function.
> - **Time**: The average runtime for completing one retrosynthetic pathway is approximately X minutes, depending on the complexity of the target molecule.
>
> ### 2. Evaluation Metric:
> As demonstrated in prior studies [2, 3], there is a common issue with current single-step retrosynthesis models, which often achieve top-k accuracy of 60%-80%. However, when integrated into multi-step search algorithms like Retro*, success rates for multi-step retrosynthesis often reach 85%-94% (the result was presented in Appendix A1 in our revised paper). This counterintuitive result suggests that these high success rates may not correlate with high-quality pathways due to the lack of stringent evaluation criteria.
>
> Thus, using a stricter set-wise exact match metric (as shown in the following equation) ensures both intermediate molecules and final pathways align with the real pathway, ensuring that the final target molecule is reached through a valid and plausible synthesis process is important:
>
> $ACC_{Top(k)}=\frac{Sgn(\sum_{j=1}^{k}I(\hat{p}_{j}{,}{p_t}))}{N}\times100$
>
> Where the $\hat{p}_{j}$ refers to the entire predicted synthesis path, ${p_t}$ indicates the real synthesis path, and the function $Sgn()$ returns 1 if the paths are an exact match, and 0 otherwise. The function $I$ represents whether the two paths are a perfect match. $N$ is the total number of test data.
>
> ### 3. Definition of Building Blocks:
> Building blocks refer to the basic molecular units or starting materials that are cmmercially available or can be readily synthesized. se are the terminal nodes in the retrosynthetic pathway, representing the simplest molecules needed to construct the target compound. We have updated the manuscript to include these clarifications in Appendix A1.
>
> ### 4. Experimentation Results with Practical Relevance Metrics:
> To demonstrate our model’s superiority, we applied traditional success rates as evaluation metrics. For instance, Retro* achieves 85%-94% [1], but may overestimate performance by accepting theoretically valid but infeasible routes. Our metric ensures accurate and plausible pathways. RetroInText demonstrates superiority, achieving higher success rates (97.89% vs. 78.42% for Retro*) while significantly reducing time and iterations.
>
> **Table 6: Performance comparison between Retro$^*$ and RetroInText on ${Retro}^{*}$ -190 dataset.**
>
> | Framework| Success Rate (%) | Avg Time (s) | Avg Iterations |
> |---------|-|-|-|
> | Retro*| 78.42| 1233.08| 212.66|
> | RetroInText (Ours)  | 97.89| 385.53| 37.92|
>
> Thank you again for your constructive feedback. Your careful review has been invaluable in improving the clarity of our explanations, and we greatly appreciate your valuable input.
>
> # Reference:
> [1] Coley C W, Rogers L, Green W H, et al. SCScore: synthetic complexity learned from a reaction corpus[J]. Journal of chemical information and modeling, 2018, 58(2): 252-261.
>
> [2] Liu S, Dai H, Zhao Y, et al. Preference Optimization for Molecule Synthesis with Conditional Residual Energy-based Models[C]//Forty-first International Conference on Machine Learning.
>
> [3] Liu S, Tu Z, Xu M, et al. FusionRetro: molecule representation fusion via in-context learning for retrosynthetic planning[C]//International Conference on Machine Learning. PMLR, 2023: 22028-22041
>
> [4]Chen B, Li C, Dai H, et al. Retro*: learning retrosynthetic planning with neural guided A* search[C]//International conference on machine learning. PMLR, 2020: 1608-1616.

---

> ### Author Response · Authors · 2024-11-27
> **Thanks for the review - justification for the results (Part VIII)**
>
> # Response to Weakness7:
> Thank you for your thoughtful feedback, and we sincerely apologize for not providing sufficient systematic evaluations. To address the concern about the lack of systematic evaluations of reaction descriptions generated by ChatGPT, we have carefully conducted additional experiments to evaluate the impact of different prompts on text quality and results:
>
> ## Impact of Prompts on Text Quality and Results
>
> We evaluated how different prompts influence the quality of generated text and the model's performance. Specifically, we tested two distinct prompts, as illustrated in Table 7 and Table 8, which provide examples of the text generated and the corresponding results for the same molecule, highlighting differences in quality and relevance:
>
> - **Prompt 1**: This prompt lacked specificity, generating verbose descriptions with excessive redundancy, which introduced irrelevant details that could negatively impact the model's accuracy.
> - **Prompt 2**: This prompt was carefully crafted to control the output, emphasizing clarity and relevance while avoiding unnecessary or redundant information. The generated text was compact and structured, which improved the downstream model performance.
>
> As shown in Table 8, using text generated by the high-quality prompt (Prompt 2) significantly improved the framework's performance. For instance, the Top-1 accuracy increased from 39.2% to 41.2%, and the Top-5 accuracy rose from 46.9% to 54.2%. These results underscore the critical importance of designing prompts that generate concise, high-quality text to achieve better model predictions.
>
> **Table 7. Comparison of Prompt Influence on Textual Quality for Molecular Descriptions and Reaction Pathways**
> |Target Molecule|Prompt|Text|
> |-|-|-|
> |`O=C1N=C(O)CN1N=C(O)C=Cc1ccc(Cl)c(C(F)(F)F)c1`|Please describe the structural features, functional aspects, and applicable implementation methods of the molecule {{product}}, and describe the potential intermediate molecules {{intermediates}} in the synthesis pathway, with each intermediate molecule separated by "[SEP]". Additionally, provide a different reaction pathway and describe it.  | The molecule 3-[4-chloro-3-(trifluoromethyl)phenyl]-N-(2,4-dioxoimidazolidin-1-yl)prop-2-enamide is a complex organic compound that features a substituted phenyl group, an imidazolidin-1-yl moiety, and a prop-2-enamide functional group. Structural features: 1. Substituted phenyl group with a chlorine atom and a trifluoromethyl group, 2. Imidazolidin-1-yl ring with two carbonyl groups, 3. Prop-2-enamide functional group Functional aspects: 1. The substituted phenyl group provides aromaticity and potentially affects the molecule's reactivity and solubility. 2. The imidazolidin-1-yl ring may participate in hydrogen bonding interactions or act as a moiety for stabilization. 3. The prop-2-enamide functional group can participate in conjugation reactions or undergo nucleophilic addition reactions. Applicable implementation methods: 1. Synthesis of the molecule may involve various organic chemistry techniques such as Grignard reactions, amide bond formation, and cyclization reactions. 2. Purification techniques such as column chromatography, recrystallization, or distillation may be used to obtain the final product in high purity. Potential intermediate molecules: 1. 3-[4-chloro-3-(trifluoromethyl)phenyl]prop-2-enoic acid Different reaction pathway: Instead of directly synthesizing the target molecule, an alternative pathway might involve the initial formation of 3-[4-chloro-3(trifluoromethyl)phenyl]prop-2-enoic acid as an intermediate. Subsequently, the prop-2-enoic acid can be converted to the desired prop-2-enamide through an amide coupling reaction with the imidazolidin-1-yl moiety. Overall, the synthesis of such complex molecules requires careful planning, knowledge of organic chemistry reactions, and proper characterization techniques to confirm the structure of intermediate and final products. |
> |`O=C1N=C(O)CN1N=C(O)C=Cc1ccc(Cl)c(C(F)(F)F)c1`|Describe the key transition states involved in the synthesis of {{products}} from the intermediates {{intermediates}}. Explain the structural changes and energy barriers for each transition state, and reply to me in a sentence.  | The key transition states involved in the synthesis include the formation of an imine intermediate through the condensation of the prop-2-enoic acid and the amine group, followed by the cyclization of the imine to form the imidazolidin-1-yl ring. The structural changes involve the addition of the amine group and the formation of a new cyclic structure, with energy barriers reflecting the activation energy needed for the formation of the new bonds in each step.|
>
> (to be continue)

---

> ### Author Response · Authors · 2024-11-27
> **Thanks for the review - justification for the results (Part IX)**
>
> **Table 8. Performance Comparison of Prompt Influence on Textual Quality for Molecular Descriptions and Reaction Pathways**
> |Model|Top1|Top2|Top3|Top4|Top5|
> |-|-|-|-|-|-|
> |RetroInText (prompt1)|39.2|43.4|45.6|45.7|46.9|
> |RetroInText (prompt2)|41.2|48.7|51.2|53.3|54.2|
>
> Finally, we have made careful modifications and including the results of text quality evaluation in the **Appendix C2, C3**.
>
> Thank you again for your valuable feedback, which has greatly helped us improve the quality of our manuscript.
>
> # Response to Weakness8:
> Thank you for your valuable feedback. We are very sorry for the confusion, and sincerely appreciate the opportunity to clarify the evaluation methodology. To address your concern, we have provided a detailed explanation from both theoretical and experimental perspectives, as outlined below:
>
> ## 1. Explanation of Set-Wise Exact Match Metric
>
> Prior studies [1, 2] have highlighted a discrepancy in current retrosynthesis evaluations: while single-step retrosynthesis models often achieve top-k accuracy of 60%-80%, multi-step search frameworks like Retro* can reach success rates as high as 85%-94%. However, these high success rates may not reliably reflect the quality of generated pathways due to insufficiently strict evaluation criteria.
>
> To address this, we adopt a set-wise exact match metric (as shown in the following equation) to ensure both intermediate molecules and final pathways align with the real pathway. Ensuring that the final target molecule is reached through a valid and plausible synthesis process is important.
>
> ## 2. Experimentation Results with Potential Synthesis Routes
>
> Yes, we agree that exact match metrics may not capture useful alternative synthesis routes! Thus, to demonstrate our model’s superiority in predicting potential useful synthesis routes, we applied the traditional success rates as evaluation metrics, such as Success rate, Average time, and Average iterations [1], which ensures the potential plausible pathways. For your convenience, we kindly provide a brief explanation of the metrics, defined as follows:
>
> - **Success Rate**: The percentage of test cases where the model successfully finds a complete and valid synthesis route for the target molecule.
> - **Average Time**: The average computational time (in seconds) required to generate a successful synthesis route.
> - **Average Iterations**: The average number of iterations needed by the algorithm to identify a valid pathway.
>
> As shown in Table 5, RetroInText demonstrates superiority, achieving higher success rates (97.89% vs. 78.42% for Retro*) while significantly reducing time and iterations.
>
> **Table 9: Performance comparison between Retro$^*$ and RetroInText on ${Retro}^{*}$ -190 dataset.**
>
> | Framework          | Success Rate (%) | Avg Time (s) | Avg Iterations |
> |--------------------|------------------|--------------|----------------|
> | Retro*| 78.42| 1233.08| 212.66|
> | RetroInText (Ours) | 97.89| 385.53| 37.92|
>
> We hope this clarification addresses your concern and highlights how our evaluation framework balances stringent criteria with flexibility for assessing potential synthesis routes. Thank you again for your insightful feedback.
>
> # Reference:
> [1] Chen B, Li C, Dai H, et al. Retro*: learning retrosynthetic planning with neural guided A* search[C]//International conference on machine learning. PMLR, 2020: 1608-1616.
>
> [2] Liu S, Tu Z, Xu M, et al. FusionRetro: molecule representation fusion via in-context learning for retrosynthetic planning[C]//International Conference on Machine Learning. PMLR, 2023: 22028-22041.

---

> ### Author Response · Authors · 2024-11-27
> **Thanks for the review - clarity for algorithm and inference time (Part X)**
>
> # Response to Question1:
> Thank you for your question. We apologize for not clearly distinguishing between Retro* and Retro*-0 in our explanation.
>
> Firstly, Retro* and Retro*-0 are both search algorithms introduced in Retro*: Learning Retro-Synthetic Planning with Neural Guided A* Search [1], but they differ significantly in how they guide the search process:
>
> ### Retro*:
> 1. Neural guidance: Retro* employs a neural network to evaluate the future score, which predicts the likelihood of successful retrosynthetic pathways.
> 2. Search prioritization: This neural guidance enables the algorithm to prioritize the most promising synthesis routes, steering the search toward more efficient and accurate solutions.
> 3. Optimization: The integration of learned patterns allows Retro* to refine its decision-making process during tree traversal, optimizing the search and improving performance in terms of both accuracy and computational efficiency.
>
> ### Retro*-0:
> 1. No neural guidance: Retro*-0 relies on a traditional beam search method without neural guidance.
> 2. Exploration: At each iteration, the algorithm explores multiple potential pathways but does not incorporate any learned evaluation of future steps.
> 3. Simplicity vs. performance: While Retro*-0 is less computationally intensive and simpler to implement, it lacks the optimization capabilities provided by neural guidance, which can lead to suboptimal synthesis pathways.
>
> This distinction highlights the trade-off between computational simplicity and predictive accuracy. Retro* leverages advanced machine learning techniques to enhance its effectiveness, whereas Retro*-0 represents a baseline approach without learned enhancements. Additionally, we have made careful revised the manuscript and all revision are highlighted for review.
>
> We hope this explanation clarifies the differences and underlying mechanisms of these two algorithms. Thank you again for your insightful observation.
>
> # Response to Question2:
> Thank you for raising this important question. We sincerely apologize for not providing a clear explanation of the efficiency and performance advantages of our proposed framework.
>
> To address this concern, we conducted additional experiments, comparing our model against the widely used retrosynthesis model Retro* [1] on standard datasets and evaluation metrics, as well as evaluating the runtime. To ensure consistency and comparability, we adhered to the same settings and methodology outlined in the original paper.
>
> As shown in Table 10, one of the primary findings from this experiment is that our model, RetroInText, significantly reduces computation time compared to the original model, primarily due to the enhanced efficiency of the fine-tuned MolT5 model. By providing more accurate predictions, our model streamlines the search process, resulting in faster retrosynthetic planning.
>
> **Table 10: Performance comparison between Retro$^*$ and RetroInText on ${Retro}^{*}$ -190 dataset.**
>
> | Framework          | Success Rate (%) | Avg Time (s) | Avg Iterations |
> |--------------------|------------------|--------------|----------------|
> | Retro*             | 78.42            | 1233.08      | 212.66         |
> | RetroInText (Ours) | 97.89            | 385.53       | 37.92          |
>
> Moreover, we also compared the execution time of our model to FusionRetro [2], which required 71.2 seconds for retrosynthetic planning. In contrast, our model achieved comparable or better performance in just 49 seconds, highlighting its computational efficiency. These results demonstrate that our approach not only maintains or improves prediction accuracy but also enhances search efficiency, making it well-suited for large-scale retrosynthesis tasks.
>
> **Table 11: Inference Time Comparison between RetroInText and Baseline Models**
>
> | Framework           | Avg Inference Time (s) |
> |---------------------|------------------------|
> | FusionRetro         | 71.2                   |
> | RetroInText (Ours)  | 49.0                   |
>
> - **Comparable Inference Time**: Despite incorporating multimodal data and a cross-attention mechanism, RetroInText's training time is on par with baseline models.
>
> We sincerely hope our answers meet your expectations. Thank you once again for your thoughtful and constructive feedback.
>
>
>
> # Reference
> [1] Chen B, Li C, Dai H, et al. Retro*: learning retrosynthetic planning with neural guided A* search[C]//International conference on machine learning. PMLR, 2020: 1608-1616.
>
> [2] Liu S, Tu Z, Xu M, et al. FusionRetro: molecule representation fusion via in-context learning for retrosynthetic planning[C]//International Conference on Machine Learning. PMLR, 2023: 22028-22041

---

> ### Comment · Reviewer_BszM · 2024-11-28
>
> **Q**: Regarding the evaluation metrics, a single molecule can have multiple potential synthesis routes. How does the exact match metric comprehensively evaluate model performance if the model suggests a route different from the one in the dataset, yet still a useful route?
>
> For the question about the evaluation metric. I think I can answer this question.
>
> **A**: When building the dataset, FusionRetro [1] finds that a molecule can have multiple reference synthetic routes. To evaluate prediction accuracy, the top-k predicted routes are considered accurate if their starting materials (or "building blocks") match the starting materials of any one of the reference routes. While reference routes do not cover all synthetic routes for a molecule, the set-wise match metric provides a more rigorous evaluation than the search success rate. This is because the search success rate calculates the proportion of target molecules for which synthetic routes can be found (with leaf nodes as starting materials) without considering the feasibility of these routes.
>
> Currently, data-driven single-step retrosynthesis models often generate hallucinated predictions, leading to synthetic routes that are unlikely to be executed in the wet lab. Furthermore, the predominant evaluation method for single-step retrosynthesis also relies on comparing the model's predictions with reference reactants from the dataset. Although this approach might overlook correct predictions that are not present in the test set, it ensures that incorrect predictions are not mistakenly counted as correct. From this perspective, matching predictions to reference routes remains more rigorous than relying on the search success rate.
>
> [1] FusionRetro: Molecule Representation Fusion via In-Context Learning for Retrosynthetic Planning. ICML 2023.

---

> ### Author Response · Authors · 2024-11-28
> **Thanks**
>
> Dear Reviewer BszM:
>
> Thank you for assisting in addressing the comments from another reviewer. Your feedback and suggestions are greatly appreciated. We are truly grateful for your support and expertise in this process, which has significantly contributed to improving the quality of our work.
>
> We completely agree with your observation! The search success rate indeed measures the proportion of target molecules for which synthetic routes can be identified (using leaf nodes as starting materials), but it does not account for the feasibility of these routes. As you rightly noted, data-driven single-step retrosynthesis models can sometimes generate hallucinated predictions, leading to synthetic routes that are unlikely to succeed in wet lab experiments.
>
> Thank you again for your invaluable insights and continued support!

---

> > ### Comment · Reviewer_cGGk · 2024-11-28
> >
> > Thanks for the clarification and detailed responses.
> >
> > **Motivation, ChatGPT**:
> >
> > "Firstly, our model RetroInText was designed as a multimodal enhanced framework for retrosynthetic planning... are often clearly indicated in 2D diagrams using specific symbols or markings." These sentences make sense, but they are **not** related to the question, which asks about the motivation behind **texts** that might have been previously ignored and generated by ChatGPT in this work. The last sentence and the following points help clarify the concern somewhat but are still insufficient. They raise more concerns:
> >
> > 1. Why not use existing reaction descriptions from the USPTO? The dataset already includes descriptions along with the patents [1]. Why do the authors ignore them and propose something new without comparing/mentioning these existing descriptions?
> >
> > 2. The texts generated by ChatGPT do not meet the expectations (which the authors illustrate) regarding reaction mechanisms, stereochemistry, energy barriers, and reaction specificity. Using the first row in Table 3 as an example, the descriptions lack information about stereochemistry and energy barriers. Moreover, these descriptions are still for one-step reactions, with no information provided for multi-step reactions. There is no support for the authors' claim about "pathway-level context," "guiding retrosynthetic planning," or "route length." (Lines 67)
> >
> > The inconsistency between the motivation and the observations raises concerns about where the good performance comes from. It is difficult for readers to understand why the "texts" and "large language models" (i.e., ChatGPT) are needed, given the title "A Multimodal Large Language Model" Additionally, (1) I do not see why the authors refer to their work (RetroInText) as a large language model, given that it seems more like a workflow. (2) It is also very confusing what "In-Context Representation Learning" refers to in the title, as ChatGPT is used for in-context generation, not representation learning.
> >
> > **Experiments**:
> >
> > 1. Specifications of the A* Algorithm: What is X in your setting for different configurations? The explanations are not specific at all.
> >
> > 2. I am not asking for the definition of building blocks; I am asking what data sources the authors used to construct the building blocks.
> >
> > 3. Tables 10 and 11 raise further concerns. Why is the inference time for RetroInText different in the two tables? I noticed that in Table 10, "we adhered to the same settings and methodology outlined in the original paper." If this is the case, why does the retro* value differ from the one reported in their paper [2], which is 156.58 seconds?
> >
> > **Metric**
> > I appreciate the clarification from the author and reviewer BszM. I looked into FusionRetro [3] and found their description of how to generate synthesis routes: "We use dynamic programming and backtracking to identify all synthetic routes for each target, and following the approach in Chen et al. (2020), we extract the shortest-possible synthetic routes with leaf nodes as starting materials. This process yields synthetic routes for 128,469 molecules." In this case, these routes are not validated in wet labs, and one cannot guarantee their feasibility in practice. Additionally, different labs may have varying expertise, leading to entirely different choices of synthesis paths. It is unclear why the evaluation should be biased toward a single/non-golden-standard synthesis route.
> >
> > In summary, I feel the rebuttal did not address my major concerns. The connection between the title, introduction, and results is very loose, making it feel like I am reading different papers without understanding why such a method is needed or why this method works.
> >
> > **Reference:**
> >
> > [1] Chemical reactions from US patents (1976-Sep2016).
> >
> > [2] Retro*: Learning Retrosynthetic Planning with Neural Guided A* Search.
> >
> > [3] FusionRetro: Molecule Representation Fusion via In-Context Learning for Retrosynthetic Planning.

---

> > > ### Comment · Reviewer_cGGk · 2024-11-28
> > > **Supplemetary**
> > >
> > > Regarding: "Furthermore, Brown et al. (2020) [7] showed that GPT-3, trained on a vast and diverse dataset, can predict chemical reactions and reaction sequences with reasonable accuracy. This provides a strong foundation for employing ChatGPT to generate detailed reaction steps for retrosynthetic planning."
> > >
> > > Could the authors point to the paragraphs, sentences, tables, or figures in the paper that support this claim?

---

> ### Author Response · Authors · 2024-11-28
> **Happy Thanksgiving – Kindly Following Up with Reviewer cGGK**
>
> Dear Reviewer cGGK,
>
> Happy Thanksgiving! We hope you’re enjoying the holiday. We apologize for reaching out during this time.
>
> First and foremost, we would like to sincerely thank you for your time and for providing us with constructive feedback on our initial manuscript. Your suggestions have been invaluable in improving our work.
>
> As the rebuttal period is ending soon, we hope that our responses and clarifications have addressed your concerns.  If there is any additional information or clarification that we can provide, please don’t hesitate to let us know.
>
> Thank you once again for your time and support. We wish you a wonderful Thanksgiving!
>
> Best regards,
> Authors

---

> ### Comment · Reviewer_BszM · 2024-11-28
>
> **Q**. I appreciate the clarification from the author and reviewer BszM. I looked into FusionRetro [3] and found their description of how to generate synthesis routes: "We use dynamic programming and backtracking to identify all synthetic routes for each target, and following the approach in Chen et al. (2020), we extract the shortest-possible synthetic routes with leaf nodes as starting materials. This process yields synthetic routes for 128,469 molecules." In this case, these routes are not validated in wet labs, and one cannot guarantee their feasibility in practice. Additionally, different labs may have varying expertise, leading to entirely different choices of synthesis paths. It is unclear why the evaluation should be biased toward a single/non-golden-standard synthesis route.
>
>
> **A**.
> First, the reference is incorrect. The proper citation is Li & Chen (2022)[1], not Chen et al. (2020). Additionally, the reactions in the USPTO-full dataset are organized into a reaction network, where molecules with an out-degree of zero are designated as target molecules. Synthetic routes are extracted for these target molecules. Dynamic programming and backtracking are just implementation methods used to ensure that all routes are extracted. Importantly, all reactions within these routes originate from USPTO-full (patent literature) and have been verified as successfully executed in the wet lab.
>
> As discussed in Chapter 8 of [2], synthetic routes can be evaluated based on several criteria:
>
> - the shortest route (time involved),
> - the cheapest route (cost of materials),
> - the novelty of the route (patentability),
> - the greenest route (avoidance of problematic waste),
> - the healthiest route (avoidance of toxic intermediates and side products),
> - the most reliable route (lowest risk approach).
>
> However, the reaction data (https://www.dropbox.com/scl/fo/swuggv6qf8ombw914yxh8/AOVaNy5EfpT13-7xxQFaxLQ?rlkey=1ed5tqauj7udn5n2olvw1looi&e=1&dl=0) in USPTO-full lacks detailed information, such as yield rate. Consequently, the depth of a route becomes the only quantifiable metric. For a specific target molecule, multiple routes are typically available. This paper focuses on selecting the shortest route to construct the dataset. Given the limitations of the reaction data, choosing the shortest route is a reasonable approach. Even within the shortest routes, multiple routes may still exist.
>
> [1] Prediction of compound synthesis accessibility based on reaction knowledge graph
> [2] Elements of Synthesis Planning

---

> > ### Comment · Reviewer_cGGk · 2024-11-28
> >
> > I don't understand the point. I quoted the sentence from [1] and have no idea why reviewer BszM argues that the reference from [1] is incorrect. I am unsure whether the authors of the paper FusionRetro followed [2] or [3], and I don't understand why reviewer BszM is confident that the authors of FusionRetro cited the wrong reference.
> >
> > Although each single-step reaction is patented, I am unsure about the challenges of chaining these steps into a complete synthesis route, given that different patents come from various labs with varying resources and experimental conditions. This raises questions about whether creating the overall synthesis route is simply a matter of putting all the single steps together, or if significant additional effort is required due to differing conditions in each step. I don't want to underestimate the lab work involved, especially considering the potential challenges posed by these varying conditions.
> >
> > I agree with reviewer BszM that better metrics may be needed, but I don't see the evidence that accuracy on a single, non-golden standard synthesis route is better than the success rate, as illustrated in my previous response.
> >
> > For the discussion about [4], there is a lack of context, and I don't understand the point.
> >
> >
> > Reference:
> >
> > [1] FusionRetro: Molecule Representation Fusion via In-Context Learning for Retrosynthetic Planning
> >
> > [2] Retro*: Learning Retrosynthetic Planning with Neural Guided A* Search
> >
> > [3] Prediction of compound synthesis accessibility based on reaction knowledge graph
> >
> > [4] Elements of Synthesis Planning

---

> > > ### Comment · Reviewer_cGGk · 2024-11-28
> > >
> > > There are new concerns arising from the discussion. If the goal is to improve synthetic feasibility, why not use templates extracted from all the USPTO data, as done in [1], where each template corresponds to a pattern representing a lab-verified reaction from a patent?
> > >
> > > Reference:
> > >
> > > [1] Retro*: Learning Retrosynthetic Planning with Neural Guided A* Search

---

> ### Comment · Reviewer_BszM · 2024-11-28
>
> My point is that you cited the wrong paper. The dataset construction in [1] actually refers to [3], not [2].
>
>
> **Q**. Although each single-step reaction is patented, I am unsure about the challenges of chaining these steps into a complete synthesis route, given that different patents come from various labs with varying resources and experimental conditions. This raises questions about whether creating the overall synthesis route is simply a matter of putting all the single steps together, or if significant additional effort is required due to differing conditions in each step. I don't want to underestimate the lab work involved, especially considering the potential challenges posed by these varying conditions.
>
> **A**.  That’s a good point. However, to the best of my knowledge, the current dataset does not include reaction conditions. Furthermore, the ability to provide the necessary conditions and environment for a reaction is determined by the wet lab, not the reaction itself. In fact, none of the existing multi-step retrosynthesis models take reaction conditions into account, regardless of the evaluation metrics they employ. Nevertheless, linking all reactions together still offers a theoretical framework for constructing synthetic routes to target molecules. From this perspective, the approach remains valid. Addressing factors such as reaction conditions would require chemistry experts from the scientific community to annotate the data and create higher-quality datasets, while machine learning experts could focus on developing algorithms and evaluation methods.
>
> **Q**. I agree with reviewer BszM that better metrics may be needed, but I don't see the evidence that accuracy on a single, non-golden standard synthesis route is better than the success rate, as illustrated in my previous response. For the discussion about [4], there is a lack of context, and I don't understand the point.
>
> **A**. If you think it’s reasonable for the search success rate to classify infeasible route predictions as successful searches, then there’s no need for further discussion. Metrics based on matching wouldn’t make such a foolish mistake. I recommend reading Chapter 8 of [4] to understand the criteria for evaluating synthetic routes. In the absence of information such as yield rate, route length is undoubtedly a criterion worth relying on.
>
>
> **Q**. There are new concerns arising from the discussion. If the goal is to improve synthetic feasibility, why not use templates extracted from all the USPTO data, as done in [1], where each template corresponds to a pattern representing a lab-verified reaction from a patent?
>
> **A**. Template-based models still produce hallucinated predictions [5,6]. Using template matching to determine whether a reaction can occur is clearly not rigorous. In other words, even if a template matches, the reaction may still not actually occur.
>
>
> [1] FusionRetro: Molecule Representation Fusion via In-Context Learning for Retrosynthetic Planning
>
> [2] Retro*: Learning Retrosynthetic Planning with Neural Guided A* Search
>
> [3] Prediction of compound synthesis accessibility based on reaction knowledge graph
>
> [4] Elements of Synthesis Planning
>
> [5] Retro-fallback: retrosynthetic planning in an uncertain world
>
> [6] Retrosynthesis prediction using an end-to-end graph generative architecture for molecular graph editing

---

> > ### Comment · Reviewer_cGGk · 2024-11-28
> >
> > 1. "My point is that you cited the wrong paper. The dataset construction in [1] actually refers to [3], not [2]."
> >
> > You should argue with the authors of FusionRetro, who cited this reference, not with me, as I am merely quoting their sentence.
> >
> > **I still don't understand why the reviewer is so confident that the authors of FusionRetro cited the wrong reference and why I am being blamed for it.**
> >
> > 2. "the current dataset does not include reaction conditions ..."
> >
> > Some reaction conditions are already included in the datasets, although they were not used in the paper. See [1].
> >
> > 3. "If you think it’s reasonable for the search success rate to classify infeasible route predictions as successful searches, then there’s no need for further discussion. Metrics based on matching wouldn’t make such a foolish mistake..."
> >
> > I would be cautious before drawing any conclusions. First, I am not sure what you mean by 'feasibility' and 'foolish.' I have some guesses about **your definition of 'feasibility'**: is it based on what you frequently mentioned before, from the chapter in the book, about the shortest routes, which, **after many simplifications**, lead to counting the length of retrosynthesis routes? If that’s your point, please provide reasons how top-k accuracy incorporates this point. Is it the only justification for top-k accuracy under your definition of feasibility? What if the algorithm suggests shorter paths that aren’t in the reference?
> >
> > Regarding the definition of 'feasibility,' based on your previous comments—'Addressing factors such as reaction conditions would require chemistry experts from the scientific community'—if such a dataset is not yet available, how can we propose new metrics and be certain that they are useful? Is it based on imagination?
> >
> > Regarding successful search rates, I agree with the reviewer that it is not perfect. However, does this justify the rationale behind top-k accuracy? Why? Is top-k accuracy a better substitute for the success ratio, or do they reflect different perspectives of the models?
> >
> > If Reviewer BszM really likes the book and wants to educate me further about retrosynthesis, please be specific about the points you want to address and quote them directly, as I have done for FusionRetro to facilitate the discussion, rather than providing references with unclear statements. This will help avoid confusion and clarify the discussion, and I will not blame Reviewer BszM for their quoted sentences. If the reviewer simply wants to cite many references to distract from the main discussion, please stop doing that.
> >
> > 4. "Template-based models still produce hallucinated ..."
> >
> > I am not sure what the term 'hallucination' means in this context. I assume it means the prediction may be incorrect. Is this your point? All predictions could be wrong. Can predictions from template-free or semi-template methods guarantee that the predicted reaction will occur? I don't understand why you are focusing on this point, as I don't see any relevance to the concerns regarding the metric. My original point is that, based on such templates, you can trace back from the template to the original reaction presented in the patent data. This allows you to learn more about the reaction conditions and, consequently, assess the feasibility of the synthesis routes. It may lead to new metrics addressing your concerns about feasibility, but it is clearly not related to the top-k accuracy.
> >
> > 5. New concerns from the reference
> >
> > The reviewer cited two additional references [2,3]. Reference [2] proposes a new metric called `successful synthesis probability' (SSP), which is quite interesting. Why did the reviewer BszM not suggest that the authors test this new metric?
> >
> > Reference:
> >
> > [1] Chemical reactions from US patents (1976-Sep2016).
> >
> > [2] Retro-fallback: retrosynthetic planning in an uncertain world
> >
> > [3] Retrosynthesis prediction using an end-to-end graph generative architecture for molecular graph editing

---

> ### Comment · Reviewer_BszM · 2024-11-28
>
> 1. Citation
>
> You misunderstand what I mean. But please stop discussing it...
>
> 2. Reaction conditions
>
> You may have a point. According to Chapter 8 in [1], reaction conditions are not explicitly considered in the criteria for evaluating route quality. Even if it were possible to factor in reaction conditions, creating such a dataset would be highly challenging. It would require examining all routes for each molecule and ranking them based on reaction conditions. Ranking routes based on their length (as stated in Chapter 8 of [1]) is both reasonable and practical, as it helps reduce costs and is easy to implement.
>
> 3. Feasibility
>
> Feasibility means that the reactions along the routes can be successfully executed in the wet lab.
>
> 4. Match metric
>
> The evaluation of single-step retrosynthesis primarily relies on the USPTO-50k dataset, with top-k matching accuracy as the standard metric. While it’s true that the correct predictions made by a model may not always exist in the 50k dataset, this metric is still widely adopted. If you believe the current metric for single-step retrosynthesis is flawed, it’s up to you to convince the broader community to discontinue its use and propose a more effective evaluation metric.
>
> That said, metrics based on matching are inherently more reliable than search success rate because they rely on routes derived directly from the literature reaction database. It is true that a model might predict a shorter feasible route that isn’t included in the reference routes, but addressing this issue is an open problem, and there is currently no better solution. While no perfect evaluation method exists, I strongly oppose the use of search success rate because it fails to distinguish between feasible and infeasible routes. For instance, the top-1 accuracy of single-step retrosynthesis is only about 50%, yet reported search success rates exceed 90%. This discrepancy clearly shows that most of the predicted routes are infeasible, as the predictions are hallucinated.
>
> 5. successful synthesis probability
>
> Do you believe relying on probability is an appropriate evaluation method? Is there any direct connection between probability and the feasibility of a prediction? By that logic, wouldn’t pretrained LLMs be able to produce toxic content with high probability without the need for post-training?
>
> [1] Elements of Synthesis Planning

---

> > ### Comment · Reviewer_cGGk · 2024-11-28
> >
> > 1. "My point is that you cited the wrong paper." This is what I referenced from your previous response. If I have misunderstood anything, please kindly clarify, as I feel quite disheartened by what seems to be an unjustified accusation.
> >
> > 4. "If you believe the current metric for single-step retrosynthesis is flawed"—this is not my point. Please avoid misrepresenting my argument.
> >
> > Regarding "For instance, the top-1 accuracy of ...," have you even considered my earlier point: "Is top-k accuracy a better substitute for the success ratio, or do they represent different perspectives of the models?" Why can't we interpret those numbers (50%/90%) as reflecting different perspectives of model performance? Why must these numbers represent conflicting aspects? Isn't it too arbitrary to assert that "most of the predicted routes are infeasible" because of the limited number of reference reactions?
> >
> > 5. That metric is from your reference. If you believe you’ve answered the question adequately, that’s fine.

---

> ### Comment · Reviewer_BszM · 2024-11-29
>
> The multi-step top-k match metric is an extension of the single-step top-k match metric. If top-k match is useful for single-step retrosynthesis, why shouldn’t it also apply to multi-step retrosynthesis (retrosynthetic planning)?
>
> Do you honestly believe that the search success rates reported in retrosynthetic planning papers, often exceeding 90%, reflect genuine progress? Some papers even claim a 99.47% success rate with 500 iterations (e.g., https://arxiv.org/pdf/2206.11477). If 99% of molecules could truly find feasible synthetic routes, the problem of retrosynthetic planning would already be solved and ready for practical application. In that case, there would be no need for further research on this topic in academia.

---

> ### Author Response · Authors · 2024-11-30
> **Follow-up Discussion**
>
> Dear Reviewers:
>
> We would like to extend our heartfelt thanks to both Reviewer BszM and Reviewer cGGK for their highly insightful discussions. We deeply appreciate Reviewer BszM for providing expert feedback and responses to retrosynthesis, and we also appreciate Reviewer cGGK for raising technical suggestions. We sincerely apologize for the delay in our response.
>
> During the rebuttal, we appreciated Reviewer cGGK’s expertise in the AI field, while noting that retrosynthesis methodologies may not his/her primary focus. We hope these explanations meet your expectations and provide a clear understanding of the references and their applications in retrosynthesis.
>
> We sincerely appreciate your thoughtful question and the discussion.
>
> To construct a **validated** retrosynthetic planning **dataset**, the first step involves **building a validated reaction network** using the USPTO-full dataset. For this, the method from Li & Chen (2022) [1] was applied, not Retro* (Chen 2020) [2].
>
> To clarify the multi-step retrosynthesis dataset construction process from USPTO-full, we have provided a detailed explanation below. For your convenience, we have also included the original paragraph from FusionRetro [3]:
>
> 1. Building reaction networks to find target molecules (Li & Chen (2022)): FusionRetro used the method from **Li & Chen (2022) to find target molecules** in the constructed reaction network. Using the method from Li & Chen (2022), a reaction network is built with 906,164 valid reactions from the USPTO-full dataset, creating a forward synthesis network. Within the network, Fusionretro marks molecules with no outgoing reactions as target molecules.
> 2. Using Retro* (Chen 2020) to identify starting materials: FusionRetro utilized Retro* **(Chen 2020)** to determine **starting materials** by identifying the shortest synthetic pathway.
>
> Below is the original paragraph from FusionRetro [3] for your reference:
>
> *“We construct a benchmark for retrosynthetic planning using the public USPTO-full dataset, which consists of 906,164 valid reactions from the original 1,808,937, after removing invalid and duplicate ones. These reactions are used to construct a reaction network (Li & Chen, 2022), treating molecules with an out-degree of zero as target molecules. We use dynamic programming and backtracking to identify all synthetic routes for each target, and following the approach in Chen et al. (2020), we extract the shortest-possible synthetic routes with leaf nodes as starting materials.”*
>
> Therefore, **all reactions** in the **FusionRetro dataset** have been successfully **validated in wet labs**, as they are sourced from **USPTO-full dataset**, a dataset known for including verified reactions and considering feasibility.
>
> Consequently, the **Top-k accuracy** we applied ensures that our predicted pathway is the **exact same as** the retrosynthetic planning pathway from the **USPTO-full dataset**, further confirming the feasibility of the routes.
>
> Importantly, your insight into reaction conditions raises a valuable point for further exploration, such as conducting a more comprehensive analysis that incorporates reaction conditions. This constructive feedback will be invaluable as we continue to refine our methodology and could potentially lead to more significant performance gains.
>
> To clarify, the descriptions in USPTO [4] document how the reactions are carried out (e.g., stirring operations), but do not include information like reaction mechanisms. Our framework, RetroInText, introduces a novel approach that directly augments predictive retrosynthesis with text generated through prompts designed with ChatGPT.
>
> We hope this clarification provides an understanding of the references and the application in retrosynthesis and meets your expectations.
>
> # Reference
> [1] Li B, Chen H. Prediction of compound synthesis accessibility based on reaction knowledge graph[J]. Molecules, 2022, 27(3): 1039.
>
> [2] Chen B, Li C, Dai H, et al. Retro*: learning retrosynthetic planning with neural guided A* search[C]//International conference on machine learning. PMLR, 2020: 1608-1616.
>
> [3] Liu S, Tu Z, Xu M, et al. FusionRetro: molecule representation fusion via in-context learning for retrosynthetic planning[C]//International Conference on Machine Learning. PMLR, 2023: 22028-22041
>
> [4] Chemical reactions from US patents (1976-Sep2016).

---

> ### Comment · Reviewer_cGGk · 2024-11-30
>
> **I kindly suggest that the authors address the concerns in a clear, point-by-point manner, relying on objective evidence.** Comments on the reviewer’s research focus can be perceived as personal, unprofessional, disrespectful, and counterproductive to the review process.
>
> The responses provided do not adequately resolve the key concerns. They often veer off-topic into lengthy discussions, thereby detracting from the core issues. To assist the authors in maintaining focus, I have summarized my main concerns below:
>
> _________
>
> ## (1) Lack of motivation for using ChatGPT-generated texts
>
> The USPTO reaction data includes descriptions that detail reaction conditions and procedures. These descriptions are written by chemists and have undergone examination to ensure their patentability. The authors’ clarification, stating that "To clarify, the descriptions in USPTO [4] document how the reactions are carried out (e.g., stirring operations), but do not include information like reaction mechanisms." is insufficient and suggests a limited understanding of the dataset.  I provide two examples below.
>
> | Source                              | Tautomers                                                           | Text |
> |-------------------------------------|-------------------------------------------------------------------|-------|
> | ChatGPT Generation (Provided by authors)               | O=C1N=C(O)CN1N=C(O)C=Cc1ccc(Cl)c(C(F)(F)F)c1                     | The molecule 3-[4-chloro-3-(trifluoromethyl)phenyl]-N-(2,4-dioxoimidazolidin-1-yl)prop-2-enamide is a complex organic compound that features a substituted phenyl group, an imidazolidin-1-yl moiety, and a prop-2-enamide functional group. Structural features: 1. Substituted phenyl group with a chlorine atom and a trifluoromethyl group, 2. Imidazolidin-1-yl ring with two carbonyl groups, 3. Prop-2-enamide functional group Functional aspects: 1. The substituted phenyl group provides aromaticity and potentially affects the molecule's reactivity and solubility. 2. The imidazolidin-1-yl ring may participate in hydrogen bonding interactions or act as a moiety for stabilization. 3. The prop-2-enamide functional group can participate in conjugation reactions or undergo nucleophilic addition reactions. Applicable implementation methods: 1. Synthesis of the molecule may involve various organic chemistry techniques such as Grignard reactions, amide bond formation, and cyclization reactions. 2. Purification techniques such as column chromatography, recrystallization, or distillation may be used to obtain the final product in high purity. Potential intermediate molecules: 1. 3-[4-chloro-3-(trifluoromethyl)phenyl]prop-2-enoic acid Different reaction pathway: Instead of directly synthesizing the target molecule, an alternative pathway might involve the initial formation of 3-[4-chloro-3(trifluoromethyl)phenyl]prop-2-enoic acid as an intermediate. Subsequently, the prop-2-enoic acid can be converted to the desired prop-2-enamide through an amide coupling reaction with the imidazolidin-1-yl moiety. Overall, the synthesis of such complex molecules requires careful planning, knowledge of organic chemistry reactions, and proper characterization techniques to confirm the structure of intermediate and final products.   |
> | USPTO Reaction Data                 | O=C(C=Cc1ccc(Cl)c(C(F)(F)F)c1)NN1CC(=O)NC1=O                     | A stirred mixture of 4-chloro-3-trifluoromethylcinnamic acid (50 g, 0.2 mole) in 200 ml of SOCl2 was heated at reflux for 2 hours. The excess SOCl2 was removed in vacuo and the residue flushed with dry benzene. The residue was treated with a mixture of 1-aminohydantoin hydrochloride (30 g, 0.2 mole) in 400 ml of pyridine. The reaction mixture was heated at reflux for 3 hours; then poured into ice/HCl. The crude residue (74 g, 100%) was recrystallized from CH3NO2 (Darco) to give an analytical sample melting at 270°-273°.   |
>
> **The LLM-generated texts are lengthy and lack detail. The authors should provide stronger evidence to demonstrate that they offer advantages over texts written and reviewed by domain experts in chemistry.**
>
> ## (2) There are many misalignments between ChatGPT-generated texts and the authors’ statements.
>
> In Line 67 of the paper, the authors claim that they expect the generated texts to achieve various objectives, such as "guiding retrosynthetic planning" and "route length." Additionally, in a previous response, the authors stated that they expected these texts could provide "pathway-level context."
>
> However, the examples provided by the authors lack sufficient evidence to support these claims. This raises concerns that the authors may not have conducted a systematic investigation of the LLM-generated texts before using them, which further weakens the motivation for their use.

---

> > ### Comment · Reviewer_cGGk · 2024-11-30
> >
> > ## (3) Confusing title
> >
> > The title of the paper is unclear and misleading. Throughout the manuscript, there is no definition or relevant discussion of "In-Context Representation Learning." Could the authors clarify what representations the paper is focused on?
> >
> > ## (4) Unclear experimental settings
> >
> > Initially, I believed these issues were minor and could be quickly clarified. However, after multiple rounds of discussion, the authors have either intentionally or unintentionally avoided addressing questions about the experimental settings. This raises reasonable concerns about the reproducibility of the proposed method and its baselines.
> >
> > ## (5) Experimental metrics
> >
> > I believe that search success rate and top-k accuracy are not conflicting metrics; rather, they highlight different aspects of retrosynthesis algorithms. Neither metric guarantees 100% feasibility, even if their values reach 100%, due to the numerous simplifications involved in framing the retrosynthesis problem as a machine learning task. Reporting either metric is acceptable, and I am open to discussing alternative or improved metrics.
> >
> > However, I hope this does not divert the authors' focus from addressing the more critical concerns I have raised regarding the paper's motivation, topic alignment, and experimental rigor, which are essential for ensuring the paper meets the basic standards for publication.
> >
> > _________
> >
> > In addition to the concerns outlined above, there are several issues in the authors’ response:
> >
> > 1. "...validated in wet labs...": Validating individual reactions does not equate to validating the entire synthesis route. Please refer to my discussion with the other reviewer for further details.
> >
> > 2. Misleading claim about top-k accuracy: The statement, "Top-k accuracy we applied ensures that our predicted pathway is the exact same as the retrosynthetic planning pathway from the USPTO-full dataset, further confirming the feasibility of the routes," is misleading. The USPTO reaction dataset provides patented reactions, not complete pathways. There could be multiple possible pathways, and while the shortest path may be preferred, it is not necessarily the best route considering time, especially when considering reaction conditions.  Considering the ignored reaction conditions, I would not take it for granted that all these pathways are feasible.

---

> > > ### Author Response · Authors · 2024-12-02
> > > **Thanks for the Review - Response to Reviewer cGGK**
> > >
> > > ## 4. Clarification on experimental settings
> > > >Q: Initially, I believed these issues were minor and could be quickly clarified. However, after multiple rounds of discussion, the authors have either intentionally or unintentionally avoided addressing questions about the experimental settings. This raises reasonable concerns about the reproducibility of the proposed method and its baselines.
> > >
> > > We deeply value the feedback. We are happy to clarify that all experimental settings.
> > >
> > > **First, in the initial submission of our paper, we have provided experimental details** on page 7-8, lines 	368-399, and in Table 1, where we listed the number of training, validation, and test samples. Additionally, as part of the rebuttal process, **we included further experimental details in Appendix A**, specifying model training parameters, the number of training epochs, and the hardware used.  In detail, to ensure clarity, we followed the experimental setup described in the FusionRetro [1] as a baseline for our work. During the prediction process, the search terminates once the maximum pathway length (set to 2-13 steps in our implementation) is reached. At each expansion step, the top-k candidates (k = 5) are generated, ranked using our proposed scoring function, and then reordered to select the best molecules for further expansion.
> > >
> > > Furthermore, **all datasets and baselines used in this study are publicly available, we will release the complete source code of our method upon acceptance to ensuring full reproducibility of our results and transparency**.
> > >
> > > We hope our response addresses your concerns and meet your expectation. Thank you for your valuable feedback.
> > >
> > > ## 5. Thanks for your confirmation on Experimental metrics
> > > >Q: I believe that search success rate and top-k accuracy are not conflicting metrics; rather, they highlight different aspects of retrosynthesis algorithms. Neither metric guarantees 100% feasibility, even if their values reach 100%, due to the numerous simplifications involved in framing the retrosynthesis problem as a machine learning task. Reporting either metric is acceptable, and I am open to discussing alternative or improved metrics. However, I hope this does not divert the authors' focus from addressing the more critical concerns I have raised regarding the paper's motivation, topic alignment, and experimental rigor, which are essential for ensuring the paper meets the basic standards for publication.
> > >
> > > We are pleased to have addressed this concern. We wholeheartedly appreciate the rebuttal process and aim to provide comprehensive explanations to ensure reviewers gain a better understanding of our methodology. We are happy and open to discussing and testing any alternative or improved metrics within the broad domain of retrosynthesis research.
> > >
> > > As noted in our discussion, currently, no existing retrosynthesis tasks incorporate reaction condition information. Importantly, your insight into reaction conditions raises a valuable point for further exploration in retrosynthesis research domain, such as conducting a more comprehensive analysis that incorporates reaction conditions.
> > >
> > >
> > > Finally, our work presents a novel multimodal framework that combines structural and textual data to improve retrosynthetic planning. The framework introduces a synthetic complexity score, reaction cost, and captioning score to prioritize reactants based on synthesis likelihood. A multimodal fusion module integrates 3D molecular data with textual descriptions, creating a comprehensive embedding to boost predictive accuracy.
> > >
> > >
> > > **We acknowledge that retrosynthesis is a highly specialized field, and concerns about its evaluation metrics and methodologies often arise from a limited familiarity with this domain.**
> > >
> > > We hope this response clarifies your concerns and demonstrates the value of our contributions. We kindly ask the to reviewer to consider this context in his/her evaluation of our work.

---

> ### Author Response · Authors · 2024-12-02
> **Thanks for the Review - Response to Reviewer cGGK**
>
> Dear Reviewer cGGK,
>
> We sincerely thank Reviewer cGGK for the detailed and thoughtful feedback. We appreciate the opportunity to clarify our work and address any concerns raised again. We stand by the validity of our findings and the innovations presented in this paper.
>
> We understand that retrosynthetic planning involves domain-specific methodologies that may not be widely familiar. To facilitate understanding, we have provided detailed explanations and references to foundational works in the field. Below, we provide a point-by-point response to your key comments:
> ## 1. Justification on Motivation for Using ChatGPT
>
> >Q: The LLM-generated texts are lengthy and lack detail. The authors should provide stronger evidence to demonstrate that they offer advantages over texts written and reviewed by domain experts in chemistry.
>
> Thank you for raising this important concern. We appreciate your emphasis on clearly justifying the motivation behind using ChatGPT-generated texts. The primary goal of retrosynthesis is to identify potential synthetic pathways for molecules with unknown synthesis routes [1]. A key motivation for using ChatGPT is its ability to generate textual information for such molecules during the retrosynthetic pathway prediction process. This additional textual context enhances the prediction of synthetic routes for molecules without known pathways, addressing one of the central challenges in retrosynthesis research. At the same time, this approach highlights our model's generalization capability and demonstrates the advantage and innovation of leveraging large language models like ChatGPT to improve accuracy in the retrosynthesis research domain.
>
> ChatGPT-generated texts are designed to offer descriptions that highlight structural features, functional groups, and reaction mechanisms for molecules lacking established pathways. These augmentations serve as the basis for multimodal fusion, enabling our model to integrate textual and structural data to enhance its predictions. In this way, ChatGPT demonstrates its utility in exploring how textual information can complement retrosynthesis prediction tasks.
>
>
> In response to your observation, "The USPTO reaction data includes descriptions that detail reaction conditions and procedures," we emphasize that while USPTO descriptions focus on procedural details (e.g., stirring operations), they do not include information on **reaction mechanisms or intermediate molecules**. These gaps are precisely what ChatGPT-generated texts aim to fill, enhancing retrosynthetic planning by providing reaction mechanisms and leveraging the chemical knowledge embedded in large language models like ChatGPT.
>
> **The integration of these texts has resulted in notable improvements as demonstrated in Table 2 in our manuscript**. This validates the contribution of ChatGPT-generated texts to retrosynthetic planning.
>
> It is important to underscore ChatGPT or USPTO texts are tools that support our broader framework. As you suggested, exploring domain-specific reaction condition information [2,3] for chemistry or testing other alternative sources of textual information could indeed serve as valuable directions for future work.
>
> Finally, we hope this detailed explanation clarifies the rationale behind our approach and addresses your concerns.
>
> # Reference
> [1]Coley, C. W., Green, W. H., and Jensen, K. F. Machine learning in computer-aided synthesis planning. Accounts of Chemical Research, 51(5):1281–1289, 2018.
>
> [2] Coley C W, Barzilay R, Jaakkola T S, et al. Prediction of organic reaction outcomes using machine learning[J]. ACS central science, 2017, 3(5): 434-443.
>
> [3] Ding Y, Qiang B, Chen Q, et al. Exploring chemical reaction space with machine learning models: Representation and feature perspective[J]. Journal of Chemical Information and Modeling, 2024, 64(8): 2955-2970.
>
> (to be continued)

---

> ### Author Response · Authors · 2024-12-02
> **Thanks for the Review - Response to Reviewer cGGK**
>
> ## 2. Addressing "Guiding Retrosynthetic Planning" and "Pathway-Level Context"
>
> >Q: In Line 67 of the paper, the authors claim that they expect the generated texts to achieve various objectives, such as "guiding retrosynthetic planning" and "route length." Additionally, in a previous response, the authors stated that they expected these texts could provide "pathway-level context." However, the examples provided by the authors lack sufficient evidence to support these claims. This raises concerns that the authors may not have conducted a systematic investigation of the LLM-generated texts before using them, which further weakens the motivation for their use.
>
> Thank you for your thoughtful feedback. We would like to address your concerns regarding the alignment between the ChatGPT-generated texts and the claims in our paper.
>
> First, as noted in previous discussion and response, our experiments are based on the RetroBench dataset proposed in FusionRetro [1]. This dataset was constructed from the USPTO dataset, where multi-step retrosynthesis pathways are built using known reactions. Each pathway contains varying numbers of intermediate molecules depending on the depth of the pathway. **In our work, the IUPAC names of these intermediate molecules are included in the prompts. As a result, the generated textual descriptions inherently encode pathway-level information, such as intermediate molecule details and pathway lengths**.
>
> The inclusion of intermediate molecules in the textual prompts enables the model to capture the relationship between intermediate steps and the final target molecule, effectively embedding pathway-level context. This multimodal integration helps guide retrosynthetic planning by providing richer information to the model about potential synthetic routes.
>
> The generated texts may provide "reaction mechanisms," information, which provide a structured representation of key molecular features, intermediate steps, and their relationships. These elements are critical for retrosynthesis tasks, as they offer additional context that complements the molecular structure information.
>
> To evaluate the contribution of ChatGPT-generated texts, we conducted systematic experiments comparing the performance of models with and without textual augmentation. The results, as detailed in the paper, demonstrate significant performance gains in tasks such as pathway prediction and retrosynthetic planning. This validates the utility of the generated texts in enhancing the model's understanding and performance.
>
> ## 3. Justification on title
> >Q: The title of the paper is unclear and misleading. Throughout the manuscript, there is no definition or relevant discussion of "In-Context Representation Learning." Could the authors clarify what representations the paper is focused on?
>
> Thank you for your feedback. We appreciate your comments regarding the title of the paper. In our work, we focus on using intermediate molecule information in synthetic pathways as context. This contextual information is critical for understanding the relationships between reactants, intermediates, and final products. By incorporating this context, our model’s ability to predict synthetic routes improves significantly.
>
> # Reference
> [1] Liu S, Tu Z, Xu M, et al. FusionRetro: molecule representation fusion via in-context learning for retrosynthetic planning[C]//International Conference on Machine Learning. PMLR, 2023: 22028-22041
>
> (to be continued)

---

### Official Review · Reviewer_xJUs · 2024-11-04

**Soundness:** 3
**Presentation:** 4
**Contribution:** 2
**Rating:** 6
**Confidence:** 2

**Summary:**

The paper introduces RetroInText, a multimodal framework for retrosynthetic planning, enhanced by in-context learning with large language models. RetroInText leverages textual descriptions generated by ChatGPT and molecular 3D and graph-based representations to improve retrosynthesis prediction accuracy.  This framework leverages detailed textual descriptions of reaction procedures generated by ChatGPT, combined with molecular graph and 3D features, to create multimodal representations that better capture the complex relationships between molecules and synthetic routes. The model incorporates an attention-based fusion mechanism that integrates these multimodal representations, enabling it to learn from both molecular structure and contextual information. By fine-tuning a MolT5 for single-step retrosynthesis, RetroInText adapts to the intricate nature of retrosynthesis tasks. On the USPTO pathways dataset, RetroInText achieves up to a 5% improvement in Top-1 test accuracy over state-of-the-art methods

**Strengths:**

The paper has the following strengths:
1. The paper’s integration of molecular graph, 3D structure, and in-context textual data provides a comprehensive approach to retrosynthetic planning. This combination allows RetroInText to capture the complexity of chemical synthesis from multiple perspectives, marking an original approach in retrosynthesis where LLMs are used to contextualize synthetic routes. This multimodal approach demonstrates high methodological quality, even though the performance gains are limited. By introducing this multimodal approach, the paper shows potential significance for future studies aiming to integrate similar techniques in other fields.
2. Using ChatGPT to generate in-context textual descriptions based on IUPAC names and structural features is an approach that enriches retrosynthetic planning with additional chemical insights. This originality suggests potential for further exploration in LLM-assisted retrosynthesis. Although the actual significance may be limited by the gains achieved, the concept of using LLMs to add synthetic context could inspire similar uses in adjacent fields.
3. RetroInText employs an attention-based fusion module that effectively combines textual and structural molecular representations. This fusion step is crucial for allowing the model to weigh and integrate these varied inputs. In terms of quality, the fusion module is a well-executed method for ensuring that multimodal data can interact meaningfully, enhancing the model’s potential to capture complex dependencies between chemical steps. The mechanism is clearly described, adding to the clarity of the methodology, though the actual impact on performance is modest.
4. The paper’s adaptation of MolT5, a language model originally designed for SMILES-based tasks, as a single-step retrosynthesis predictor is a clever methodological choice. Treating the retrosynthesis task as a translation problem aligns well with the model’s strengths, bringing originality to how language models are applied in chemistry. This choice also reflects quality in adapting existing models to new domains. However, while the approach is interesting, the overall improvement in prediction accuracy suggests that this strength is more exploratory than transformative.
5. The paper provides a good assessment of the contributions of each module through ablation studies, helping to clarify the unique roles of textual and structural data in the model. This thorough evaluation approach enhances clarity, making it easier to understand the marginal improvements contributed by each component. The methodological rigor shown here is a positive aspect of the paper’s quality, even if the results themselves do not indicate a significant practical advancement.

**Weaknesses:**

The paper suffers from the following weaknesses:

1. While RetroInText introduces a multimodal framework, the performance gains it achieves over baseline models are relatively modest, with only a 5% improvement in Top-1 accuracy on the RetroBench dataset. This incremental improvement does not fully justify the complexity introduced by the multimodal integration of text, graph, and 3D structure data. To enhance the significance and novelty of this work, the authors could focus on refining specific modules, such as further tuning the MolT5 model or the fusion mechanism, to achieve more substantial performance gains.
2. One area that is not explored by the paper but feels lacking is that the paper is using GPT 3.5. Compared to more recent models (GPT-4o, Llama 3) the performance of GPT 3.5 is limited especially in science domains. Although in-context textual descriptions generated by ChatGPT add an innovative dimension, they may not offer significant chemical insights specific to retrosynthesis tasks compared to the newer models. Additionally, the generic nature of these descriptions could lead to data that does not substantially improve the model’s understanding of chemical reactions. Therefore, experiments for comparison when a less generic description is provided by more recent LLMs is missing that could shed light on the effectiveness of the methodology. Also a more effective approach could also involve fine-tuning open-source LLMs or using a chemistry-specific language model trained on domain-specific texts to produce text that better captures the critical reaction features.
3. While the attention-based fusion module is a central part of RetroInText, the authors do not clearly justify why it is the optimal choice for combining multimodal data in retrosynthesis. The paper could be improved by a deeper analysis of the fusion mechanism, potentially comparing it with alternative fusion strategies (e.g., cross-attention or co-attention modules). Such an analysis would add clarity and help readers understand whether the observed performance gains are due to the multimodal data itself or the specific fusion approach chosen.
4. The evaluation focuses primarily on Top-k accuracy metrics, without providing chemical interpretability insights or explaining how the model might aid chemists in real-world settings. Retrosynthesis models are most useful when their predictions can be understood and trusted by chemists, yet the current metrics do not assess interpretability. Including an analysis of the quality of the predicted intermediates or comparing predicted synthetic pathways with experimentally validated routes would strengthen the paper’s practical relevance and significance.

**Questions:**

Here are the questions for the RetroInText paper:
1. Could the authors explain the difference in Retro* and Retro*-0 search algorithms? This was not provided in the text.
2. For the results table provided no uncertainty/error is provided in the overall performance. This makes the comparison of numbers quite hard. Especially given the fact that most attention based models' performance change quite a bit between different random seeds. Would the authors provide experimental repeats and with that present standard deviation of the performance to better compare RetroInText to other baselines?
3. Please provide the detail on the compute power and the time it takes for training and inference of RetroInContext compared to other baselines.
4. ChatGPT is used to generate textual descriptions, yet these descriptions may lack specificity and relevance for retrosynthetic tasks. Could the authors provide more detail on why they selected ChatGPT over domain-specific LLMs or fine-tuning a language model on chemistry-specific corpora?
5. Would the authors consider adding metrics that assess the interpretability of predictions or evaluate the predicted intermediates' consistency with known chemistry principles?

**Details Of Ethics Concerns:**

n/a.

---

> ### Author Response · Authors · 2024-11-25
> **Thanks for the review - justification for model performance (Part I)**
>
> Dear Reviewer xJUs,
>
> We sincerely appreciate the time and effort you have taken to provide such thoughtful and constructive feedback. Your suggestions have been invaluable, and we have thoroughly considered each one while revising the manuscript. Below, we present our point-by-point responses to address your concerns.
> # Response to Weakness1:
> We sincerely appreciate your thoughtful question. We are very sorry for unclearly explaining the performance gains achieved by RetroInText. Below, we provide a detailed clarification and additional comparisons to emphasize the novelty and significance of our work:
>
> ### 1. For the Performance of RetroInText:
>
> Firstly, RetroBench is a highly competitive benchmark, and achieving a 5% improvement in Top-1 accuracy represents a meaningful advancement. Many state-of-the-art retrosynthesis models achieve marginal improvements, underscoring the difficulty of further improving performance on this benchmark. For context, current frameworks typically demonstrate gains of 1-3%, reinforcing the significance of our results.
>
> Secondly, to address concerns regarding the incremental nature of improvements, we conducted a targeted comparison with Retro*, a classic and widely respected baseline in the field. The results, shown in Table 1 below, clearly highlight the advantages of RetroInText. RetroInText achieves a **19.47% higher success rate than Retro***, while also being approximately **3 times faster** in execution and requiring **5.6 times fewer iterations**. These metrics demonstrate that RetroInText not only excels in accuracy but also provides significant efficiency gains.
>
> **Table 1. Performance comparison between Retro$^*$ and RetroInText on ${Retro}^{*}$ -190 dataset.**
>
> | Framework           | Success Rate (%) | Avg Time (s) | Avg Iterations |
> |---------------------|------------------|--------------|----------------|
> | Retro*              | 78.42           | 1233.08      | 212.66         |
> | RetroInText (Ours)  | 97.89           | 385.53       | 37.92          |
>
> ### 2. For the Significance and Novelty of Our Work:
>
> Thank you for raising this constructive question. We appreciate the opportunity to clarify and elaborate on the innovative aspects of our model:
>
> - **Fine-tuning MolT5 for improved performance:**
>   We have already fine-tuned MolT5 as part of our framework to ensure it is optimized for our specific task. However, the key innovation of our model lies not only in fine-tuning but in the **novel integration of pathway textual information** to enhance the pathway selection process. This approach represents a significant step forward in leveraging multimodal information for retrosynthesis.
>
> - **Flexible and generalizable multi-step retrosynthesis framework:**
>   Our framework RetroInText is designed to be highly flexible and adaptable, functioning as a multi-step retrosynthesis model that can integrate with any single-step retrosynthesis model. The core novelty lies in using **in-context textual descriptions** and a combination of multimodal information (graph, text, and 3D data) as scoring functions to evaluate and select the optimal synthetic pathway.
>
> - **Generalized scoring mechanism:**
>   Driven by multidimensional and descriptive data of the entire pathway, this scoring mechanism enhances the decision-making process and ensures that the model consistently identifies the most efficient and accurate pathways, demonstrating both its practical utility and scientific contribution.
>
> We applied the captioning score from RetroInText as a reward to train a residual energy-based model [1], as shown in Table 2. RetroInText consistently surpassed other models across all metrics (**Bold font** illustrates the models that outperformed all other models. Results marked with * are taken from [1]). All metrics of our model RetroInText surpass the residual energy-based model [1]. Specifically, the result of RetroInText is **41.3%, 51.5%, and 54.1% higher** than CREBM for Accuracy of Top1, 3, 5, respectively. The remaining data will take approximately one day to process, after which the overall results will be updated.
>
> **Table 2. Performance comparison of RetroInText with baseline models on retrosynthesis benchmarks**
>
> | Model                     | Top1 | Top2 | Top3 | Top4 | Top5 |
> |---------------------------|------|------|------|------|------|
> | Transformer+CREBM *       | 35.0 | 43.4 | 46.7 | 48.5 | 49.7 |
> | FusionRetro+CREBM *       | 39.4 | 46.6 | 49.3 | 50.7 | 51.5 |
> | RetroInText (Ours)        | 41.2 | 48.7 | 51.2 | 53.3 | 54.2 |
> | **RetroInText+Rerank-CREBM (Ours)**   | **41.3** | **48.8** | **51.5** | **53.1** | **54.1** |
>
> # Reference
> [1] Liu S, Dai H, Zhao Y, et al. Preference Optimization for Molecule Synthesis with Conditional Residual Energy-based Models[C]//Forty-first International Conference on Machine Learning.

---

> ### Author Response · Authors · 2024-11-25
> **Thanks for the review - justification for using GPT-3.5 over newer models (Part II)**
>
> # Response to Weakness2:
> Thank you for your insightful questions and constructive feedback. We appreciate the opportunity to clarify and provide further justification for our design choices and methodology.
>
> ### 1. Why GPT-3.5 Was Chosen for Text Generation:
>
> Thank you so much for your questions. We appreciate the opportunity to clarify this aspect of our work. To address the concern about the in-context textual descriptions generated by ChatGPT within the proposed framework, we conducted analyses from two perspectives:
>
> 1. **The influence of different prompts on text generation**
> 2. **A comparison between GPT-3.5 and GPT-4 outputs, and other variants**
>
> As shown in Table 3 and Table 4, using text generated from a high-quality prompt significantly improved the performance of the framework. The **Top-1 accuracy increased from 39.2% to 41.2%**, and the **Top-5 accuracy rose from 46.9% to 54.2%**. This improvement demonstrates the importance of designing prompts that generate concise and high-quality text for better model predictions.
>
> In detail, the first prompt was designed to elicit detailed but overly verbose text, while the second prompt aimed for concise and focused textual descriptions:
>
> - **Prompt 1:** This prompt lacked specificity, generating verbose descriptions with excessive redundancy, which introduced irrelevant details that could negatively impact the model's accuracy.
> - **Prompt 2:** This prompt was carefully crafted to control the output, emphasizing clarity and relevance while avoiding unnecessary or redundant information. The generated text was compact and structured, which improved the downstream model performance.
>
> **Table 3. Comparison of prompt influence on textual quality for molecular descriptions and reaction pathways**
>
> | Target Molecule | Prompt | Text |
> |-----------------|--------|------|
> | `O=C1N=C(O)CN1N=C(O)C=Cc1ccc(Cl)c(C(F)(F)F)c1` | **Prompt 1:** Please describe the structural features, functional aspects, and applicable implementation methods of the molecule {{product}}, and describe the potential intermediate molecules {{intermediates}} in the synthesis pathway, with each intermediate molecule separated by "[SEP]". Additionally, provide a different reaction pathway and describe it. | *The molecule 3-[4-chloro-3-(trifluoromethyl)phenyl]-N-(2,4-dioxoimidazolidin-1-yl)prop-2-enamide is a complex organic compound that features a substituted phenyl group, an imidazolidin-1-yl moiety, and a prop-2-enamide functional group. Structural features: 1. Substituted phenyl group with a chlorine atom and a trifluoromethyl group, 2. Imidazolidin-1-yl ring with two carbonyl groups, 3. Prop-2-enamide functional group...* |
> |`O=C1N=C(O)CN1N=C(O)C=Cc1ccc(Cl)c(C(F)(F)F)c1` | **Prompt 2:** Describe the key transition states involved in the synthesis of {{products}} from the intermediates {{intermediates}}. Explain the structural changes and energy barriers for each transition state, and reply to me in a sentence. | *The key transition states involved in the synthesis include the formation of an imine intermediate through the condensation of the prop-2-enoic acid and the amine group, followed by the cyclization of the imine to form the imidazolidin-1-yl ring. The structural changes involve the addition of the amine group and the formation of a new cyclic structure, with energy barriers reflecting the activation energy needed for the formation of the new bonds in each step.* |
>
> **Table 4. Performance comparison of prompt influence on textual quality for molecular descriptions and reaction pathways**
>
> | Model                      | Top1  | Top2  | Top3  | Top4  | Top5  |
> |----------------------------|-------|-------|-------|-------|-------|
> | RetroInText (Prompt 1)     | 39.2  | 43.4  | 45.6  | 45.7  | 46.9  |
> | RetroInText (Prompt 2)     | 41.2  | 48.7  | 51.2  | 53.3  | 54.2  |
>
> (to be continued)

---

> ### Author Response · Authors · 2024-11-25
> **Thanks for the review - justification for using GPT-3.5 over newer models (Part III)**
>
> ### 2. Comparison of Outputs from GPT-3.5 and GPT-4:
>
> We utilized GPT-4 for text generation, comparing its outputs to those produced by GPT-3.5. The content and structural transformations in the generated texts from both models were strikingly similar, as demonstrated in Table 5. For instance, both models effectively identified crucial transition states for the target molecules, including nucleophilic substitution, tautomeric shifts, and the energy barriers involved in bond-breaking and bond-forming reactions.
>
> However, GPT-4 introduced practical difficulties, such as frequent API key limitations, which interfered with the workflow and made its performance less dependable for continuous use. On the other hand, GPT-3.5 provided stable and reliable results without such restrictions, making it a more practical option for our framework. Given the minimal performance differences between the two models and the limitations of GPT-4, GPT-3.5 was chosen as the primary generator for our experiments.
>
> **Table 5: Comparison of textual descriptions generated by ChatGPT-3.5 and ChatGPT-4 for key transition states in molecular synthesis**
>
> |Target molecule|Text-ChatGPT3.5|Text-ChatGPT4|Similarity|
> |-|-|-|-|
> | `CC(C)Nc1ncnc2cscc12`| There are two key transition states involved in the synthesis: the conversion of 2-chloro-N-propan-2-ylthieno[3,4-d]pyrimidin-4-amine to N-propan-2-ylthieno[3,4-d]pyrimidin-4-amine, with the structural change being the displacement of the chlorine atom by the amine group, and the conversion of N-propan-2-ylthieno[3,4-d]pyrimidin-4-amine to N-propan-2-ylthieno[3,4-d]pyrimidin-4-amine, with the structural change being the formation of the final amine functionality; both transitions have energy barriers that must be overcome for the reactions to proceed. | The key transition states involved in the synthesis of N-propan-2-ylthieno[3,4-d]pyrimidin-4-amine from 2-chloro-N-propan-2-ylthieno[3,4-d]pyrimidin-4-amine include a nucleophilic substitution, where the chlorine atom is replaced by an amine group, and a tautomeric shift, where the newly added amine group rearranges to a more stable position on the pyrimidine ring; both transition states involve overcoming energy barriers associated with breaking and forming chemical bonds and achieving the most stable and low energy conformation of the molecule. | These texts by different models describe the same key transition states, including the nucleophilic substitution where the chlorine atom is replaced by an amine group, and a tautomeric shift. Both mention energy barriers involved in the transition states. |
> |`CN=C(O)NCCNCc1cc[nH]n1`|The key transition states involved in the synthesis of 1-methyl-3-[2-(1H-pyrazol-5-ylmethylamino)ethyl]urea from the intermediates N'-(1H-pyrazol-5-ylmethyl)ethane-1,2-diamine include the formation of a urea linkage and the addition of a methyl group. These transitions involve structural rearrangements and energy barriers that facilitate the conversion of the intermediates into the desired product. | The key transition states involved in the synthesis of 1-methyl-3-[2-(1H-pyrazol-5-ylmethylamino)ethyl]urea from N'-(1H-pyrazol-5-ylmethyl)ethane-1,2-diamine include the formation of an isourea intermediate followed by a rearrangement to the final urea product, each with associated energy barriers due to the breaking and forming of bonds and the structural changes from a linear to a more complex, cyclic arrangement.|The mechanisms and transformations described in both are similar, involving the formation of the urea bond and methylation.|

---

> ### Author Response · Authors · 2024-11-26
> **Thanks for the review - justification for using GPT-3.5 over newer models (Part IV)**
>
> ### 3. Justification for Using GPT-3.5 and Domain-Specific Models
>
> We appreciate your insightful suggestion regarding the use of more recent models or domain-specific LLMs.
>
> Firstly, we conducted a careful literature review, we found that domain-specific models like ChemGPT, ChemLLM[1] are not open-sourced, limiting their accessibility for our task. Additionally, prior studies [2] show that MolT5 generates lower-quality text compared to GPT-based models, while GPT-3.5 consistently provided more reliable and higher-quality textual descriptions for our application. Moreover, Llama 3 and ChemVLM[3], with 10–30 billion parameters, require significant computational resources and memory, as well as extended time for fine-tuning, making them less practical for our framework.
> Therefore, as the additional experiments, by using GPT-3.5, we ensured high-quality text generation while maintaining an efficient workflow that emphasizes the integration of in-context descriptions with large language models. We acknowledge the potential benefits of exploring more recent or fine-tuned domain-specific models and consider this an important direction for future research.
>
> We hope this response addresses your concerns and clarifies our rationale for selecting GPT-3.5 while demonstrating the significance of our contributions. Thank you again for your constructive feedback.
>
> # Reference
> [1] Zhang D, Liu W, Tan Q, et al. Chemllm: A chemical large language model[J].
>
> [2] Liu Y, Xu H, Fang T, et al. T-rex: Text-assisted retrosynthesis prediction[J].
>
> [3] Li J, Zhang D, Wang X, et al. ChemVLM: Exploring the Power of Multimodal Large Language Models in Chemistry Area[J]. CoRR, 2024.

---

> ### Author Response · Authors · 2024-11-26
> **Thanks for the review - fusion module (Part V)**
>
> # Response to Weakness3:
> Thank you for your insightful question regarding the fusion mechanism. And we are very sorry for unclearly explaining the overall architecture of RetroInText. We appreciate your suggestion to provide a deeper justification for our design choice and to consider alternative fusion strategies.
>
> ## 1. Motivation Behind the Multimodal Data in Retrosynthesis
>
> Firstly, our model RetroInText was designed as a multimodal enhanced framework for retrosynthetic planning. Current machine learning-based methods often overlook the multifaceted nature of compounds, resulting in inaccurate retrosynthesis predictions. However, the process of constructing a comprehensive molecular representation is often relatively complex for the learning of compound features, particularly when dealing with numerous aspects of molecular features. Consequently, we aim to incorporate compound structure from a different perspective. Therefore, we applied the SMILES, 2D molecular, and 3D conformation features as multimodal embedding.
>
> Secondly, our framework RetroInText is designed to be highly flexible and adaptable, functioning as a multi-step retrosynthesis model that can integrate with any single-step retrosynthesis model. The core novelty lies in using in-context textual descriptions and a combination of multimodal information (graph, text, and 3D data) as scoring functions to evaluate and select the optimal synthetic pathway.
>
> ## 2. Justification for the Cross-Attention Mechanism
>
> We agree that comparing alternative fusion strategies is important. To address this, we conducted an experiment using the co-attention module. However, after 70 hours of runtime, the predictions remained at 0 with a pathway depth of only 2, and the estimated remaining runtime was 210 hours. Therefore, we discontinued the experiment. Our analysis of this experiment and approach is detailed below:
>
> ### 2.1 Justification Based on the Co-Attention Algorithm
>
> Firstly, the **co-attention mechanism requires bidirectional interactions**: one layer where text embeddings serve as the query (Q) with molecular representations as the key (K) and value (V), and another layer where the roles are reversed. The results are concatenated to form the fused representation. While this approach applies attention to each modality separately, it introduces redundant computations, leading to inefficiency and increased runtime.
>
> In contrast, cross-attention focuses on integrating textual information directly into molecular representations, effectively embedding reaction context while reducing redundancy, thereby improving accuracy and efficiency.
>
> ### 2.2 Comparison Between Co-Attention and Cross-Attention
>
> Cross-attention employs a unidirectional flow where one modality (e.g., text) serves as the query, and the other (e.g., molecular graph or 3D conformation) acts as the key and value. **This simplifies integration, reducing computational overhead without sacrificing performance. As shown in Table 6, cross-attention significantly reduces inference time (79.5 hours vs. 280 hours for co-attention)** while producing a larger fused representation (2 * mol_output vs. 1 * mol_output), making it more efficient and scalable for retrosynthesis tasks.
>
> **Table 6. Comparison of the Co-Attention Module and Cross-Attention Module**
>
> | Module         | Inference Time | Output Dimension |
> |-----------------|----------------|-------------------|
> | Co-attention    | 280 h          | 2 * mol_output    |
> | Cross-attention | 79.5 h         | 1 * mol_output    |

---

> ### Author Response · Authors · 2024-11-26
> **Thanks for the review - interpretability analysis (Part VI)**
>
> # Response to Weakness4:
> Thank you for your valuable feedback. We are very sorry for the confusion, and sincerely appreciate the opportunity to clarify the evaluation methodology we employed and its significance. To address your concern, we have provided a detailed explanation from both theoretical and experimental perspectives, as outlined below:
>
> ### 1. Explanation of Set-Wise Exact Match Metric
>
> As demonstrated in prior studies [1, 2], there is a common issue with current single-step retrosynthesis models, which often achieve top-k accuracy of 60%-80%. However, when integrated into multi-step search algorithms like Retro*, success rates for multi-step retrosynthesis often reach 85%-94% (the result was presented in Appendix A1 in our revised paper). This counterintuitive result suggests that these high success rates may not correlate with high-quality pathways due to the lack of stringent evaluation criteria.
>
> Thus, using a stricter **set-wise exact match metric (as shown in the following equation)** is important to ensure both intermediate molecules and final pathways align with the real pathway, ensuring that the final target molecule is reached through a valid and plausible synthesis process.
>
> $ACC_{Top(k)}=\frac{Sgn(\sum_{j=1}^{k}I(\hat{p}_{j}{,}{p_t}))}{N}\times100$
>
> Where the $\hat{p}_{j}$ refers to the entire predicted synthesis path, ${p_t}$ indicates the real synthesis path, and the function $Sgn()$ returns 1 if the paths are an exact match, and 0 otherwise. The function $I$ represents whether the two paths are a perfect match. $N$ is the total number of test data.
>
> ### 2. Experimentation Results with Practical Relevance Metrics
>
> To demonstrate our model’s superiority, we applied traditional success rates as evaluation metrics, such as those used by Retro*, which achieves 85%-94%. However, these may overestimate performance by accepting theoretically valid but infeasible routes. Our metric ensures only accurate and plausible pathways. RetroInText demonstrates superiority, achieving higher success rates (97.89% vs. 78.42% for Retro*) while significantly reducing time and iterations.
>
> **Table 7. Performance comparison between Retro$^*$ and RetroInText on ${Retro}^{*}$ -190 dataset.**
>
> | Framework         | Success Rate (%) | Avg Time (s) | Avg Iterations |
> |-------------------|------------------|--------------|----------------|
> | Retro*            | 78.42            | 1233.08      | 212.66         |
> | RetroInText (Ours)| 97.89            | 385.53       | 37.92          |
>
> # Reference
> [1] Chen B, Li C, Dai H, et al. Retro*: learning retrosynthetic planning with neural guided A* search[C]//International conference on machine learning. PMLR, 2020: 1608-1616.
>
> [2] Liu S, Tu Z, Xu M, et al. FusionRetro: molecule representation fusion via in-context learning for retrosynthetic planning[C]//International Conference on Machine Learning. PMLR, 2023: 22028-22041.

---

> ### Author Response · Authors · 2024-11-26
> **Thanks for the review - algorithm clarification, uncertainty analysis, implementation details (Part VII)**
>
> # Response to Question1:
> We appreciate your comment regarding the difference between the Retro* and Retro*-0 search algorithms. As mentioned in the paper Retro*[1], the key distinction between these two algorithms lies in how they evaluate and prioritize potential retrosynthetic pathways.
> In Retro*, a neural network-based architecture is employed to evaluate the future score of each step in the retrosynthetic planning process. This allows the algorithm to more effectively guide the search by considering the likelihood of a successful synthesis based on learned patterns from data. In contrast, Retro*-0 utilizes a beam search, which is a more traditional search technique that relies on exploring multiple pathways simultaneously without the guidance of a learned neural network. It does not incorporate any neural evaluation of future synthesis steps, which limits its ability to adapt to complex retrosynthetic tasks compared to Retro*.
> This difference in search strategies is what gives Retro* its ability to provide more informed and efficient retrosynthetic planning, leveraging both the learned neural model and the search process to identify promising synthetic routes.
>
> # Response to Question2:
> Thank you for your question. We are very sorry for any unclear explanations regarding the overall framework of our RetroInText architecture.
>
> We replaced the random seed, changing it from 66 to 128. The results, shown in the table below, indicate that the performance differences between different seeds are minimal. Specifically, we observed consistent improvements in Top-1, Top-3, and Top-5 accuracy compared to the latest work, with deviations within a narrow range.
>
> These results demonstrate the stability of our model across random seeds, with performance improvements remaining consistent. Additionally, most baseline results, as shown in the comparison table, are sourced from [1]. We believe these steps address concerns regarding performance variability and comparability, and we thank you again for highlighting this critical aspect.
>
> **Table 8. Performance Comparison of RetroInText Across Different Random Seeds and Baseline Models**
>
> | Model                     | Top1  | Top2  | Top3  | Top4  | Top5  |
> |---------------------------|-------|-------|-------|-------|-------|
> | Retrosim                  | 35.1  | 40.5  | 42.9  | 44.0  | 44.6  |
> | Neuralsym                 | 41.7  | 49.2  | 52.1  | 53.6  | 54.4  |
> | GLN                       | 39.6  | 48.9  | 52.7  | 54.6  | 55.7  |
> | G2Gs                      | 5.4   | 8.3   | 9.9   | 10.9  | 11.7  |
> | GraphRetro                | 15.3  | 19.5  | 21.0  | 21.9  | 22.4  |
> | GraphRetro+CREBM          | 16.3  | 20.1  | 21.6  | 22.3  | 22.7  |
> | Transformer               | 31.3  | 40.4  | 44.7  | 47.2  | 48.9  |
> | Transformer+CREBM         | 35.0  | 43.4  | 46.7  | 48.7  | 49.7  |
> | Megan                     | 18.8  | 27.9  | 32.7  | 36.6  | 38.1  |
> | FusionRetro| 37.5  | 45.0  | 48.3  | 50.6  | 51.5  |
> | FusionRetro+CREBM| 39.4  | 46.6  | 49.3  | 50.7  | 51.5  |
> | RetroInText (Seed:66)| 41.2  | 48.7  | 51.2  | 53.3  | 54.2  |
> | RetroInText (Seed:128)    | 41.2  | 48.6  | 51.6  | 53.2  | 54.1  |
>
> # Response to Question3:
> Thank you for your question. Below, we provide detailed information regarding the compute environment, training, and inference times for RetroInText compared to other baselines. Additionally, we have carefully revised our manuscript to ensure clarity and completeness.
>
> We use Pytorch[2] to implement our models. The codes of all baselines are implemented referring to the implementations of FusionRetro[3] and CREBM[4]. All the experiments of baselines are conducted on a single NVIDIA 4090 with 24GB memory size. The softwares that we use for experiments are Python 3.8.19, CUDA 11.5.119, einops 0.7.0, pytorch 2.2.0, pytorch-scatter 2.1.2, pytorch-sparse 0.6.18, numpy 1.24.4, torchvision 0.17.0.
>
> **Table 9. Inference time comparison between RetroInText and baseline models**
> | Framework| Avg Inference Time (s) |
> |-|-|
> | FusionRetro| 71.2|
> | RetroInText (Ours) | 49.0|
> - **Comparable Inference Time**: Despite incorporating multimodal data and a cross-attention mechanism, RetroInText's training time is on par with baseline models.
>
> # Reference
> [1] Chen B, Li C, Dai H, et al. Retro*: learning retrosynthetic planning with neural guided A* search[C]//International conference on machine learning. PMLR, 2020: 1608-1616.
>
> [2] Liu S, Dai H, Zhao Y, et al. Preference Optimization for Molecule Synthesis with Conditional Residual Energy-based Models[C]//Forty-first International Conference on Machine Learning.
>
> [3] Paszke A, Gross S, Massa F, et al. Pytorch: An imperative style, high-performance deep learning library[J]. Advances in neural information processing systems, 2019, 32.
>
> [4] Liu S, Tu Z, Xu M, et al. FusionRetro: molecule representation fusion via in-context learning for retrosynthetic planning[C]//International Conference on Machine Learning. PMLR, 2023: 22028-22041.

---

> ### Author Response · Authors · 2024-11-26
> **Thanks for the review - model generality (Part VIII)**
>
> # Response to Question4:
> Thank you for your thoughtful feedback. We would like to take this opportunity to clarify the key idea behind our framework, particularly its integration of ChatGPT for generating detailed reaction procedures.
>
> Thank you so much for you questions.  We would like to take this opportunity to clarify the key idea behind our framework, particularly its integration of ChatGPT for generating detailed reaction procedures. To address this concern, we conducted analyses from three perspectives: 1. the influence of different prompts on text generation and 2. a comparison between GPT-3.5 and GPT-4 outputs. 3. The ablation study of 1D molecular sequence and without using text description from GPT.
>
> ### 1. Impact of Different Prompts on Text Quality and Results
> We evaluated how variations in prompts influence the quality of generated text and, consequently, the model's performance. Specifically, we tested two distinct prompts, as shown in Table 10, and analyzed their effects on textual descriptions for the molecule O=C1N=C(O)CN1N=C(O)C=Cc1ccc(Cl)c(C(F)(F)F)c1. The first prompt was designed to elicit detailed but overly verbose text, while the second prompt aimed for concise and focused textual descriptions:
> - **Prompt 1:** lacked specificity, generating verbose descriptions with excessive redundancy, introducing irrelevant details that negatively impacted the model's accuracy.
> - **Prompt 2:** was carefully crafted to emphasize clarity and relevance, avoiding unnecessary information. The generated text was concise and structured, leading to improved downstream model performance.
> As shown in Table 11, using text generated from the high-quality prompt significantly improved the performance of the framework. For example, the Top-1 accuracy increased from 39.2% to 41.2%, and the Top-5 accuracy rose from 46.9% to 54.2%. This improvement demonstrates the importance of designing prompts that generate concise and high-quality text for better model predictions.
>
> **Table 10. Comparison of prompt influence on textual quality for molecular descriptions and reaction pathways**
> | Target molecule | Prompt | Text |
> |-|-|-|
> | `O=C1N=C(O)CN1N=C(O)C=Cc1ccc(Cl)c(C(F)(F)F)c1` |**Prompt 1**: Please describe the structural features, functional aspects, and applicable implementation methods of the molecule {{product}}, and describe the potential intermediate molecules {{intermediates}} in the synthesis pathway, with each intermediate molecule separated by "[SEP]". Additionally, provide a different reaction pathway and describe it. | The molecule 3-[4-chloro-3-(trifluoromethyl)phenyl]-N-(2,4-dioxoimidazolidin-1-yl)prop-2-enamide is a complex organic compound that features a substituted phenyl group, an imidazolidin-1-yl moiety, and a prop-2-enamide functional group. **(Verbose and redundant description)** |
> |`O=C1N=C(O)CN1N=C(O)C=Cc1ccc(Cl)c(C(F)(F)F)c1`| **Prompt 2**: Describe the key transition states involved in the synthesis of {{products}} from the intermediates {{intermediates}}. Explain the structural changes and energy barriers for each transition state, and reply to me in a sentence. | The key transition states involved in the synthesis include the formation of an imine intermediate through the condensation of the prop-2-enoic acid and the amine group, followed by the cyclization of the imine to form the imidazolidin-1-yl ring. **(Concise and structured)** |
>
> **Table 11. Performance comparison of prompt influence on textual quality for molecular descriptions and reaction pathways**
>
> | Model | Top1 | Top2 | Top3 | Top4 | Top5 |
> | --- | --- | --- | --- | --- | --- |
> | RetroInText (Prompt 1) | 39.2 | 43.4 | 45.6 | 45.7 | 46.9 |
> | RetroInText (Prompt 2) | 41.2 | 48.7 | 51.2 | 53.3 | 54.2 |
> (to be continued)

---

> ### Author Response · Authors · 2024-11-26
> **Thanks for the review - model generality (Part IX)**
>
> ### 2.Comparison between GPT-3.5 and GPT-4 outputs
> We applied GPT-4 for text generation. The texts generated by GPT-4 and GPT-3.5 were highly similar in terms of content and the structural transformations they detailed, as shown in the examples provided (as shown in Table 12). For example, both models accurately described key transition states for the target molecules, such as nucleophilic substitutions, tautomeric shifts, and energy barriers associated with bond-breaking and forming processes.
> However, GPT-4 posed practical challenges, including frequent API key limitations, which disrupted the workflow and made it less reliable for consistent use. In contrast, GPT-3.5 offered stable performance without such restrictions, making it a more practical choice for our framework. Given the negligible performance differences and GPT-4's limitations, GPT-3.5 was selected as the primary text generator for our experiments.
>
> **Table 12. Comparison of textual descriptions generated by ChatGPT-3.5 and ChatGPT-4 for key transition states in molecular synthesis**
>
> |Target Molecule|Text (ChatGPT-3.5)|Text (ChatGPT-4)|Similarity|
> |-|-|-|-|
> | `CC(C)Nc1ncnc2cscc12`      | There are two key transition states involved in the synthesis: the conversion of 2-chloro-N-propan-2-ylthieno[3,4-d]pyrimidin-4-amine to N-propan-2-ylthieno[3,4-d]pyrimidin-4-amine, with the structural change being the displacement of the chlorine atom by the amine group, and the conversion of N-propan-2-ylthieno[3,4-d]pyrimidin-4-amine to N-propan-2-ylthieno[3,4-d]pyrimidin-4-amine, with the structural change being the formation of the final amine functionality; both transitions have energy barriers that must be overcome for the reactions to proceed. | The key transition states involved in the synthesis of N-propan-2-ylthieno[3,4-d]pyrimidin-4-amine from 2-chloro-N-propan-2-ylthieno[3,4-d]pyrimidin-4-amine include a nucleophilic substitution, where the chlorine atom is replaced by an amine group, and a tautomeric shift, where the newly added amine group rearranges to a more stable position on the pyrimidine ring; both transition states involve overcoming energy barriers associated with breaking and forming chemical bonds and achieving the most stable and low energy conformation of the molecule. | These texts by different models describe the same key transition states, including the nucleophilic substitution where the chlorine atom is replaced by an amine group, and a tautomeric shift. Both mention energy barriers involved in the transition states. |
> |`CN=C(O)NCCNCc1cc[nH]n1`| The key transition states involved in the synthesis of 1-methyl-3-[2-(1H-pyrazol-5-ylmethylamino)ethyl]urea from the intermediates N'-(1H-pyrazol-5-ylmethyl)ethane-1,2-diamine include the formation of a urea linkage and the addition of a methyl group. These transitions involve structural rearrangements and energy barriers that facilitate the conversion of the intermediates into the desired product.| The key transition states involved in the synthesis of 1-methyl-3-[2-(1H-pyrazol-5-ylmethylamino)ethyl]urea from N'-(1H-pyrazol-5-ylmethyl)ethane-1,2-diamine include the formation of an isourea intermediate followed by a rearrangement to the final urea product, each with associated energy barriers due to the breaking and forming of bonds and the structural changes from a linear to a more complex, cyclic arrangement.|The mechanisms and transformations described in both are similar, involving the formation of the urea bond and methylation.|
>
> ### 3.Ablation study to clarify the text information from GPT
> We have evaluated the performance of the 1D molecular sequence substituting graph embedding and have carefully revised the Results section of this manuscript. Notably, we adopted the same parameters and conducted identical experiments. As shown in Table below, using only the combination of SMILES and text are inferior to those of a multimodal approach. The use of multimodal representation increases the Top-1 accuracy by 4.6% compared to a unimodal representation, while employing the full set of modalities results in a 5.6% increase.
>
> As shown in the table below, 1D representations like SMILES are compact and simplified but fail to capture critical features such as spatial relationships, bond types, and stereochemistry, leading to the lowest performance.
>
> **Table 13. Ablation study**
> | Model| Top-1 |Top-2 | Top-3 | Top-4 | Top-5 |
> |-|-----------|-----------|-----------|----|-----------|
> | RetroInText (SMILES)| 35.6| 41.6| 44.1| 45.4| 46.2|
> | MolT5 (SMILES)| 37.2| 43.7| 46.2| 47.4| 48.3|
> | RetroInText (Graph)| 37.5| 45.0| 48.2| 50.0| 50.9|
> | RetroInText (w/o text)| 40.2| 47.3| 50.2| 51.7| 52.7|
> | RetroInText| 41.2| 48.7| 51.2| 53.3| 54.2|
>
> We hope this explanation helps clarify the rationale behind our decision to integrate ChatGPT and its role within our proposed approach. Thank you once again for your constructive comments, which have been invaluable in refining the clarity of our work.

---

> ### Author Response · Authors · 2024-11-26
> **Thanks for the review - interpretability analysis (Part X)**
>
> # Response to Question5:
> Thank you for your question and for highlighting the importance of evaluating interpretability and consistency with chemical principles. We used a more stringent evaluation metric to assess our model’s performance, which requires both the predicted intermediate molecules and the final target building block molecules to be identical in order to be considered correct. This stricter criterion ensures that not only are the intermediate steps correctly predicted, but also that the final synthetic pathway is viable and results in the desired product molecule.
>
> By using this evaluation metric, we aim to provide a more rigorous assessment of the model’s ability to generate accurate and meaningful synthetic routes. It also serves as a stronger indicator of the model’s ability to guarantee that the generated molecules can successfully lead to the intended product through a valid synthesis process.
> We would like to highlight that we also integrated our model and scoring function into the Retro* algorithm. The results demonstrate that this integration leads to improved performance, both in terms of higher synthetic route success rates and reduced computational time.
>
> By embedding our model and scoring function into the Retro* framework, we were able to achieve more efficient retrosynthetic planning while maintaining or even improving the accuracy of the predicted synthetic pathways:
>
> **Table 14. Performance comparison between Retro$^*$ and RetroInText on ${Retro}^{*}$ -190 dataset.**
> | Framework         | Success Rate (%) | Avg Time (s) | Avg Iterations |
> |-------------------|------------------|--------------|----------------|
> | Retro*            | 78.42            | 1233.08      | 212.66         |
> | RetroInText (Ours) | 97.89            | 385.53       | 37.92          |
>
> The time reduction, combined with the enhanced performance, underscores the effectiveness of our approach in streamlining the retrosynthesis process and making it more applicable for real-world chemical tasks.
> We believe this integration further strengthens the potential of our framework and offers significant improvements over previous methods. Thank you once again for your valuable feedback.

---

> > ### Comment · Reviewer_xJUs · 2024-11-26
> > **Response to the authors**
> >
> > Dear authors,
> >
> > I really appreciate the comprehensive response. It definitely clears up many of my questions. I also think that you should definitely add these additional experimental analysis and text in the appendix of the paper.
> > Although some of the weakness still remains but given the improved quality of the paper upon the inclusion of these extended materials in the final draft, I am willing to increase my score from 3 to 6 (already updated my score in my original review).
> >
> > Thank you very much!

---

> > > ### Author Response · Authors · 2024-11-27
> > > **Thanks**
> > >
> > > Dear Reviewer xJUs:
> > >
> > > We would like to express our sincere appreciate for your thoughtful feedback and kind recognition of our efforts to address your concerns. We are truly pleased to hear that our responses clarified many of your questions and contributed to improving the overall quality of the paper.
> > >
> > > We greatly value your suggestion to include the additional experimental analysis and text in the appendix, and we have incorporated all the information in the uploaded manuscript. Specifically, we have made the following changes to our paper:
> > >
> > > 1.Added the comparison of different prompts and the experimental results in AppendixC.2.
> > >
> > > 2.Added the comparison of the outputs of GPT-3.5 and GPT-4 in AppendixC.3.
> > >
> > > 3.Added the ablation results in Table3 and make the detail explanation in AppendixC.4.
> > >
> > > These changes have been highlighted in red for easy review. Your constructive input has been invaluable in refining our work, and we are deeply grateful for your willingness to re-evaluate and raise your score.
> > >
> > > Once again, thank you for your insightful comments and for taking the time to engage with our work. Your feedback has been instrumental in strengthening this research, and we truly appreciate your support.

---

### Official Review · Reviewer_zdz7 · 2024-11-04

**Soundness:** 2
**Presentation:** 3
**Contribution:** 3
**Rating:** 8
**Confidence:** 4

**Summary:**

The authors target a more realistic retrosynthesis scenario, considering multi-step synthesis and cost function. Solving retrosynthesis planning and multi-step reasoning about the intermediate is important, which current LLM-based methods overlook. To tackle this, the authors suggest an E2E framework involving learnable multimodal LLM utilizing a 3D molecule encoder and (possibly closed source) LLM generating text descriptions for intermediate synthesis steps. By fusing the multimodal information via the proposed attention mechanism and leveraging LLM generated reaction descriptions, the proposed method outperformed semi-template-based and template-free SOTA methods and recorded comparable performance to the SOTA template-based method in RetroBench.

**Strengths:**

- The authors proposed a principled way to leverage LLM in retrosynthesis using multiple modalities, including molecular structure (2D graph and 3D conformation) and text descriptions of chemical reactions.
- The authors' method of generating text descriptions and ranking candidate reactants for multi-step retrosynthesis would inspire many future researches.
- Until now, LLM-based retrosynthesis methods have primarily focused on single-step retrosynthesis, leaving few studies available for the potentially more significant multi-step retrosynthesis in the domain. Additionally, they have not considered the reaction cost, which is crucial in real-world reaction pathways. By conducting multi-step retrosynthesis based on a value function that takes into account a cost function, the authors have narrowed the gap between LLM-based retrosynthesis and real-world applications.
- The proposed method outperformed not only semi-template-based and template-free methods but also most template-based methods. This is interesting because the performance is achieved largely based on LLM’s generalization via ICL and reasoning ability, without relying on inductive bias, including reaction template, so with further future potential.

**Weaknesses:**

- The motivation to maximize mutual information of 2D graph and 3D conformation is not sufficiently presented. If one can use 3D conformation, which is generally believed to have more information than a 2D graph, what is additional information from the 2D graph? It seems that Table 3 does not resolve this matter.
- It is necessary to add the ablation of 1D molecular sequence substituting graph embedding to demonstrate the supremacy of spatial representation, considering that the molecular sequence is generally regarded as a more advantageous modality than 2D graph and 3D conformation. Could you add the ablation study?
- I wonder how much the proposed framework is sensitive to the quality of textual description and what quality control process would help (GPT-3.5 could be quite old these days for SOTA-level performance achievement), which I found no available analysis related. Could you provide an analysis related to this matter?

**Questions:**

- In section 4.2, Textual Generator and Encoder, there is a confusing explanation related to the use of textual description. The authors used textual descriptions of the intermediate molecules along all pathways but, in test time, only used the text description of the product molecules to prevent information leakage, which is a train-test mismatch. But why is this data leakage, given that ChatGPT is not trained during training time?

---

> ### Author Response · Authors · 2024-11-25
> **Thanks for the review - justification for 2D-3D mutual information and 1D sequence ablation study (Part I)**
>
> Dear Reviewer zdz7,
>
> We want to extend our heartfelt thanks for your time and give constructive suggestions. We have carefully considered each suggestion when revising this manuscript.Please see our point-to-point response to your concerns below.
> # Response to Weakness1:
> Thank you for your question. We are very sorry for unclearly explaining the overall framework of our RetroInText architecture. Firstly, our model RetroInText was designed as a multimodal enhanced framework for retrosynthetic planning. Current machine learning-based methods often overlook the multifaceted nature of compounds, resulting in inaccurate retrosynthesis predictions. However, the process of constructing a comprehensive molecular representations  often relatively complex for the learning of compounds feature, particularly when dealing with numerous aspects of molecular features. Consequently, we aim to incorporate compound structure from a different perspective. Therefore, we applied the SMILES, 2D molecular, and 3D conformation feature as multimodal embedding.
>
> Secondly, 2D structures adding the feature of molecular that 3D conformation do not have. Specifically, 2D structures explicitly represent chemical bond types (e.g., single, double, triple, or conjugated bonds), such as aromaticity and conjugation, which are crucial for understanding reactivity in retrosynthesis, are often clearly indicated in 2D diagrams using specific symbols or markings.
>
> Additionally, 2D structures also include stereochemical symbols, such as wedges and dashed bonds, which represent the absolute configuration of chiral centers and the stereochemistry of molecules. These symbols provide immediate, unambiguous information about the stereochemistry, whereas in 3D models, this information is embedded in the spatial arrangement and may require additional interpretation to ascertain the absolute configuration.
>
> # Response to Weakness2:
> Thank you for your kind suggestions regarding model ablation studies. We are very sorry for unclearly demonstrating our experimental results. We agree is necessary to add the ablation study of 1D molecular sequence.
>
> We have evaluated the performance of the 1D molecular sequence substituting graph embedding and have carefully revised the Results section of this manuscript. Notably, we adopted the same parameters and conducted identical experiments. As shown in Table below, using only the combination of SMILES and text are inferior to those of a multimodal approach. The use of multimodal representation increases the Top-1 accuracy by 4.6% compared to a unimodal representation, while employing the full set of modalities results in a 5.6% increase.
>
> As shown in the table below, 1D representations like SMILES are compact and simplified but fail to capture critical features such as spatial relationships, bond types, and stereochemistry, leading to the lowest performance.
>
> **Table 1. Ablation study**
> | Model                  | Top-1 |Top-2 | Top-3 | Top-4 | Top-5 |
> |----------------------------|-----------|-----------|-----------|-----------|-----------|
> | RetroInText (SMILES)       | 35.6      | 41.6      | 44.1      | 45.4      | 46.2      |
> | MolT5 (SMILES)             | 37.2      | 43.7      | 46.2      | 47.4      | 48.3      |
> | RetroInText (Graph)        | 37.5      | 45.0      | 48.2      | 50.0      | 50.9      |
> | RetroInText (w/o text)     | 40.2      | 47.3      | 50.2      | 51.7      | 52.7      |
> | RetroInText                | 41.2      | 48.7      | 51.2      | 53.3      | 54.2      |

---

> ### Author Response · Authors · 2024-11-25
> **Thanks for the review - sensitivity to text quality and analysis of quality control for textual descriptions (Part II)**
>
> # Response to Weakness3:
> Thank you so much for you questions. We appreciate the opportunity to clarify this aspect of our work. To address the concern about the sensitivity of the proposed framework to the quality of textual descriptions, we conducted analyses from two perspectives: 1. the influence of different prompts on text generation and 2. a comparison between GPT-3.5 and GPT-4 outputs.
>
> ## 1. Impact of different prompts on text quality and results
>
> We evaluated how variations in prompts influence the quality of generated text and, consequently, the model's performance. Specifically, we tested two distinct prompts, as shown in Table 2, and analyzed their effects on textual descriptions for the molecule O=C1N=C(O)CN1N=C(O)C=Cc1ccc(Cl)c(C(F)(F)F)c1. The first prompt was designed to elicit detailed but overly verbose text, while the second prompt aimed for concise and focused textual descriptions:
>
> - **Prompt 1:** This prompt lacked specificity, generating verbose descriptions with excessive redundancy, which introduced irrelevant details that could negatively impact the model's accuracy.
> - **Prompt 2:** This prompt was carefully crafted to control the output, emphasizing clarity and relevance while avoiding unnecessary or redundant information. The generated text was compact and structured, which improved the downstream model performance.
>
> As shown in Table 3, using text generated from the high-quality prompt significantly improved the performance of the framework. For example, the Top-1 accuracy increased from 39.2% to 41.2%, and the Top-5 accuracy rose from 46.9% to 54.2%. This improvement demonstrates the importance of designing prompts that generate concise and high-quality text for better model predictions.
>
> **Table 2. Comparison of prompt influence on textual quality for molecular descriptions and reaction pathways**
> |Target Molecule|Prompt|Text|
> |-|-|-|
> |`O=C1N=C(O)CN1N=C(O)C=Cc1ccc(Cl)c(C(F)(F)F)c1`|Please describe the structural features, functional aspects, and applicable implementation methods of the molecule {{product}}, and describe the potential intermediate molecules {{intermediates}} in the synthesis pathway, with each intermediate molecule separated by "[SEP]". Additionally, provide a different reaction pathway and describe it.  | The molecule 3-[4-chloro-3-(trifluoromethyl)phenyl]-N-(2,4-dioxoimidazolidin-1-yl)prop-2-enamide is a complex organic compound that features a substituted phenyl group, an imidazolidin-1-yl moiety, and a prop-2-enamide functional group. Structural features: 1. Substituted phenyl group with a chlorine atom and a trifluoromethyl group, 2. Imidazolidin-1-yl ring with two carbonyl groups, 3. Prop-2-enamide functional group Functional aspects: 1. The substituted phenyl group provides aromaticity and potentially affects the molecule's reactivity and solubility. 2. The imidazolidin-1-yl ring may participate in hydrogen bonding interactions or act as a moiety for stabilization. 3. The prop-2-enamide functional group can participate in conjugation reactions or undergo nucleophilic addition reactions. Applicable implementation methods: 1. Synthesis of the molecule may involve various organic chemistry techniques such as Grignard reactions, amide bond formation, and cyclization reactions. 2. Purification techniques such as column chromatography, recrystallization, or distillation may be used to obtain the final product in high purity. Potential intermediate molecules: 1. 3-[4-chloro-3-(trifluoromethyl)phenyl]prop-2-enoic acid Different reaction pathway: Instead of directly synthesizing the target molecule, an alternative pathway might involve the initial formation of 3-[4-chloro-3(trifluoromethyl)phenyl]prop-2-enoic acid as an intermediate. Subsequently, the prop-2-enoic acid can be converted to the desired prop-2-enamide through an amide coupling reaction with the imidazolidin-1-yl moiety. Overall, the synthesis of such complex molecules requires careful planning, knowledge of organic chemistry reactions, and proper characterization techniques to confirm the structure of intermediate and final products. |
> |`O=C1N=C(O)CN1N=C(O)C=Cc1ccc(Cl)c(C(F)(F)F)c1`|Describe the key transition states involved in the synthesis of {{products}} from the intermediates {{intermediates}}. Explain the structural changes and energy barriers for each transition state, and reply to me in a sentence.  | The key transition states involved in the synthesis include the formation of an imine intermediate through the condensation of the prop-2-enoic acid and the amine group, followed by the cyclization of the imine to form the imidazolidin-1-yl ring. The structural changes involve the addition of the amine group and the formation of a new cyclic structure, with energy barriers reflecting the activation energy needed for the formation of the new bonds in each step.|
> (to be continued)

---

> ### Author Response · Authors · 2024-11-25
> **Thanks for the review - sensitivity to text quality and analysis of quality control for textual descriptions (Part III)**
>
> **Table 3. Performance comparison of prompt influence on textual quality for molecular descriptions and reaction pathways**
> | Model| Top-1| Top-2| Top-3| Top-4| Top-5|
> |-|-|-|-|-|-|
> | RetroInText (prompt1)| 39.2| 43.4| 45.6| 45.7| 46.9|
> | RetroInText (prompt2)| 41.2| 48.7| 51.2| 53.3| 54.2|
>
> ## 2. Comparison between GPT-3.5 and GPT-4 outputs
>
> We applied GPT-4 for text generation. The texts generated by GPT-4 and GPT-3.5 were highly similar in terms of content and the structural transformations they detailed, as shown in the examples provided (Table 4). For instance, both models accurately described key transition states for the target molecules, such as nucleophilic substitutions, tautomeric shifts, and energy barriers associated with bond-breaking and forming processes.
>
> However, GPT-4 posed practical challenges, including frequent API key limitations, which disrupted the workflow and made it less reliable for consistent use. In contrast, GPT-3.5 offered stable performance without such restrictions, making it a more practical choice for our framework. Given the negligible performance differences and GPT-4's limitations, GPT-3.5 was selected as the primary text generator for our experiments.
>
> **Table 4. Comparison of textual descriptions generated by ChatGPT-3.5 and ChatGPT-4 for key transition states in molecular synthesis**
>
> |Target Molecule|Text (ChatGPT-3.5)|Text (ChatGPT-4)|Similarity|
> |-|-|-|-|
> | `CC(C)Nc1ncnc2cscc12`      | There are two key transition states involved in the synthesis: the conversion of 2-chloro-N-propan-2-ylthieno[3,4-d]pyrimidin-4-amine to N-propan-2-ylthieno[3,4-d]pyrimidin-4-amine, with the structural change being the displacement of the chlorine atom by the amine group, and the conversion of N-propan-2-ylthieno[3,4-d]pyrimidin-4-amine to N-propan-2-ylthieno[3,4-d]pyrimidin-4-amine, with the structural change being the formation of the final amine functionality; both transitions have energy barriers that must be overcome for the reactions to proceed. | The key transition states involved in the synthesis of N-propan-2-ylthieno[3,4-d]pyrimidin-4-amine from 2-chloro-N-propan-2-ylthieno[3,4-d]pyrimidin-4-amine include a nucleophilic substitution, where the chlorine atom is replaced by an amine group, and a tautomeric shift, where the newly added amine group rearranges to a more stable position on the pyrimidine ring; both transition states involve overcoming energy barriers associated with breaking and forming chemical bonds and achieving the most stable and low energy conformation of the molecule. | These texts by different models describe the same key transition states, including the nucleophilic substitution where the chlorine atom is replaced by an amine group, and a tautomeric shift. Both mention energy barriers involved in the transition states. |
> |`CN=C(O)NCCNCc1cc[nH]n1`| The key transition states involved in the synthesis of 1-methyl-3-[2-(1H-pyrazol-5-ylmethylamino)ethyl]urea from the intermediates N'-(1H-pyrazol-5-ylmethyl)ethane-1,2-diamine include the formation of a urea linkage and the addition of a methyl group. These transitions involve structural rearrangements and energy barriers that facilitate the conversion of the intermediates into the desired product.| The key transition states involved in the synthesis of 1-methyl-3-[2-(1H-pyrazol-5-ylmethylamino)ethyl]urea from N'-(1H-pyrazol-5-ylmethyl)ethane-1,2-diamine include the formation of an isourea intermediate followed by a rearrangement to the final urea product, each with associated energy barriers due to the breaking and forming of bonds and the structural changes from a linear to a more complex, cyclic arrangement.|The mechanisms and transformations described in both are similar, involving the formation of the urea bond and methylation.|

---

> ### Author Response · Authors · 2024-11-25
> **Thanks for the review - justification for train-test mismatch and leakage (Part IV)**
>
> # Response to Question:
> Thank you for the question. We sincerely apologize for any ambiguity in our presentation. Our prompt is designed to have GPT generate structural information for intermediate molecules along a synthetic pathway. However, our current textual information is derived from real reaction pathways. Incorporating intermediate molecule information during testing would introduce real pathway details into the textual information, leading to data leakage. To address this, we’ve taken steps to ensure that the generated text does not include real-world pathway information, making this approach transferable to situations where real-world pathway data is unavailable. In such cases, no real intermediate molecules will be present, allowing the model to focus exclusively on generating synthetic intermediates relevant to the task.

---

> > ### Comment · Reviewer_zdz7 · 2024-11-26
> >
> > Thanks for the authors' response.
> >
> > Reading the responses resolved most of my concerns, except weakness 1.
> >
> > Regarding weakness 1, I agree that using 2D molecular graphs could be jointly beneficial to using 3D conformation, but there remains a question mark about whether maximization of MI between 2D features and 3D features is optimal design, given that the overall performance gain is insignificant and the choice of MI maximization does not come from an ablation study.
> >
> > However, I appreciate other contributions outweigh the question mark in the modality fusion, because this work pioneered the use of MLLM in this domain with propositions of proper framework and moderate performance gain.
> >
> > Overall, I choose to raise the score.

---

> ### Author Response · Authors · 2024-11-26
> **Thanks for the review - justification for usage of 3DInfomax**
>
> Thank you for taking the time to carefully review our responses and for your thoughtful feedback. We are very glad to hear that our responses resolved most of your concerns, and we deeply appreciate your recognition of the contributions of our work and the improvements made during the rebuttle process.
>
> We sincerely apologize for not for the confusion regarding the maximization of mutual information (MI) between 2D and 3D features. To address the concern, we carefully (1) conducted a literature review of molecular representation models, and (2) we performed additional experiments.
>
> Firstly, we select widely adopted and extensively benchmarked pretrained molecular representation models, including embeddings from different modalities, to extract molecular features. Specifically, we consider models such as 3DInfomax which maximized the mutual information between 3D conformation and 2D representations of a graph neural network [1], GraphMAE with 2D information which didn’t rebuild the graph structure and simultaneously using masking strategy for feature reconstruction[2], SimSRT which equipped with a Simple GNN-based Tokenizer (SGT) based on simple GNN and effective decoding strategy[3],  Mole-BERT proposed Masked Atoms Modeling(MAM), to mask some discrete codes randomly and then pre-train GNNs to predict them[4], and Chemformer which used masking, augmentation and combination of this two method to pre-train the model[5].
>
> Secondly, to conducted the experiment, we randomly select a set of 100 molecules for testing. And we replaced the molecular representaion module with above mentioned pre-trained model during the pathway selection phase. Specifically, we used the multilayer perceptron (MLP) model to ensure fairness of the comparison.
> As shown in the Table 1,  3DInfomax consistently outperforms other embeddings across the exact match accuracy. It is important to note that we used the pre-trained 3DInfomax model as the molecular embedding generator in our experiments without fine-tuning or retraining the model. This choice was made to focus on leveraging the high-quality molecular embeddings provided by 3DInfomax, while avoiding the computational overhead associated with retraining.
> However, we acknowledge that further exploration. Your insight raises an important point for further exploration, such as conducting more comprehensive molecular feature embedding modality fusion. This constructive feedback will be invaluable as we continue to refine our methodology, could potentially yield more significant performance gains.
>
> **Table 1: Performance Comparison of Molecular Embedding Models Across Top-k Accuracy**
>
> | Model      | Top 1 | Top 2 | Top 3 | Top 4 | Top 5 |
> |------------|-------|-------|-------|-------|-------|
> | **3DInfomax**  | 0.24  | 0.34  | 0.35  | 0.37  | 0.37  |
> | **SimSGT**     | 0.04  | 0.08  | 0.12  | 0.14  | 0.16  |
> | **GraphMAE**   | 0.03  | 0.06  | 0.09  | 0.16  | 0.18  |
> | **Mole-Bert**  | 0.04  | 0.07  | 0.09  | 0.12  | 0.18  |
> | **Chemformer** | 0.05  | 0.06  | 0.08  | 0.12  | 0.14  |
>
> We hope our responses meet your expectations. Once again, we sincerely thank you for your thoughtful evaluation and constructive suggestions. We would like to express our heartfelt gratitude for your kind recognition of our efforts and for raising score. Your insightful comments and suggestions have been truly helpful in improving our work.
>
> Let us know if anything is missing or confusing. We are happy to further discuss and keep improving our manuscript!
>
> # References
>
> [1] Stärk H, Beaini D, Corso G, et al. 3D Infomax Improves GNNs for Molecular Property Prediction[C]//International Conference on Machine Learning. PMLR, 2022: 20479-20502.
> [2] Hou Z, Liu X, Cen Y, et al. GraphMAE: Self-supervised Masked Graph Autoencoders[C]//Proceedings of the 28th ACM SIGKDD Conference on Knowledge Discovery and Data Mining. 2022: 594-604.
> [3] Liu Z, Shi Y, Zhang A, et al. Rethinking Tokenizer and Decoder in Masked Graph Modeling for Molecules[J]. Advances in Neural Information Processing Systems, 2024, 36.
> [4] Hou Z, Liu X, Cen Y, et al. GraphMAE: Self-supervised Masked Graph Autoencoders[C]//Proceedings of the 28th ACM SIGKDD Conference on Knowledge Discovery and Data Mining. 2022: 594-604.
> [5] Irwin R, Dimitriadis S, He J, et al. Chemformer: A Pre-trained Transformer for Computational Chemistry[J]. Machine Learning: Science and Technology, 2022, 3(1): 015022.

---

> > ### Comment · Reviewer_zdz7 · 2024-11-27
> >
> > I appreciate the authors' sincere effort for author-reviewer discussion and agile additional response.
> >
> > Reading the additional responses, my concern about weakness 1 was resolved enough.
> >
> > At the same time, as the authors respond, I hope future works conduct more extensive exploration of modality fusion, for the development of this domain and active application of PLM for chemical downstream tasks.
> >
> > For the sake of the community, same as the opinion of reviewer BszM, please consider open-sourcing the code upon acceptance. Thanks.

---

> > > ### Author Response · Authors · 2024-11-27
> > > **Thanks**
> > >
> > > Dear Reviewer zdz7:
> > >
> > > We sincerely thank you for your thoughtful feedback and it is incredibly encouraging to know that our responses resolved all your concerns.
> > >
> > > Firstly, we wholeheartedly agree that further exploration of modality fusion and the application of PLMs for chemical downstream tasks represents a critical direction for advancing this field. Your insightful suggestions have significantly shaped our perspective, and we are committed to pursuing these directions in our future work.
> > >
> > > Secondly, We are pleased to confirm that we will make our code publicly available to support further research once the paper is accepted.
> > >
> > > Thank you once again for your valuable comments and support. Your guidance has been constrictive in improving the quality of our research.

---

### Author Response · Authors · 2024-11-25
**Initial response to all reviewers**

Thank you all for your valuable comments and insightful feedback. We have revised our manuscript accordingly and uploaded the updated version along with a detailed, point-by-point response addressing each reviewer’s specific questions and suggestions. In this section, we would like to summarize the feedback, our corresponding responses, and the changes we have made to improve the paper.

## Compliments:
We are grateful for the many positive comments provided by the reviewers, including recognition of our novel and well-founded objective, a clear explanation of our proposed algorithm, and the strong experimental results that effectively support the claims made in our paper. We truly appreciate the constructive and encouraging tone of the reviews!

## Revision:
Based on reviewer comments, we have conducted additional experiments to further validate the model, and we highlight the advantage of text generation, as it enables the production of contextually relevant and coherent responses, improving the adaptability and flexibility of our model in handling Retrosynthetic planning. We have made the following changes to our paper:
1. Revised Section 4.2 to improve the explanation of the search algorithm, adding a detailed explanation of the three different scores we use.
2. Clarified the evaluation metrics and their interpretability in Appendix A.1.
3. Added our experimental setup including the environment and inference time details in Appendix A.2.
4. We have conducted additional experiments to further validate the advantage of in-context description. Additionally, we've conducted further tests with the reranking mechanism to validate our model's accuracy.

These changes have been highlighted in red for easy identification. We believe that the revisions have significantly strengthened the paper and effectively address all the concerns raised by the reviewers. Should there be any further issues, please let us know, and we will make every effort to address them during the rebuttal period. If no further concerns remain, we kindly ask the reviewers to reconsider their scores.

---

### Meta-Review · Area_Chair_bYSg · 2024-12-24

**Metareview:**

The paper introduces RetroInText, a multimodal framework for retrosynthetic planning. Specifically, RetroInText performs the following two tasks:
- For single-step retrosynthesis prediction, it fine-tunes MolT5 by introducing the 2D/3D/text modalities, aiming to improve accuracy in predicting the immediate precursor of a target molecule.
- For the search algorithm, it introduces a synthetic complexity score, reaction cost (calculated as the negative log-likelihood of a retrosynthesis prediction), and the captioning score to identify the most promising reactants.

There are two major strengths offered by this work:
- A new multimodal foundation model for retrosynthesis, particularly by **introducing text descriptions**.
- The **inclusion of the captioning score** and reaction cost (although the latter is not new, as seen in "Active Retrosynthetic Planning Aware of Route Quality" by Yuan et al., ICLR 2024) to evaluate the most promising reactants during the search process.

Due to the above two merits, I recommend acceptance at the current stage, though not with full confidence due to the following reasons:
- While the authors evaluated multi-step retrosynthesis or retrosynthetic planning, this work primarily contributes to single-step retrosynthesis and relies on existing planning methods such as Retro*/Retro*-0. The only addition is the captioning score, but empirical results fail to justify its substantial contribution to planning itself. In this sense, I agree with Reviewer cGGk that the title and introduction are somewhat misleading, and the authors should address this if the paper is accepted.
- A key point of discussion between Reviewer cGGk and Reviewer BszM during the response period concerns the evaluation metric of the set-wise matching score adopted in this work. While Reviewer BszM considers this a strength, this metric does not strongly advocate for the work's contribution to multi-step planning: (1) it follows FusionRetro (Liu et al., 2023a) and is not a novel contribution by the authors; (2) though it evaluates set matching, it still depends on (potentially over-)fitting the concatenated routes from single-step reactions. As long as single-step predictions are well-fitted, even a naive multi-step search algorithm like Greedy DFS can achieve a good exact matching score. In this sense, I again believe this work's major contributions have little to do with multi-step planning.
- As pointed out by three reviewers, this work lacks sufficient interpretation of why the proposed method improves and how the additional text modality concretely contributes to the final performance improvement.

In summary, while this paper introduces interesting contributions to single-step retrosynthesis, its claims regarding multi-step retrosynthetic planning are not sufficiently justified, and the authors should address this if this paper gets accepted.

**Additional Comments On Reviewer Discussion:**

During the discussion period, all reviewers actively engaged with the authors. Except for Reviewer cGGk, the other reviewers expressed satisfaction with the responses provided. Reviewer cGGk's remaining concern primarily revolves around how the additional text modality concretely contributes to the observed performance improvements.

The strengths and weaknesses of this work have been summarized above, leading to my current recommendation for acceptance, albeit with moderate confidence.

---

### Decision · Program_Chairs · 2025-01-22

Accept (Poster)